# FLASHSKETCH: Sketch-Kernel Co-Design for Fast Sparse Sketching on GPUs

**Rajat Vadiraj Dwaraknath** [1]  **Sungyoon Kim** [2]  **Mert Pilanci** [2]

## Abstract

Sparse sketches such as the sparse Johnson–Lindenstrauss transform are a core primitive in randomized numerical linear algebra because they leverage random sparsity to reduce the arithmetic cost of sketching, while still offering strong approximation guarantees. Their random sparsity, however, is at odds with efficient implementations on modern GPUs, since it leads to irregular memory access patterns that degrade memory bandwidth utilization. Motivated by this tension, we pursue a sketch–kernel co-design approach: we design a new family of sparse sketches, BLOCKPERM-SJLT, whose sparsity structure is chosen to enable FLASHSKETCH, a corresponding optimized CUDA kernel that implements these sketches efficiently. The design of BLOCKPERM-SJLT introduces a tunable parameter that explicitly trades off the tension between GPU-efficiency and sketching robustness. We provide theoretical guarantees for BLOCKPERM-SJLT under the oblivious subspace embedding (OSE) framework, and also analyze the effect of the tunable parameter on sketching quality. We empirically evaluate FLASHSKETCH on standard RandNLA benchmarks, as well as an end-to-end ML data attribution pipeline called GraSS. FLASHSKETCH pushes the Pareto frontier of sketching quality versus speed, across a range of regimes and tasks, and achieves a global geomean speedup of roughly $1.7\times$ over the prior state-of-the-art GPU sketches.

## 1. Introduction

Randomized sketching is a standard mechanism to reduce the dimension of a large collection of vectors while ap-proximately preserving their geometry. This dimensionality reduction enables running downstream computational algorithms faster, using less memory, and with lower communication costs (Woodruff et al., 2014; Clarkson & Woodruff, 2017; Martinsson & Tropp, 2020). The canonical sketch is the Johnson–Lindenstrauss (JL) transform, which projects $n$ points in $d$ dimensions down to $k = O(\varepsilon^{-2} \log n)$ dimensions while preserving pairwise distances up to a $(1 \pm \varepsilon)$ factor (Johnson & Lindenstrauss, 1984; Freksen, 2021). It can be implemented by multiplying the input matrix $A \in \mathbb{R}^{d \times n}$ of $n$ vectors in $\mathbb{R}^d$ by a random matrix $S \in \mathbb{R}^{k \times d}$, where $S_{ij} \sim \mathcal{N}(0, 1/k)$. $S$ is the sketching matrix. This operation is at the core of classical randomized numerical linear algebra (RandNLA) algorithms for least squares, low-rank approximation, regression, etc. More recently, it has found use in end-to-end ML pipelines such as data attribution (Hu et al., 2025). Sparse JL transforms (SJLT) and related constructions such as OSNAP (Dasgupta et al., 2010; Nelson & Nguyen, 2013; Kane & Nelson, 2014) reduce the arithmetic cost of applying $S$ by imposing sparsity on $S$, while retaining the geometric guarantees that make JL useful. Fundamentally, the power of sketches originates from their randomness, which leads to strong concentration phenomena, enabling *subspace-oblivious* embedding guarantees.

Modern computing hardware is increasingly heterogeneous, with GPUs playing a central role in large-scale numerical linear algebra and machine learning workloads. Performing sketching efficiently on GPUs is therefore essential to practically realizing the end-to-end speedups promised by RandNLA algorithms. Optimizing workloads on GPUs requires careful attention to the memory hierarchy and parallel execution model (Nvidia, 2011), and does not always align with CPU-centric cost models that usually focus on reducing arithmetic complexity. In particular, sparse matrix multiplication (SpMM) is a well-studied primitive on CPUs, but achieving high performance for SpMM on GPUs is challenging due to irregular memory access patterns and atomic contention (Yang et al., 2018). A fundamental tension therefore arises:

*The very randomness that underpins the theoretical strength of sparse sketches also disrupts the structure required for their efficient implementations on modern hardware.*

This motivates our **co-design thesis**:

---
[1]Institute for Computational and Mathematical Engineering (ICME), Stanford University [2]Department of Electrical Engineering, Stanford University. Correspondence to: Rajat Vadiraj Dwaraknath <rajatvd@stanford.edu>.

*Proceedings of the 43$^{rd}$ International Conference on Machine Learning*, Seoul, South Korea. PMLR 306, 2026. Copyright 2026 by the author(s).

*Balancing this tension requires co-designing sparse sketches and their corresponding GPU kernels.*

In this work, we pursue this co-design thesis by designing a sparse sketch with *hardware-informed structured randomness*, allowing us to implement a specialized CUDA kernel, FLASHSKETCH, that can better balance the tension between sketch quality and realized speed, pushing the Pareto frontier for sketching on GPUs (see Figure 1 for a preview).

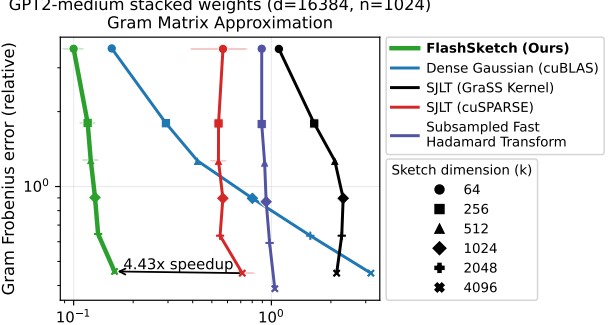

*Figure 1.* **Quality–speed Pareto frontier (preview).** We compare sketching methods by runtime on an NVIDIA RTX 4090 GPU (x-axis) and quality measured using error in Gram matrix approximation (y-axis). **Lower is better for both axes.** FLASHSKETCH pushes the Pareto frontier.

**Key idea: Structured Sparsity for Block Locality and Mixing.** Our key idea is to restrict the sparsity pattern of SJLT to simultaneously enable block locality and sufficient mixing. We achieve this by designing a block-structured SJLT in which the block-level sparsity pattern is a union of edge-disjoint permutations. We call the resulting sketch family BLOCKPERM-SJLT. Within each selected block, we maintain the standard SJLT sparsity to provide fine-grained mixing.

We implement this sketch with FLASHSKETCH, a CUDA kernel that streams tiles of the input $A$ through shared memory, accumulates an output tile privately in shared memory (using shared-memory atomics), and writes each output tile to global memory once. This kernel avoids global atomic operations during the accumulation phase, which are a major bottleneck in prior GPU SJLT implementations (Hu et al., 2025; Higgins et al., 2025). This is enabled by the bi-regular structure of the union-of-permutations block-sparsity graph, which implies that each output block can be assigned to a *single CUDA thread-block* without overlap. This key kernel optimization enabled by the sketch structure emphasizes our co-design thesis.

**Contributions.**

- **A hardware-informed sparse sketch family.** In Section 4, we introduce a new sparse sketch, BLOCKPERM-SJLT, a restricted SJLT family that preserves block locality while improving mixing via a union-of-permutations block wiring.
- **A specialized CUDA kernel.** In Section 5 we present FLASHSKETCH, a thread-block-tiled kernel that applies BLOCKPERM-SJLT with thread-block-local accumulation using shared-memory atomics and a single global write per output tile, as well as techniques for on-the-fly randomness generation.
- **Theory via permutation-coupled localization.** In Section 6, building on localized sketching (Srinivasa et al., 2020), we prove an oblivious subspace embedding style guarantee controlled by a *neighborhood-coherence* quantity induced by the permutation wiring. Further, under a simplified model of independent random permutations, we show that the number of permutations $\kappa$ controls the neighborhood-coherence, providing a tunable tradeoff between sketch quality and speed.
- **Benchmarks and an ML pipeline.** In Section 7, we benchmark FLASHSKETCH on standard RandNLA tasks and an end-to-end ML pipeline for data attribution called GraSS (Hu et al., 2025). We compare against strong dense and sparse baselines, and our experiments demonstrate that FLASHSKETCH pushes the quality–speed Pareto frontier by achieving a global geomean speedup of roughly $1.7\times$ while preserving quality over a range of regimes and tasks.

We release all our code in the following GitHub repository: https://github.com/rajatvd/flash-sketch-arxiv.

## 2. Related Work

**Sketching in RandNLA.** Random projections for dimensionality reduction trace back to the seminal Johnson–Lindenstrauss lemma and are now foundational in RandNLA (Johnson & Lindenstrauss, 1984; Drineas & Mahoney, 2016; Martinsson & Tropp, 2020). Theoretical guarantees for random projections are often framed in the oblivious subspace embedding (OSE) model developed in (Clarkson & Woodruff, 2017). A thorough survey of oblivious subspace embeddings and sketching in general can be found in (Woodruff et al., 2014). Sparse embeddings such as CountSketch, SJLT, and OSNAP reduce application cost by constructing sparse sketching matrices that still possess guarantees under the OSE framework with sparsity requirements (Dasgupta et al., 2010; Nelson & Nguyen, 2013; Kane & Nelson, 2014; Charikar et al., 2004).

**Structured and localized sketches.** A long line of work designs structured embeddings that trade randomness for fast application, such as the subsampled Randomized

Hadamard Transform (SRHT) and related Hadamard-based transforms (Ailon & Chazelle, 2009). Another approach to constructing sparse sketches is through the lens of expander graphs, and there is a rich literature on using expander-based constructions for JL and OSE guarantees (Hoory et al., 2006; Jafarpour et al., 2009; Berinde et al., 2008). In particular, (Puder, 2015) explores constructions of expanders using union-of-permutations approaches. Localized sketching (Srinivasa et al., 2020) makes the locality–distortion trade-off explicit: block-diagonal (or block-local) sketches can be analyzed in terms of how the target subspace distributes across a chosen partition. They show OSE guarantees for such block-diagonal SJLTs that depend on a natural *block-coherence* parameter of the input subspace. Our sketch, BLOCKPERM-SJLT, is a generalization of this block-local SJLT construction using a union-of-permutations structure at the block level to improve mixing while retaining block locality.

**GPU implementations of sketches.** For dense sketches, GPUs benefit from mature GEMM kernel libraries like cuBLAS. Recent work has targeted tensor cores explicitly for random projection and mixed-precision RandNLA (Ootomo & Yokota, 2023; Carson & Daužickaitė, 2025). On the sparse side, kernels are less mature than their dense counterparts. (Hu et al., 2025) present an open-source CUDA implementation of SJLT. (Higgins et al., 2025) develop a high-performance CountSketch kernel for streaming data applications in HPC, though do not release code. Both implementations follow the *scatter-add* pattern and rely on global atomics to handle concurrent writes to the same output row. For Hadamard-based sketches, the state of GPU kernels is more mature, with (Agarwal et al., 2024) providing an open source tensor-core-native implementation of the Fast Hadamard Transform for 16-bit precision, and (Dao-AILab, 2024) providing a general FHT kernel for FP32. A distributed block SRHT implementation appears in (Balabanov et al., 2022).

# 3. Background

### 3.1. Notation

We write $A \in \mathbb{R}^{d \times n}$ for the input matrix and $S \in \mathbb{R}^{k \times d}$ for a random sketch, with input dimension $d$, sketch dimension $k$, and number of vectors $n$. The sketched output is $Y := SA \in \mathbb{R}^{k \times n}$. We use $\|\cdot\|_2$ for the spectral norm and $\|\cdot\|_F$ for the Frobenius norm.

### 3.2. Johnson–Lindenstrauss and Oblivious Subspace Embeddings (OSEs)

The Johnson–Lindenstrauss (JL) lemma states that for *any* set of $n$ points $\{x_i\}_{i=1}^n \subset \mathbb{R}^d$ and distortion $\varepsilon \in (0, 1)$, a random projection into dimension $k = \mathcal{O}(\varepsilon^{-2} \log n)$ pre-

serves all pairwise distances up to $(1 \pm \varepsilon)$ multiplicative factors with high probability (Johnson & Lindenstrauss, 1984; Freksen, 2021). For this work, we focus on the *oblivious subspace embedding* (OSE) model, which demands guarantees for all vectors in an arbitrary $n$-dimensional subspace simultaneously (Clarkson & Woodruff, 2017; Woodruff et al., 2014).

**Definition 3.1** (OSE)**.** A distribution over $S \in \mathbb{R}^{k \times d}$ is an $(\varepsilon, \delta, n)$ OSE if for all orthonormal $U \in \mathbb{R}^{d \times n}$,

$$\mathbb{P}\Big[ \|U^\top S^\top S U - I_n\|_2 \le \varepsilon \Big] \ge 1 - \delta. \qquad (1)$$

OSEs are a useful model for sketching because they yield strong theoretical guarantees for downstream tasks like least squares and regression (Clarkson & Woodruff, 2017; Martinsson & Tropp, 2020). The canonical example is a dense Gaussian sketch with $S_{ij} \sim \mathcal{N}(0, 1/k)$, which is an $(\varepsilon, \delta, n)$ OSE with sketch dimension $k = \mathcal{O}(\varepsilon^{-2}(n + \log(1/\delta)))$. Similarly, dense Rademacher sketches with $S_{ij} \sim \mathrm{Unif}\{\pm 1/\sqrt{k}\}$ are also OSEs with similar parameters (Woodruff et al., 2014).

### 3.3. Sparse JL Transforms and OSNAP

Sparse sketches aim to reduce the cost of applying $S$ to $A$ by imposing sparsity structure on $S$. The canonical example is the sparse Johnson–Lindenstrauss transform (SJLT), where each column of $S$ contains a small, fixed number of nonzeros, $s$ (usually chosen to be Rademacher variables with scale $1/\sqrt{s}$) (Dasgupta et al., 2010; Kane & Nelson, 2014). OSNAP further refines the construction and analysis to obtain sparse OSEs that enable input-sparsity-time algorithms (Nelson & Nguyen, 2013; Clarkson & Woodruff, 2017). SJLTs and OSNAP are also OSEs for sufficiently large sparsity $s = \mathcal{O}(\varepsilon^{-1} \log(n/\delta))$ and sketch dimension $k = \mathcal{O}(\varepsilon^{-2} n \log(n/\delta))$.

### 3.4. Localized Sketching

Localized sketching (Srinivasa et al., 2020) further imposes structure on sketches by studying block-diagonal JLTs. This is an alternative route to sparsity for reducing application cost. Localized sketches also have OSE guarantees, but the required sketch dimension now depends on a *block-coherence* parameter of the input subspace.

**Definition 3.2** (Block coherence (Srinivasa et al., 2020))**.** Let $U \in \mathbb{R}^{d \times r}$ have orthonormal columns. Partition the rows into $M$ contiguous blocks of size $B_c = d/M$, and write $U = [U^{(1)}; \ldots; U^{(M)}]$ with $U^{(h)} \in \mathbb{R}^{B_c \times r}$. The block coherence of $U$ is

$$\mu_{\mathrm{blk}}(U) := M \max_{h \in [M]} \|U^{(h)}\|_2^2. \qquad (2)$$

## 3.5. GPUs and the Memory Hierarchy

We provide a brief overview of the architectural details of GPUs that are relevant to this paper; further details can be found in the CUDA programming guide (Nvidia, 2011). A GPU consists of thousands of CUDA cores that can independently execute threads of computation in parallel. Along with a large number of compute units, the GPU also has a hierarchy of memory that the cores can access. The hierarchy stems from a fundamental trade-off between memory size, latency, and bandwidth. The fastest memory is the register memory, which is local to each thread and is used to store intermediate results. The next level of memory is the shared memory, which is shared between a local group of threads. The global memory is the largest and slowest memory, but it is accessible by all threads. However, if multiple threads attempt to read or write to the same location in global memory simultaneously, it can lead to contention and we need to use *atomic* operations to maintain correctness. These operations lead to serialization of memory accesses, which can significantly degrade performance.

## 3.6. GPU Bottlenecks for Sparse Sketching

Sparse sketching on GPUs is bottlenecked by memory bandwidth, since compute is relatively cheap compared to moving data through the memory hierarchy. Further, existing sparse sketching kernels such as those of (Hu et al., 2025; Higgins et al., 2025) rely on global atomic operations to handle collisions in the sketching matrix. Specifically, they implement a **scatter-add** approach, where each thread processes an input row (or a tile of rows) and writes $s$ signed contributions into output rows. Correctness requires that these writes be atomic because hashed updates can collide. Since the sparsity patterns of SJLT/CountSketch are random and irregular, it is difficult to leverage the memory hierarchy effectively. Specifically, shared memory reuse cannot be done reliably.

An alternative approach is to explicitly form the sparse sketching matrix $S$ in memory and perform a sparse-dense matrix multiplication (SpMM) to compute $Y = SA$. This avoids custom kernels and leverages existing highly-optimized SpMM libraries like cuSPARSE (Naumov et al., 2010). However, this approach requires materializing $S$ in a sparse format, which incurs overheads in both memory and computation. Additionally, it cannot exploit the specific structure of the entries of $S$, like the Rademacher signs in SJLT/CountSketch, which can be generated on-the-fly.

The common architectural point is that global memory traffic is expensive and synchronization across thread blocks is slow. By contrast, shared memory supports fast, block-local communication, and shared-memory atomics are often far cheaper than global atomics.

## 4. BLOCKPERM-SJLT: A Block-Permuted Sparse JL Transform

This section describes the sketch distribution used throughout the paper. The GPU implementation in Section 5 exploits the particular block structure of this distribution.

**Block model.** Let $M$ be a positive integer. Partition the $d$ input coordinates into $M$ contiguous blocks of size $B_c := d/M$, and partition the $k$ output coordinates into $M$ blocks of size $B_r := k/M$. For simplicity, we assume that $d$ and $k$ are divisible by $M$. We deal with general cases in practice by padding.

We index output blocks by $g \in [M]$ and input blocks by $h \in [M]$.

The sketch matrix $S \in \mathbb{R}^{k \times d}$ is therefore composed of $M \times M$ blocks, each of size $B_r \times B_c$. Note that under this partition, the block sketch matrix $S$ is always square at the block level.

**Block-level wiring as a union of permutations.** This key structural element enables describing BLOCKPERM-SJLT as a union of $\kappa$ permutations at the block level.

Specifically, sample $\kappa$ **edge-disjoint** permutations $\{\pi_\ell\}_{\ell=1}^\kappa$ of $[M]$. Note, importantly, that the permutations are edge-disjoint, meaning that no two permutations map the same output block to the same input block. Precisely, for all $g \in [M]$ and $\ell \neq \ell'$, we have $\pi_\ell(g) \neq \pi_{\ell'}(g)$. Equivalently, the permutations are pairwise derangements. This is necessary to ensure that each output block mixes information from $\kappa$ distinct input blocks. For each output block $g$, define the neighborhood of input blocks

$$\mathcal{N}(g) := \{\pi_\ell(g)\}_{\ell=1}^\kappa \subseteq [M].$$

We then define the block sparsity pattern of $S$ by connecting output block $g$ to input blocks in $\mathcal{N}(g)$ only. The edge-disjoint restriction implies that the induced block bipartite graph is $\kappa$-regular on both sides: each output block touches exactly $\kappa$ input blocks, and each input block participates in exactly $\kappa$ output blocks.

**Intra-block SJLT mixing.** For every nonzero block $(g, h)$ with $h \in \mathcal{N}(g)$, we draw an independent sparse JL matrix $\Phi_{g,h} \in \mathbb{R}^{B_r \times B_c}$ with exactly $s$ nonzeros per column, with entries $\pm 1/\sqrt{s}$ at uniformly random row positions. Then, we define the full sketch matrix $S$ using these blocks as

$$S_{g,h} = \begin{cases} \frac{1}{\sqrt{\kappa}} \Phi_{g,h} & \text{if } h \in \mathcal{N}(g), \\ 0 & \text{otherwise.} \end{cases}$$

Thus each input coordinate has exactly $\kappa s$ nonzeros in its column of $S$, each with magnitude $1/\sqrt{\kappa s}$. Note that this

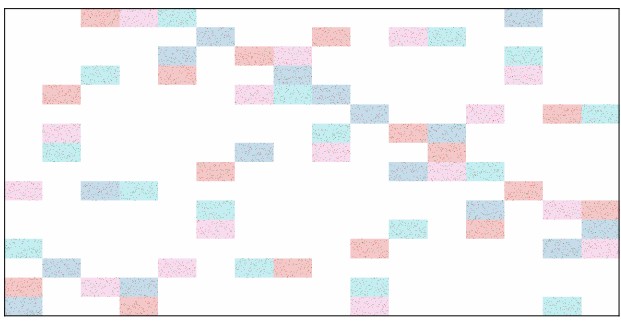

*Figure 2.* **Example** $S \sim$ **BLOCKPERM-SJLT**. Sparsity is a union of edge-disjoint permutations (degree-$\kappa$ regular at the block level). Each block is a sparse JL matrix with $s$ nonzeros per column. Here, $M = 16$, with block size $B_r = 64$, $B_c = 128$. The block degree is $\kappa = 4$ and the intra-block sparsity is $s = 2$. The resulting sketch therefore has $\kappa s = 8$ nonzeros per column, input dimension $d = MB_c = 2048$ and output dimension $k = MB_r = 1024$. Each of the 4 permutations is colored differently for visualization.

sketch is a structured subclass of SJLT and a generalization of localized JL with sparse blocks (Srinivasa et al., 2020). Specifically, when $\kappa = 1$, this reduces to a localized, block-diagonal SJLT.

We use the bi-regularity of the block sparsity both in the kernel design and the theoretical analysis. Increasing $\kappa$ expands each output neighborhood while reducing block locality, providing a tunable and quantitative tradeoff between mixing and memory efficiency. We will concretely see this tradeoff from the theory side in Section 6 and the systems side in Section 7. An example of BLOCKPERM-SJLT is illustrated in Figure 2.

## 5. FLASHSKETCH: CUDA Kernel

This section describes FLASHSKETCH, the CUDA implementation of BLOCKPERM-SJLT. FLASHSKETCH achieves high performance through two main optimizations:

1. It leverages the block sparsity structure of BLOCKPERM-SJLT to completely eliminate global atomic operations, while still maintaining a scatter-add structure local to each thread-block to maximize occupancy and memory throughput.
2. It generates all sketch randomness on the fly within each thread-block, reducing bandwidth demand as well as global memory footprint.

We elaborate on these two optimizations below.

### 5.1. Eliminating Global Atomics

Let $Y = SA \in \mathbb{R}^{k \times n}$ with $k = MB_r$, and $S \sim$ BLOCKPERM-SJLT. To compute $Y$, we launch a 2D grid of thread blocks, indexed by $(g, j)$ where $g \in [M]$ selects an output block-row and $j$ selects a tile

of $T_n$ columns. Thread block $(g, j)$ computes the tile $Y[gB_r : (g+1)B_r, \ jT_n : (j+1)T_n] \in \mathbb{R}^{B_r \times T_n}$ (we follow *NumPy* indexing notation).

For a fixed output block $g$, the kernel generates its random block neighborhood $\mathcal{N}(g) = \{\pi_\ell(g)\}_{\ell=1}^{\kappa}$ on the fly. For each input block $h \in \mathcal{N}(g)$, we stream the corresponding rows of $A$ through shared memory in tiles of height $T_k$ and width $T_n$. We allocate two shared arrays: $sA \in \mathbb{R}^{T_k \times T_n}$ for the input tile and $sY \in \mathbb{R}^{B_r \times T_n}$ for the output tile. Each loaded element of $sA$ participates in $s$ signed updates into $sY$, where the update destinations and signs are generated from a per-element hash. Since $sY$ is private to the thread-block, these updates use shared-memory atomics rather than global atomics, which are much faster and have lower contention costs. After processing all $\kappa$ neighbor blocks and all row tiles within each block, the thread-block cooperatively writes $sY$ to global memory once.

### 5.2. On-The-Fly Randomness Generation

The kernel never materializes $S$. Instead, it generates (i) the block wiring $\pi_\ell(g)$ and (ii) the intra-block SJLT hashes on the fly. For the wiring, we use a structured permutation family so that $\pi_\ell(g)$ can be computed using simple integer arithmetic. Specific details are in Section D. For intra-block hashing, each input row index within the current input block is combined with $(g, h)$ and a seed to produce $s$ target rows in $[B_r]$ and $s$ independent signs. This design eliminates the bandwidth and cache pressure associated with reading a sparse index structure, and it keeps the inner loop branch-free. For reference, we include pseudocode for one thread-block of the FLASHSKETCH kernel in Algorithm 1.

---

**Algorithm 1** FLASHSKETCH (one thread-block)

---

1: **Input:** $A \in \mathbb{R}^{d \times n}$, block id $g$, column tile $j$, params $(M, B_r, B_c, \kappa, s, T_k, T_n)$.
2: Compute random neighborhood $\mathcal{N}(g) = \{\pi_\ell(g)\}_{\ell=1}^{\kappa}$.
3: Initialize shared output tile $sY \leftarrow 0$.
4: **for** each $h \in \mathcal{N}(g)$ **do**
5:    **for** $u_0 = 0, T_k, 2T_k, \ldots, B_c - T_k$ **do**
6:       Load $A[hB_c + u_0 : hB_c + u_0 + T_k, \ jT_n : (j+1)T_n]$ into shared tile $sA$.
7:       **for** each element $(u, t)$ of $sA$ in parallel **do**
8:          **for** $i = 1$ to $s$ **do**
9:             Hash $(g, h, u_0 + u, i)$ to unique destination $r_i \in [B_r]$ and sign $\sigma_i$.
10:             `atomicAdd(`$sY[r_i, t], \ \sigma_i \cdot sA[u, t]$`)`.
11:          **end for**
12:       **end for**
13:    **end for**
14: **end for**
15: Scale $sY$ by $\frac{1}{\sqrt{\kappa s}}$.
16: Write $sY$ to $Y[gB_r : (g+1)B_r, \ jT_n : (j+1)T_n]$.

---

## 5.3. Comparison to Prior GPU SJLT Kernels

The specialized GPU SJLT GraSS sketch kernel (Hu et al., 2025) and the GPU CountSketch implementations in (Higgins et al., 2025) follow a *global* scatter-add pattern: each input element contributes to $s$ output rows and resolves collisions with *global* atomic adds. For dense $A$, this induces $\Theta(s\,d\,n)$ global atomic updates and forces contention whenever many coordinates hash to the same output rows. This can severely degrade memory performance, especially in the $d \gg k$ regime.

By construction, all $s$ updates per input element in FLASHS-KETCH are performed in shared memory private to each thread-block (using shared memory atomics), eliminating global atomic contention. As a result, FLASHSKETCH performs *no global atomics* and only $\Theta(k\,n)$ global writes, which are naturally coalesced. Further, the kernel in (Hu et al., 2025) requires forming the sparse sketch as an input to the kernel, which incurs additional memory overhead and bandwidth.

We provide a detailed discussion of tuning FLASHSKETCH and handling low-occupancy cases in Section B. Additionally, we provide a fast but fragile block-row sampling alternative in Section C.

# 6. Theory: Permutation-Coupled Localization

We build our theoretical understanding of BLOCKPERM-SJLT on the framework of localized sketching developed in (Srinivasa et al., 2020). *Localized sketching* studies block-diagonal (or block-local) sketches whose guarantees depend on how the target subspace distributes across the chosen blocks. Our sketch BLOCKPERM-SJLT follows a very similar locality principle with the key difference that a single output block mixes information from multiple input blocks via the union-of-permutations wiring. We formalize this via a new *neighborhood coherence* quantity that is a natural generalization of the block coherence in (Srinivasa et al., 2020).

## 6.1. Coherence for Permutation-Coupled Localization

We begin by defining our generalization of block-coherence Theorem 3.2, the *neighborhood coherence*.

**Definition 6.1** (Neighborhood coherence for permutation wiring). Fix $\kappa$ **edge-disjoint** permutations $\pi_1, \ldots, \pi_\kappa$ on $[M]$ and define the neighborhood of output block $g \in [M]$ by

$$\mathcal{N}(g) \; := \; \{\pi_\ell(g)\}_{\ell=1}^{\kappa}. \quad (3)$$

Let $U_{\mathcal{N}(g)} \in \mathbb{R}^{(\kappa B_c) \times r}$ denote the matrix obtained by stacking the blocks $U^{(h)}$ for $h \in \mathcal{N}(g)$. The neighborhood coherence of $U$ under $\pi$ is

$$\mu_{\mathrm{nbr}}(U; \pi) \; := \; \frac{M}{\kappa} \max_{g \in [M]} \|U_{\mathcal{N}(g)}\|_2^2. \quad (4)$$

Recall that the edge-disjoint condition ensures that each output block $g$ connects to $\kappa$ distinct input blocks.

## 6.2. Oblivious Subspace Embedding Guarantee

We now state our main theoretical result Theorem 6.2, which is an OSE guarantee for BLOCKPERM-SJLT controlled by neighborhood coherence. The flavor is very similar to the localized SJLT result of (Srinivasa et al., 2020), but with block coherence replaced by neighborhood coherence.

**Theorem 6.2** (OSE for BLOCKPERM-SJLT). *Fix* $U \in \mathbb{R}^{d \times r}$ *with orthonormal columns. Let* $S \sim$ BLOCKPERM-SJLT *with parameters* $(M, B_r, \kappa, s)$ *and embedding dimension* $k = MB_r$. *Let* $t := r + \log \frac{1}{\delta}$. *There exist absolute constants* $C, c > 0$ *such that, if we have*

$$k \; \geq \; C \, \frac{\mu_{\mathrm{nbr}}(U; \pi)}{\varepsilon^2} \, t \qquad and \qquad \kappa s \; \geq \; C \, \frac{1}{\varepsilon} \, t, \quad (5)$$

*then with probability at least* $1 - \delta$ *over* $S$,

$$\left\| U^\top S^\top S U - I_r \right\|_2 \leq \varepsilon. \quad (6)$$

The proof follows the localized sketching blueprint closely. For a fixed vector $w$, we decompose $\|SUw\|_2^2$ into a sum of independent block contributions and combine a fixed-vector tail bound with a net argument. The union-of-permutations wiring enters through a simple energy identity that ensures the local neighborhoods collectively cover the input without bias. A detailed proof appears in Section A. Note that for $\kappa = 1$, Theorem 6.2 recovers the localized SJLT regime controlled by $\mu_{\mathrm{blk}}(U)$ from (Srinivasa et al., 2020).

## 6.3. Controlling Neighborhood Coherence with Randomized Permutations

We now discuss how neighborhood coherence interpolates between block coherence and fully mixed coherence, and how random permutations can improve it in practice.

**Worst-case comparison.** First, a simple worst case comparison between neighborhood coherence and block coherence shows that permutations can reduce coherence by up to a factor of $\kappa$, but in general we cannot do better than block coherence. For any fixed set of edge-disjoint permutations $\pi$,

$$\frac{1}{\kappa} \mu_{\mathrm{blk}}(U) \; \leq \; \mu_{\mathrm{nbr}}(U; \pi) \; \leq \; \mu_{\mathrm{blk}}(U). \quad (7)$$

The upper bound is a triangle inequality and the lower bound follows because every block appears in at least one neighborhood. A detailed derivation appears in Section A.

**Randomized Permutations.** Now, if we model the permutations $\pi_1, \ldots, \pi_\kappa$ as independent and uniformly random, then we can show a stronger upper bound that improves with $\kappa$ and pushes $\mu_{\text{nbr}}(U; \pi)$ toward one. A precise statement appears as Theorem A.11 in Section A. Informally, we have the following smoothing bound. Let $L := \log\left(\frac{Mr}{\delta}\right)$. With probability at least $1 - \delta$ over the permutations, we have

$$\mu_{\text{nbr}}(U; \pi) \ \leq \ 1 + C \left( \sqrt{\frac{\mu_{\text{blk}}(U) \, L}{\kappa}} + \frac{\mu_{\text{blk}}(U) \, L}{\kappa} \right) \ (8)$$

for an absolute constant $C > 0$.

This rigorously justifies the benefit of using multiple permutations to improve mixing. When $\kappa$ is large enough, the neighborhood coherence approaches one, which is the optimal coherence for any $U$, improving the OSE guarantee in Theorem 6.2.

**Remark 6.3** (Independent permutations in practice)**.** Our analysis models $\pi_1, \ldots, \pi_\kappa$ as independent uniformly random permutations. In FLASHSKETCH, we generate $\kappa$ distinct permutations using a lightweight family for efficiency, see Section D. When $\kappa \ll M$, sampling distinct permutations is close to sampling independently, and collisions are rare. This justifies our theoretical model.

# 7. Experiments

We empirically evaluate FLASHSKETCH at two levels. First, we treat sketching as a primitive and measure the runtime and distortion of computing $SA$ for dense $A$ across a broad range of shapes. Second, we integrate FLASHSKETCH into end-to-end workloads where sketching sits on the critical path. As part of this, we consider a suite of standard RandNLA tasks as well as GraSS, an end-to-end ML application for data attribution (Hu et al., 2025) in which sketching is a core component. We time using CUDA events and report the mean over 10 iterations (after warm-up iterations). We restrict our evaluation to FP32 arithmetic for both sketching and downstream tasks for consistency, since all our baselines support FP32. Our main kernel idea extends naturally to other precisions as well, but we leave concrete exploration of precision-specific optimizations to future work.

## 7.1. Baselines

We compare against dense and sparse GPU baselines.

1. A dense JL transform with a standard Gaussian projection using cuBLAS.
2. A sparse JL transform using cuSPARSE SpMM.
3. A sparse JL transform using the GraSS SJLT kernel (Hu et al., 2025).
4. A subsampled randomized Hadamard transform

(SRHT) using the Fast Hadamard Transform (FHT) kernel from Dao-AILab (Dao-AILab, 2024).

## 7.2. Sketching

For the primitive sketching evaluation, we use the relative error in the computed Gram matrix $\|A^\top A - (SA)^\top (SA)\|_F / \|A^\top A\|_F$ as our quality metric. The preview in Figure 1 illustrates the quality–speed tradeoff on a representative input from GPT2-medium weights. More extensive ablations and Pareto frontiers across different shapes and inputs appear in Section F.

## 7.3. RandNLA Tasks

We benchmark FLASHSKETCH on three standard end-to-end RandNLA tasks:

1. Sketch-and-solve least squares regression.
2. Sketch-and-ridge regression.
3. Oblivious subspace embedding (OSE).

We run each task on a variety of input shapes and types. We use the following datasets for inputs:

1. A synthetic Gaussian matrix.
2. A synthetic low-rank + noise matrix.
3. A sparse submatrix from the SuiteSparse collection (`spal_004`, with nonzero density $\approx$ 1.4%) (Kolodziej et al., 2019).
4. A concatenated subset of weights from large language models (LLMs) (`GPT2-medium` (Radford et al., 2019) and `Qwen2-1.5B` (Yang et al., 2024)).

Figure 3 shows a representative quality–speed Pareto frontier for sketch-and-ridge regression on Qwen2-1.5B stacked weights. We present a table of aggregated speedups over

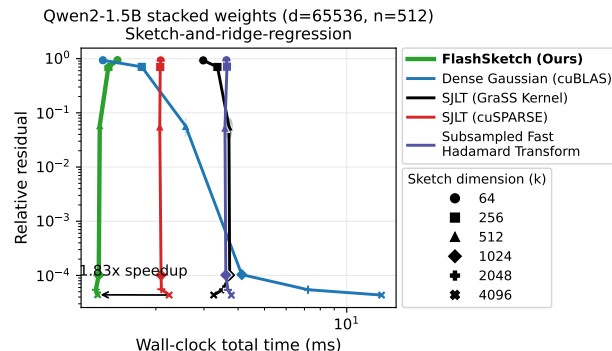

*Figure 3.* **Sketch-and-ridge regression on Qwen2-1.5B stacked weights.** We compare end-to-end runtime on an NVIDIA RTX 4090 GPU (x-axis) and quality measured using sketch-and-ridge regression residual (y-axis). **Lower is better for both axes.**

baselines across all RandNLA tasks, input shapes, and datasets in Table 1 (run on a RTX 4090). Correspond-

*Table 1.* Geomean speedups of FLASHSKETCH vs baselines aggregated over shapes, datasets and configs. **Global geomean vs next best baseline: 1.73x.**

| Task | SJLT (cuSPARSE) | SJLT (GraSS Kernel) | Subsampled FHT | Dense Gaussian (cuBLAS) |
|---|---|---|---|---|
| Gram matrix approximation | 2.67 | 4.17 | 16.22 | 7.64 |
| OSE | 2.69 | 4.16 | 16.20 | 7.63 |
| Sketch-and-ridge regression | 1.42 | 1.54 | 3.94 | 3.10 |
| Sketch+ Solve | 1.37 | 1.41 | 3.33 | 2.34 |

ing detailed plots and ablations appear in Section F.

### 7.4. End-to-End ML: GraSS for Data Attribution

We integrate FLASHSKETCH into GraSS (Hu et al., 2025), a scalable data attribution pipeline built around compressing and storing per-example training gradients. At a high level, GraSS constructs a *feature cache* for the training set by compressing per-example gradients using a sparsification followed by a random projection (or sketch). In the attribution phase, GraSS computes an analogous representation for a test/query example and produces an attribution vector using the cached features and a small downstream solve. In GraSS, the random projection step is a key bottleneck: it is invoked for every training example during cache construction (and for every query at inference time), so kernel efficiency directly impacts end-to-end wall time.

We replace GraSS's sparse projection (its SJLT CUDA kernel) with FLASHSKETCH while keeping all other pipeline components fixed. We adapt the code from their open-source repository directly (TRAIS-Lab, 2025). We focus our evaluation on the time spent in the projection/sketching step during feature cache construction.

**Attribution quality via the linear datamodeling score (LDS).** We evaluate attribution quality using the *linear datamodeling score* (LDS) introduced in TRAK (Park et al., 2023), which GraSS also adopts. LDS is a counterfactual metric: it measures how well an attribution method predicts how the model's output on a fixed example $z$ would change if we retrained on different subsets of the training set. **Note that higher LDS is better.**

Figure 4 shows the quality–speed Pareto frontier for end-to-end GraSS on an MLP trained on MNIST (LeCun et al., 2002). We refer the reader to Section E.1 and (Hu et al., 2025) for full details on the GraSS pipeline and the LDS metric.

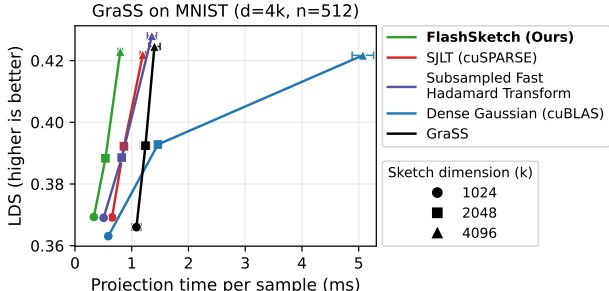

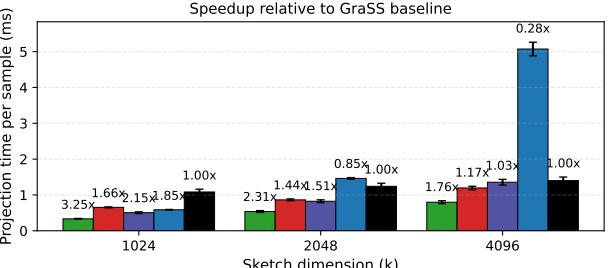

*Figure 4.* **End-to-end GraSS evaluation on MNIST.** We compare sketching methods by sketch runtime per sample on an NVIDIA RTX A6000 GPU (x-axis), and quality measured using the LDS metric (y-axis). **Lower is better for time (x-axis), higher is better for LDS (y-axis).** FLASHSKETCH pushes the Pareto frontier and achieves up to $\sim 3.2\times$ speedup over the GraSS baseline while maintaining attribution quality (top).

## 8. Limitations and Future Directions

We discuss limitations and future directions for FLASHSKETCH from theoretical and practical perspectives.

**What regimes does FLASHSKETCH help and fail in?** FLASHSKETCH is most advantageous when (i) $A$ is dense, and (ii) $k \ll d$ but $k$ is large enough to saturate the GPU. In the low $k$ regime, low occupancy limits speedups. We resolve this to an extent using a split-$B_c$ approach discussed in Section B.1. Additionally, while $\kappa$ can be tuned to trade off mixing and memory efficiency, very large $\kappa$ increases input reads, which can limit speedups.

**Statistics–hardware tradeoffs.** While a primary contribution of FLASHSKETCH is the tunable parameter $\kappa$ that directly trades off sketch quality and GPU efficiency, the optimal choice of $\kappa$ depends on the input data statistics and the hardware characteristics. In this work, we empirically choose the optimal $\kappa$ on the Pareto frontier. An interesting future direction is to more deeply explore this tradeoff and develop a principled, perhaps adaptive approach to select $\kappa$.

**Theoretical limitations.** Our theoretical analysis in Section 6 focuses on OSE guarantees for BLOCKPERM-SJLT using independent uniform permutations. In practice, we sample edge-disjoint permutations which introduce depen-

dence between neighborhoods. While this is mild in the $\kappa \ll M$ regime we focus on, it is an interesting open question to analyze BLOCKPERM-SJLT with such dependent permutations.

**Beyond OSE.** We focus on OSE-style guarantees because they capture the geometric fidelity needed in many RandNLA algorithms. An appealing direction is to derive tighter guarantees tailored to the Gram-matrix metrics we evaluate, and to connect those directly to end-to-end downstream performance in learning pipelines.

## Acknowledgments

This work was supported in part by the National Science Foundation (NSF) CAREER Award under Grant CCF-2236829, in part by the National Institutes of Health under Grant 1R01AG08950901A1, in part by the Office of Naval Research under Grant N00014-24-1-2164, and in part by the Defense Advanced Research Projects Agency under Grant HR00112490441.

## Impact Statement

This paper presents work whose goal is to advance the field of Machine Learning. There are many potential societal consequences of our work, none of which we feel must be specifically highlighted here.

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

# A. Additional Theory and Proofs

This appendix contains the proofs for the theory in Section 6. We focus on two results. First, we prove the OSE guarantee for BLOCKPERM-SJLT stated in Theorem 6.2. Second, we show that a union of random permutations reduces neighborhood coherence, which is the smoothing phenomenon discussed in Section 6.1.

The overall proof strategy follows the localized sketching framework of (Srinivasa et al., 2020). The main difference is the permutation wiring. It defines $\kappa$-block neighborhoods that retain locality while improving mixing. In the OSE analysis we condition on the wiring $\pi$ and analyze the randomness in the SJLT blocks. The final subsection then studies the randomness in $\pi$ itself.

## A.1. Construction and Notation

Let $d = MB_c$ and $k = MB_r$. For $x \in \mathbb{R}^d$, write $x = [x^{(1)}; \ldots; x^{(M)}]$ with blocks $x^{(h)} \in \mathbb{R}^{B_c}$. Given $\kappa$ edge-disjoint permutations $\{\pi_\ell\}_{\ell=1}^{\kappa}$ on $[M]$, define for each $g \in [M]$

$$\mathcal{N}(g) = \{\pi_\ell(g)\}_{\ell=1}^{\kappa}.$$

Let $x_{\mathcal{N}(g)} \in \mathbb{R}^{\kappa B_c}$ denote the concatenation of the blocks $\{x^{(h)} : h \in \mathcal{N}(g)\}$ in the order $(\pi_1(g), \ldots, \pi_\kappa(g))$.

For each $(g, h)$ with $h \in \mathcal{N}(g)$, draw an independent row-partitioned SJLT $\Phi_{g,h} \in \mathbb{R}^{B_r \times B_c}$ with exactly $s$ nonzeros per column and entries $\pm 1/\sqrt{s}$. Define the concatenated local sketch

$$\Phi_g := \begin{bmatrix} \Phi_{g,\pi_1(g)} & \cdots & \Phi_{g,\pi_\kappa(g)} \end{bmatrix}, \tag{9}$$
$$\Phi_g \in \mathbb{R}^{B_r \times (\kappa B_c)}.$$

Finally, define the full sketch $S \in \mathbb{R}^{k \times d}$ blockwise by

$$S_{g,h} = \begin{cases} \frac{1}{\sqrt{\kappa}} \Phi_{g,h} & \text{if } h \in \mathcal{N}(g), \\ 0 & \text{otherwise.} \end{cases}$$

Then the $g$-th output block of $Sx$ is

$$(Sx)^{(g)} = \frac{1}{\sqrt{\kappa}} \Phi_g \, x_{\mathcal{N}(g)} \in \mathbb{R}^{B_r}. \tag{10}$$

This is identical to the construction of BLOCKPERM-SJLT in Section 4, we repeat it here for clarity.

## A.2. Energy Identity from Permutation Wiring

The union of edge-disjoint permutations structure implies that each input block appears in exactly $\kappa$ neighborhoods.

**Lemma A.1** (Neighborhood energy identity). *For any $x \in \mathbb{R}^d$,*

$$\sum_{g=1}^{M} \|x_{\mathcal{N}(g)}\|_2^2 = \kappa \|x\|_2^2. \tag{11}$$

*Moreover, for any $U \in \mathbb{R}^{d \times r}$,*

$$\sum_{g=1}^{M} U_{\mathcal{N}(g)}^{\top} U_{\mathcal{N}(g)} = \kappa U^{\top} U. \tag{12}$$

*Proof.* For the vector identity, expand

$$\begin{aligned} \sum_{g=1}^{M} \|x_{\mathcal{N}(g)}\|_2^2 &= \sum_{g=1}^{M} \sum_{\ell=1}^{\kappa} \|x^{(\pi_\ell(g))}\|_2^2 \\ &= \sum_{\ell=1}^{\kappa} \sum_{g=1}^{M} \|x^{(\pi_\ell(g))}\|_2^2. \end{aligned} \tag{13}$$

Each $\pi_\ell$ is a bijection, so $\sum_g \|x^{(\pi_\ell(g))}\|_2^2 = \sum_h \|x^{(h)}\|_2^2 = \|x\|_2^2$, and the claim follows.

The matrix identity follows by applying the same counting argument entrywise:

$$\begin{aligned} \sum_{g=1}^{M} U_{\mathcal{N}(g)}^{\top} U_{\mathcal{N}(g)} &= \sum_{g=1}^{M} \sum_{\ell=1}^{\kappa} U^{(\pi_\ell(g))\top} U^{(\pi_\ell(g))} \\ &= \sum_{\ell=1}^{\kappa} \sum_{h=1}^{M} U^{(h)\top} U^{(h)} = \kappa U^{\top} U. \end{aligned} \tag{14}$$

$\square$

## A.3. A Fixed-Vector Bound for Row-Partitioned SJLT

We use the following standard fixed-vector tail bound for SJLT/OSNAP-style hashing matrices.

**Lemma A.2** (Fixed-vector norm preservation for row-partitioned SJLT). *Let $\Phi \in \mathbb{R}^{m \times D}$ be a row-partitioned SJLT with exactly $s$ nonzeros per column and entries $\pm 1/\sqrt{s}$ with independent signs and row positions. Then for any fixed $v \in \mathbb{R}^D$ and any $u \in (0, 1)$,*

$$\begin{aligned} \Pr\Big[ \big| \|\Phi v\|_2^2 - \|v\|_2^2 \big| > u \|v\|_2^2 \Big] \\ \leq 4 \exp\big(-c \min\{u^2 m, \, us\}\big). \end{aligned} \tag{15}$$

*for an absolute constant $c > 0$.*

We will use Theorem A.2 as a black box throughout the appendix. For completeness (and to avoid any ambiguity about constants and notation), we give a short derivation of the stated tail form from the distributional JL guarantee for the *block construction* (i.e., row-partitioned hashing) analyzed by Kane & Nelson (2014). The proof is a simple "parameter inversion": we start from a statement of the form "if $m$ and $s$ scale like $\varepsilon^{-2} \log(1/\delta)$ and $\varepsilon^{-1} \log(1/\delta)$ then failure probability is at most $\delta$", and then solve for $\delta$ as a function of $m, s, u$.

*Derivation of Theorem A.2 from Kane & Nelson (2014).*
Fix $v \neq 0$ and set $x = v/\|v\|_2$ so that $\|x\|_2 = 1$. Let $S \in \mathbb{R}^{m \times D}$ be the row-partitioned SJLT ("block construction") with $s$ nonzeros per column and entries $\pm 1/\sqrt{s}$. Following Kane & Nelson (2014), define the distortion random variable

$$Z := \|Sx\|_2^2 - 1. \tag{16}$$

In the notation of Kane & Nelson (2014), $m$ corresponds to their embedding dimension $k$, and this $Z$ is exactly the random variable defined in their Eq. (3).

**Step 1: start from a $(\varepsilon, \delta)$-style JL guarantee.** Kane & Nelson (2014, Theorem 13) shows (for the block construction) that there exist absolute constants $C_1, C_2 > 0$ such that for any $\varepsilon \in (0,1)$ and $\delta \in (0, 1/2)$, if

$$\begin{aligned} m &\geq C_1 \, \varepsilon^{-2} \log(1/\delta), \\ s &\geq C_2 \, \varepsilon^{-1} \log(1/\delta), \end{aligned} \tag{17}$$

then

$$\Pr\big[|Z| > 2\varepsilon - \varepsilon^2\big] \leq \delta. \tag{18}$$

**Step 2: convert to a tail bound in terms of $u$.** Let $u \in (0,1)$ and set $\varepsilon := u/2$. Since $2\varepsilon - \varepsilon^2 = u - u^2/4 < u$, we have the set inclusion

$$\{|Z| > u\} \subseteq \{|Z| > 2\varepsilon - \varepsilon^2\}. \tag{19}$$

Therefore, whenever the conditions on $m$ and $s$ above hold (with $\varepsilon = u/2$),

$$\Pr\big[|Z| > u\big] \leq \Pr\big[|Z| > 2\varepsilon - \varepsilon^2\big] \leq \delta. \tag{20}$$

**Step 3: solve for $\delta$ as a function of $m, s, u$.** With $\varepsilon = u/2$, the sufficient conditions become

$$\log(1/\delta) \leq \frac{u^2 m}{4 C_1} \quad \text{and} \quad \log(1/\delta) \leq \frac{us}{2 C_2}. \tag{21}$$

Equivalently, for any $c \leq \min\{(4C_1)^{-1}, (2C_2)^{-1}\}$, choosing

$$\delta = \exp\big(-c \min\{u^2 m, \, us\}\big) \tag{22}$$

ensures the hypotheses of Kane & Nelson (2014, Theorem 13) (with $\varepsilon = u/2$) are satisfied. Substituting this choice of $\delta$ into the preceding display yields

$$\Pr\big[|Z| > u\big] \leq \exp\big(-c \min\{u^2 m, \, us\}\big). \tag{23}$$

Finally, rescaling from $x = v/\|v\|_2$ back to $v$ gives

$$\begin{aligned} &\Pr\Big[\big|\|Sv\|_2^2 - \|v\|_2^2\big| > u\,\|v\|_2^2\Big] \\ &\leq \exp\big(-c \min\{u^2 m, \, us\}\big). \end{aligned} \tag{24}$$

which is the desired tail form up to adjusting constants (we state a slightly looser version with a prefactor 4 for convenience). $\qquad\square$

In particular, we will repeatedly invoke Theorem A.2 to control the centered error $\|\Phi v\|_2^2 - \|v\|_2^2$ as a sub-exponential random variable (cf. the proof of Theorem A.5).

## A.4. Coherence Metrics

We use two coherence parameters to track how vectors and subspaces align with the block partition. Block coherence measures concentration inside a single contiguous block. Neighborhood coherence measures concentration inside a union of $\kappa$ blocks selected by the wiring permutations. We use the same notation $\mu_{\mathrm{blk}}(\cdot)$ and $\mu_{\mathrm{nbr}}(\cdot; \pi)$ for vectors and matrices.

**Definition A.3** (Vector block and neighborhood coherence). Let $x \in \mathbb{R}^d$ be nonzero and partition it into contiguous blocks $x = [x^{(1)}; \ldots; x^{(M)}]$. Define

$$\mu_{\mathrm{blk}}(x) := \frac{M}{\|x\|_2^2} \max_{h \in [M]} \|x^{(h)}\|_2^2, \tag{25}$$

$$\mu_{\mathrm{nbr}}(x; \pi) := \frac{M}{\kappa \|x\|_2^2} \max_{g \in [M]} \big\|x_{\mathcal{N}(g)}\big\|_2^2. \tag{26}$$

**Definition A.4** (Matrix block and neighborhood coherence (Srinivasa et al., 2020)). Let $U \in \mathbb{R}^{d \times r}$ have orthonormal columns and write $U = [U^{(1)}; \ldots; U^{(M)}]$ for its induced row-block partition. Define

$$\mu_{\mathrm{blk}}(U) := M \max_{h \in [M]} \|U^{(h)}\|_2^2, \tag{27}$$

$$\mu_{\mathrm{nbr}}(U; \pi) := \frac{M}{\kappa} \max_{g \in [M]} \big\|U_{\mathcal{N}(g)}\big\|_2^2. \tag{28}$$

These agree with the definitions (2)–(4) in the main text.

## A.5. Fixed-Vector Guarantee for BLOCKPERM-SJLT

**Proposition A.5** (Fixed-vector JL for BLOCKPERM-SJLT). *Let $S \sim$ BLOCKPERM-SJLT with parameters $(M, B_r, \kappa, s)$ and independent $\{\Phi_g\}_{g=1}^M$. Then for any fixed $x \in \mathbb{R}^d$ and any $u \in (0,1)$,*

$$\begin{aligned} &\Pr\Big[\big|\|Sx\|_2^2 - \|x\|_2^2\big| > u\,\|x\|_2^2\Big] \\ &\leq 2\exp\Big(-c \min\Big\{\frac{u^2 k}{\mu_{\mathrm{nbr}}(x; \pi)}, \, u\,\kappa s\Big\}\Big). \end{aligned} \tag{29}$$

*where $k = MB_r$ and $c > 0$ is an absolute constant.*

*Proof.* Write the output norm as a sum over blocks using (10):

$$\|Sx\|_2^2 = \sum_{g=1}^M \|(Sx)^{(g)}\|_2^2 = \frac{1}{\kappa} \sum_{g=1}^M \|\Phi_g x_{\mathcal{N}(g)}\|_2^2.$$

Define random variables

$$Z_g = \frac{1}{\kappa}\Big(\|\Phi_g x_{\mathcal{N}(g)}\|_2^2 - \|x_{\mathcal{N}(g)}\|_2^2\Big).$$

Then $\mathbb{E}Z_g = 0$ and $\{Z_g\}$ are independent across $g$. *(Here, the randomness is only over the independent SJLT blocks $\{\Phi_g\}_{g=1}^M$. We treat the wiring $\pi$ as fixed; if $\pi$ is random, the argument holds conditional on $\pi$ and hence also unconditionally.)* Moreover,

$$\|Sx\|_2^2 - \|x\|_2^2 = \sum_{g=1}^M Z_g \quad \text{since} \quad \frac{1}{\kappa}\sum_{g=1}^M \|x_{\mathcal{N}(g)}\|_2^2 = \|x\|_2^2$$

by Theorem A.1.

We now bound $\sum_g Z_g$ using Bernstein concentration for independent sub-exponential variables. For completeness, we record the exact Bernstein inequality we use, together with a standard definition of the "$(\nu^2, b)$" parameters.

**Definition A.6** (Sub-exponential with parameters $(\nu^2, b)$). A centered random variable $X$ is sub-exponential with parameters $(\nu^2, b)$ if

$$\mathbb{E}\exp(\lambda X) \le \exp\left(\frac{\lambda^2 \nu^2}{2}\right) \qquad \text{for all } |\lambda| < 1/b. \quad (30)$$

**Lemma A.7** (Bernstein inequality for independent sub-exponential variables). *Let $\{X_i\}_{i=1}^N$ be independent, centered random variables. Suppose each $X_i$ is sub-exponential with parameters $(\nu_i^2, b_i)$ in the sense of Theorem A.6. Then there is an absolute constant $c > 0$ such that for every $t \ge 0$,*

$$\Pr\left[\left|\sum_{i=1}^N X_i\right| \ge t\right] \le 2\exp\left(-c\min\left\{\frac{t^2}{\sum_i \nu_i^2}, \frac{t}{\max_i b_i}\right\}\right). \quad (31)$$

*This is Vershynin (2018, Corollary 2.8.3).*

Define

$$\tilde{Z}_g := \|\Phi_g x_{\mathcal{N}(g)}\|_2^2 - \|x_{\mathcal{N}(g)}\|_2^2, \qquad q_g := \|x_{\mathcal{N}(g)}\|_2^2.$$

Applying Theorem A.2 with $v = x_{\mathcal{N}(g)}$ yields, for all $u > 0$,

$$\mathbb{P}[|\tilde{Z}_g| > u q_g] \le 4\exp(-c\min\{u^2 B_r, us\}).$$

We now convert this tail bound into explicit sub-exponential parameters. Rewrite the previous display in terms of an arbitrary threshold $t > 0$ by setting $u = t/q_g$:

$$\mathbb{P}[|\tilde{Z}_g| > t] \le 4\exp\left(-c\min\left\{\frac{B_r t^2}{q_g^2}, \frac{st}{q_g}\right\}\right). \quad (32)$$

We now check the mgf condition in Theorem A.6 directly. Define

$$\nu_{g,0}^2 := \frac{q_g^2}{B_r} \qquad \text{and} \qquad b_{g,0} := \frac{q_g}{s}. \quad (33)$$

Then (32) can be rewritten as

$$\Pr[|\tilde{Z}_g| > t] \le 4\exp\left(-c\min\{t^2/\nu_{g,0}^2, t/b_{g,0}\}\right). \quad (34)$$

We now derive a Bernstein-type mgf bound from (34). Fix $\lambda \ge 0$ and use the identity

$$\mathbb{E}e^{\lambda \tilde{Z}_g} = 1 + \int_0^\infty \lambda e^{\lambda t}\Pr[\tilde{Z}_g > t]dt + \int_0^\infty \lambda e^{-\lambda t}\Pr[\tilde{Z}_g < -t]dt. \quad (35)$$

Since $\Pr[\tilde{Z}_g > t]$ and $\Pr[\tilde{Z}_g < -t]$ are both bounded by (34), we obtain

$$\mathbb{E}e^{\lambda \tilde{Z}_g} \le 1 + 8|\lambda|\int_0^\infty e^{|\lambda|t} \cdot \exp\left(-c\min\{t^2/\nu_{g,0}^2, t/b_{g,0}\}\right)dt. \quad (36)$$

The same bound holds for $\lambda < 0$ by applying (35) to $-\tilde{Z}_g$. Split the integral at $t_\star := \nu_{g,0}^2/b_{g,0}$. For $t \le t_\star$ we use the quadratic part of the minimum, which gives an integrand proportional to $\exp(|\lambda|t - ct^2/\nu_{g,0}^2)$. Completing the square yields

$$\int_0^{t_\star} \exp\left(|\lambda|t - ct^2/\nu_{g,0}^2\right)dt \le C\nu_{g,0} \cdot \exp\left(C\lambda^2\nu_{g,0}^2\right). \quad (37)$$

for an absolute constant $C > 0$. For $t \ge t_\star$ we use the linear part of the minimum. If $|\lambda| \le c/(2b_{g,0})$, then

$$\int_{t_\star}^\infty \exp\left(-(c/b_{g,0} - |\lambda|)t\right)dt$$
$$\le \int_{t_\star}^\infty \exp\left(-ct/(2b_{g,0})\right)dt \le C b_{g,0}. \quad (38)$$

Substituting (37) and (38) into (36) shows that there are absolute constants $C_0, c_0 > 0$ such that

$$\mathbb{E}\exp(\lambda \tilde{Z}_g) \le \exp\left(\frac{\lambda^2 C_0 \nu_{g,0}^2}{2}\right) \text{ for all } |\lambda| < c_0/b_{g,0}. \quad (39)$$

Combining (39) with Theorem A.6 gives the following explicit parameter bounds. There exist absolute constants $C, c' > 0$ such that $\tilde{Z}_g$ is sub-exponential with parameters

$$\nu_g^2 \le C\frac{q_g^2}{B_r} \qquad \text{and} \qquad b_g \le C\frac{q_g}{s}. \quad (40)$$

Since $Z_g = \tilde{Z}_g/\kappa$, scaling (30) shows that $Z_g$ is sub-exponential with parameters

$$\nu_g^2 \le C\frac{q_g^2}{\kappa^2 B_r} \qquad \text{and} \qquad b_g \le C\frac{q_g}{\kappa s}. \quad (41)$$

We now apply Theorem A.7 with $X_g = Z_g$.

$$\Pr\Big[\Big|\sum_{g=1}^M Z_g\Big| > u\|x\|_2^2\Big]$$

$$\leq 2\exp\Big(-c\min\Big\{\frac{u^2\|x\|_2^4}{\sum_g \nu_g^2}, \quad\quad (42)$$

$$\frac{u\|x\|_2^2}{\max_g b_g}\Big\}\Big).$$

It remains to bound $\sum_g \nu_g^2$ and $\max_g b_g$ in terms of $\mu_{\mathrm{nbr}}(x;\pi)$. Recall $q_g := \|x_{\mathcal{N}(g)}\|_2^2$. Then $\sum_g q_g = \kappa\|x\|_2^2$ by Theorem A.1, and

$$\sum_{g=1}^M q_g^2 \leq \big(\max_g q_g\big)\sum_{g=1}^M q_g = \kappa\|x\|_2^2 \max_g q_g.$$

Therefore

$$\sum_{g=1}^M \nu_g^2 \lesssim \frac{1}{\kappa^2 B_r}\sum_g q_g^2 \lesssim \frac{\|x\|_2^2 \max_g q_g}{\kappa B_r}.$$

Using $\max_g q_g = (\kappa\|x\|_2^2/M)\,\mu_{\mathrm{nbr}}(x;\pi)$ from (26) yields

$$\sum_g \nu_g^2 \lesssim \frac{\mu_{\mathrm{nbr}}(x;\pi)}{MB_r}\|x\|_2^4 = \frac{\mu_{\mathrm{nbr}}(x;\pi)}{k}\|x\|_2^4.$$

Similarly,

$$\max_g b_g \lesssim \frac{\max_g q_g}{\kappa s} = \frac{\mu_{\mathrm{nbr}}(x;\pi)}{Ms}\|x\|_2^2.$$

Plugging these into Bernstein yields

$$\Pr\Big[\big|\|Sx\|_2^2 - \|x\|_2^2\big| > u\,\|x\|_2^2\Big]$$

$$\leq 2\exp\Big(-c\min\Big\{\frac{u^2 k}{\mu_{\mathrm{nbr}}(x;\pi)}, \quad\quad (43)$$

$$\frac{u\,Ms}{\mu_{\mathrm{nbr}}(x;\pi)}\Big\}\Big).$$

Finally, since $\mu_{\mathrm{nbr}}(x;\pi) \leq M/\kappa$, we have $Ms/\mu_{\mathrm{nbr}}(x;\pi) \geq \kappa s$, giving the stated bound. □

## A.6. Oblivious Subspace Embedding (OSE)

For a fixed $U \in \mathbb{R}^{d\times r}$ with orthonormal columns, define the neighborhood coherence

$$\mu_{\mathrm{nbr}}(U;\pi) = \frac{M}{\kappa}\max_{g\in[M]}\|U_{\mathcal{N}(g)}\|_2^2,$$

matching (4). For any $x = Uw$, we have $\|x_{\mathcal{N}(g)}\|_2 \leq \|U_{\mathcal{N}(g)}\|_2\|w\|_2$, so $\mu_{\mathrm{nbr}}(x;\pi) \leq \mu_{\mathrm{nbr}}(U;\pi)$.

**Theorem A.8** (OSE for BLOCKPERM-SJLT). *Fix $U \in \mathbb{R}^{d\times r}$ with orthonormal columns. Let $S \sim$ BLOCKPERM-SJLT with parameters $(M, B_r, \kappa, s)$, independent $\{\Phi_g\}$, and any fixed wiring $\pi$. There exist absolute constants $C, c > 0$ such that if*

$$k = MB_r \geq C\frac{\mu_{\mathrm{nbr}}(U;\pi)}{\varepsilon^2}\Big(r + \log\frac{1}{\delta}\Big),$$

$$\kappa s \geq C\frac{1}{\varepsilon}\Big(r + \log\frac{1}{\delta}\Big), \quad\quad (44)$$

*then with probability at least $1-\delta$,*

$$\big\|U^\top S^\top SU - I_r\big\|_2 \leq \varepsilon,$$

*equivalently $S$ is an $(\varepsilon,\delta)$-OSE for $\mathrm{range}(U)$.*

*Proof.* We prove a uniform quadratic form bound from the fixed-vector guarantee in Theorem A.5. Let

$$A := U^\top S^\top SU - I_r.$$

Since $A$ is symmetric, $\|A\|_2 = \sup_{\|x\|_2=1}|x^\top Ax|$.

**Step 1: build an $\eta$-net.** Fix $\eta = 1/4$. A set $\mathcal{T} \subseteq \{x \in \mathbb{R}^r : \|x\|_2 = 1\}$ is an $\eta$-net if for every unit vector $x$ there exists $t \in \mathcal{T}$ with $\|x - t\|_2 \leq \eta$. There exists such a net with cardinality

$$|\mathcal{T}| \leq \Big(1 + \frac{2}{\eta}\Big)^r = 9^r. \quad\quad (45)$$

This follows, for example, from Vershynin (2018, Corollary 4.2.13).

**Step 2: control $\|SUt\|_2^2$ on the net.** Fix $t \in \mathcal{T}$ and set $x = Ut$. Then $\|x\|_2 = 1$ and $\mu_{\mathrm{nbr}}(x;\pi) \leq \mu_{\mathrm{nbr}}(U;\pi)$. Applying Theorem A.5 with $u = \varepsilon/2$ gives

$$\Pr\Big[\big|\|SUt\|_2^2 - \|Ut\|_2^2\big| > \tfrac{\varepsilon}{2}\|Ut\|_2^2\Big]$$

$$\leq 2\exp\Big(-c\min\Big\{\frac{\varepsilon^2 k}{\mu_{\mathrm{nbr}}(U;\pi)},\ \varepsilon\,\kappa s\Big\}\Big). \quad\quad (46)$$

Since $\|Ut\|_2 = 1$, this is the same as $\Pr[|t^\top At| > \varepsilon/2]$.

**Step 3: union bound over the net.** Let $\mathcal{E}$ be the event that $|t^\top At| \leq \varepsilon/2$ holds for all $t \in \mathcal{T}$. By (45) and a union bound,

$$\Pr[\mathcal{E}^c] \leq 2\cdot 9^r\,\exp\Big(-c\min\Big\{\frac{\varepsilon^2 k}{\mu_{\mathrm{nbr}}(U;\pi)},\ \varepsilon\,\kappa s\Big\}\Big). \quad\quad (47)$$

Under (44) (for a suitable absolute constant $C$), we have $\Pr[\mathcal{E}] \geq 1 - \delta$.

**Step 4: extend from the net to the full sphere.** We show that $\mathcal{E}$ implies $\|A\|_2 \leq \varepsilon$. This is a standard argument based on covering the sphere by a net. See, for example, Vershynin (2018, Exercise 4.4.4). Let $x_\star \in \mathbb{R}^r$ be a unit vector achieving $\|A\|_2 = |x_\star^\top A x_\star|$. Pick $t \in \mathcal{T}$ with $\|x_\star - t\|_2 \leq \eta$. Then

$$|x_\star^\top A x_\star - t^\top A t| = |(x_\star - t)^\top A x_\star + t^\top A(x_\star - t)|$$
$$\leq 2\|x_\star - t\|_2 \|A\|_2 \leq 2\eta \|A\|_2. \quad (48)$$

Rearranging (48) yields

$$(1 - 2\eta)\|A\|_2 \leq \sup_{t \in \mathcal{T}} |t^\top A t|.$$

On $\mathcal{E}$ we have $\sup_{t \in \mathcal{T}} |t^\top A t| \leq \varepsilon/2$. With $\eta = 1/4$, this gives $\|A\|_2 \leq \varepsilon$. This is equivalent to the OSE statement. $\quad\square$

### A.7. How $\kappa$ Enters the Bounds

At this point, the fixed-vector and OSE bounds depend on the neighborhood coherence $\mu_{\mathrm{nbr}}(U; \pi)$. The following lemma relates it to the standard localized-sketching quantity $\mu_{\mathrm{blk}}(U)$. It yields inequality (7) in the main text and shows that, in the worst case, increasing $\kappa$ can buy at most a factor-$1/\kappa$ improvement. The next subsection explains why randomized wirings can do substantially better: independent permutations *smooth* neighborhood coherence toward one as $\kappa$ grows.

**Lemma A.9** (Bounding neighborhood coherence by block coherence)**.** *Let* $U = [U^{(1)}; \ldots; U^{(M)}]$ *and define* $\mu_{\mathrm{blk}}(U) = M \max_h \|U^{(h)}\|_2^2$. *Then for any wiring* $\pi$,

$$\frac{1}{\kappa} \mu_{\mathrm{blk}}(U) \leq \mu_{\mathrm{nbr}}(U; \pi) \leq \mu_{\mathrm{blk}}(U).$$

*Proof.* For any $g$,

$$\|U_{\mathcal{N}(g)}\|_2^2 = \left\| \begin{bmatrix} U^{(\pi_1(g))} \\ \vdots \\ U^{(\pi_\kappa(g))} \end{bmatrix} \right\|_2^2$$
$$\leq \sum_{\ell=1}^{\kappa} \|U^{(\pi_\ell(g))}\|_2^2 \quad (49)$$
$$\leq \kappa \max_h \|U^{(h)}\|_2^2.$$

Taking $\max_g$ and multiplying by $M/\kappa$ gives $\mu_{\mathrm{nbr}}(U; \pi) \leq \mu_{\mathrm{blk}}(U)$.

For the lower bound, pick $h^\star \in \arg\max_h \|U^{(h)}\|_2^2$ and choose any $\ell$. There exists $g^\star$ with $\pi_\ell(g^\star) = h^\star$ (since $\pi_\ell$ is a permutation), so $\|U_{\mathcal{N}(g^\star)}\|_2^2 \geq \|U^{(h^\star)}\|_2^2$. Thus

$$\mu_{\mathrm{nbr}}(U; \pi) = \frac{M}{\kappa} \max_g \|U_{\mathcal{N}(g)}\|_2^2$$
$$\geq \frac{M}{\kappa} \|U^{(h^\star)}\|_2^2 = \frac{1}{\kappa} \mu_{\mathrm{blk}}(U). \quad (50)$$

$\square$

### A.8. Smoothing of Neighborhood Coherence with Randomized Permutations

The deterministic bounds in Theorem A.9 capture the best- and worst-case effects of increasing $\kappa$. If the wiring is randomized, we can say more: a union of random permutations tends to *balance* block mass across neighborhoods, which drives $\mu_{\mathrm{nbr}}(U; \pi)$ toward 1.

**Lemma A.10** (Matrix Chernoff (upper tail))**.** *Let* $\{X_\ell\}_{\ell=1}^{\kappa}$ *be independent random PSD matrices in* $\mathbb{R}^{r \times r}$. *Assume* $0 \preceq X_\ell \preceq R I_r$ *almost surely and* $\mathbb{E}[X_\ell] = \mu I_r$ *for some* $\mu > 0$. *Then for any* $\tau \geq 0$,

$$\mathbb{P}\left[ \lambda_{\max}\left( \sum_{\ell=1}^{\kappa} X_\ell \right) \geq (1 + \tau)\kappa\mu \right]$$
$$\leq r \exp\left( -c \min\{\tau^2, \tau\} \frac{\kappa\mu}{R} \right)$$

*for an absolute constant* $c > 0$.

Theorem A.10 is a standard matrix Chernoff inequality. One convenient reference is Tropp (2012, Corollary 5.2). Our stated form follows by applying that corollary and using the elementary bound $\frac{e^\tau}{(1+\tau)^{1+\tau}} \leq \exp(-c \min\{\tau^2, \tau\})$ for an absolute constant $c > 0$.

**Proposition A.11** (Random permutations smooth neighborhood coherence)**.** *Let* $U \in \mathbb{R}^{d \times r}$ *have orthonormal columns and be partitioned into* $M$ *row blocks* $U = [U^{(1)}; \ldots; U^{(M)}]$ *of size* $B_c$. *Let* $\mu_{\mathrm{blk}}(U) = M \max_{h \in [M]} \|U^{(h)}\|_2^2$. *Suppose the wiring permutations* $\{\pi_\ell\}_{\ell=1}^{\kappa}$ *are independent and uniformly random. Then with probability at least* $1 - \delta$ *over the choice of the permutations, let* $L := \log \frac{2Mr}{\delta}$.

$$\mu_{\mathrm{nbr}}(U; \pi) \leq 1 + C\left( \sqrt{\frac{\mu_{\mathrm{blk}}(U) L}{\kappa}} + \frac{\mu_{\mathrm{blk}}(U) L}{\kappa} \right). \quad (51)$$

*for an absolute constant* $C > 0$. *In particular, when* $\kappa \gtrsim \mu_{\mathrm{blk}}(U) \log\left( \frac{Mr}{\delta} \right)$, *we have* $\mu_{\mathrm{nbr}}(U; \pi) = \mathcal{O}(1)$ *with probability* $1 - \delta$.

*Proof.* Define PSD matrices $A_h = U^{(h)\top} U^{(h)} \in \mathbb{R}^{r \times r}$. Since $U$ has orthonormal columns, $\sum_{h=1}^{M} A_h = U^\top U = I_r$. Let $\alpha := \max_h \|U^{(h)}\|_2^2$, so $\mu_{\mathrm{blk}}(U) = M\alpha$. Let $L := \log\left( \frac{2Mr}{\delta} \right)$.

Fix an output block $g \in [M]$ and consider the neighborhood Gram matrix

$$G_g := U_{\mathcal{N}(g)}^\top U_{\mathcal{N}(g)} = \sum_{\ell=1}^{\kappa} A_{\pi_\ell(g)}.$$

Let $X_\ell := A_{\pi_\ell(g)}$. For fixed $g$, the indices $\pi_\ell(g)$ are i.i.d. uniform on $[M]$ (independent uniform permutations), hence

$$\mathbb{E}[X_\ell] = \frac{1}{M}\sum_{h=1}^{M} A_h = \frac{1}{M}I_r, \qquad 0 \preceq X_\ell \preceq \alpha I_r.$$

Applying Theorem A.10 with $R = \alpha$ and $\mu = 1/M$ gives, for any $\tau \geq 0$,

$$\mathbb{P}\Big[\|G_g\|_2 \geq (1+\tau)\,\frac{\kappa}{M}\Big] \;\leq\; r\,\exp\Big(-c\,\min\{\tau^2,\tau\}\,\frac{\kappa}{\alpha M}\Big).$$

Choose $\tau \asymp \sqrt{\frac{\alpha M}{\kappa}L} + \frac{\alpha M}{\kappa}L$ so that the RHS is at most $\delta/M$. A union bound over $g \in [M]$ yields that with probability at least $1 - \delta$,

$$\max_{g\in[M]} \|G_g\|_2 \leq (1+\tau)\frac{\kappa}{M}.$$

Finally, $\mu_{\mathrm{nbr}}(U;\pi) = \frac{M}{\kappa}\max_g \|G_g\|_2 \leq 1 + \tau$. Substituting $\alpha M = \mu_{\mathrm{blk}}(U)$ gives (51). $\qquad\square$

**Remark A.12** (Theory vs. Practice: Independence vs. Distinctness). For theorems in Section A we model $\pi_1,\ldots,\pi_\kappa$ as *independent* uniform random permutations. Our kernel FLASHSKETCH instead generates $\kappa$ *distinct* permutations from a structured family (see Section D), which introduces mild dependence between the neighborhood indices $\{\pi_\ell(g)\}_{\ell=1}^\kappa$. This mismatch is small in the intended regime $\kappa \ll M$: for events depending only on a single neighborhood, sampling without replacement is close to sampling with replacement, and collisions $\pi_\ell(g) = \pi_{\ell'}(g)$ are rare when $\kappa^2 \ll M$. We therefore present the i.i.d. model to keep the analysis clean and highlight the dominant scaling with $\kappa$. Tightening the dependence analysis is left as an interesting direction for future work.

## B. Additional Details on FLASHSKETCH

### B.1. Split-$B_c$ Fallback

FLASHSKETCH assigns each output tile to exactly one thread-block and performs no global atomics. When $M \cdot \lceil n/T_n \rceil$ is too small to saturate the GPU and achieve maximum occupancy (e.g., small $n$ and large $d$), we resort to a split-$B_c$ fallback analogous to split-$K$ GEMM (Osama et al., 2023): we slice each input block along rows, launch an extra grid dimension for the slice id, and accumulate partial tiles with global atomics. While this materially deviates from our guiding principle of eliminating global atomics, it still reduces global atomics by a factor of the number of slices which is less than prior kernels such as (Hu et al., 2025), and it enables high occupancy for small problems.

### B.2. Tuning

FLASHSKETCH has several kernel parameters that we encode as compile-time constants via templates: the block

sizes $(B_r, B_c)$ and the tile sizes $(T_k, T_n)$. Tile sizes affect performance by controlling shared memory usage and occupancy. In our evaluation, we tune these parameters once over a small menu of $(B_r, T_n, T_k, \kappa, s)$ templates for different input shapes and dispatch the best choice of tiling based on observed performance. Our tuning procedure is not exhaustive, and we leave a more thorough exploration of the kernel design space to future work.

## C. FLASHBLOCKROW: A Fast but Fragile Alternative

While designing FLASHSKETCH, our choices were guided by balancing the tension between GPU efficiency and sketching robustness. An interesting direction we also explored is block-row sampling sketches, which are designed to maximize GPU friendliness with less concern for robustness. Existing theoretical work explores these ideas. We refer the reader to (Drineas et al., 2012; Tropp, 2011; Chenakkod et al., 2024) for more details. The primary bottlenecks in FLASHSKETCH are the shared-memory atomics, and the need to repeatedly stream $\kappa$ input blocks per output block. The latter is a consequence of the fixed block column sparsity pattern of BLOCKPERM-SJLT, which was designed specifically to maintain theoretical robustness of the sketch. An alternative is to relax this requirement, and simply independently sample $\kappa$ blocks per row of $S$, and simplify the kernel to pure gather operations without any atomics at any level. We call this approach FLASHBLOCKROW.

A natural consequence is that we are no longer able to guarantee a fixed number of nonzeros per column of $S$, and in fact it is possible for some columns to have no nonzeros. This is terrible for sketching quality, as those input coordinates are completely ignored. Specifically, its distortion depends on (block) leverage scores and can deteriorate in high-coherence regimes, especially when $d$ is large and $k$ is small. This fragility means that we cannot obtain OSE style guarantees for FLASHBLOCKROW. However, the simplified gather-reduce implementation is extremely fast as it both eliminates global atomics and reduces the number of times the input is read from $\kappa$ to 1.

Interestingly, we find that despite its theoretical fragility, FLASHBLOCKROW can perform well in practice on a small subset of datasets, while being significantly faster than FLASHSKETCH and all other baselines. We show this in the ablations in Section F.

For reference, we include pseudocode for one thread-block of the FLASHBLOCKROW kernel in Algorithm 2.

**Algorithm 2** FLASHBLOCKROW thread-block level pseudocode

---

**Require:** $A \in \mathbb{R}^{d \times n}$, block sizes $B_c, B_r$, tile width $T_n$, and integers $\kappa, s$
 1: Partition $A$ into input blocks $A^{(h)} \in \mathbb{R}^{B_c \times n}$ for $h \in [M]$
 2: **for** each thread-block indexed by $(g, j)$ with $g \in [M]$ and $j \in \{0, \ldots, \lceil n/T_n \rceil - 1\}$ **do**
 3:      Sample a block neighborhood $\mathcal{N}_{\text{row}}(g) \subset [M]$ with $|\mathcal{N}_{\text{row}}(g)| = \kappa$
 4:      Initialize a shared memory output tile $sY \in \mathbb{R}^{B_r \times T_n}$ to zero
 5:      **for** each $h \in \mathcal{N}_{\text{row}}(g)$ **do**
 6:          Load $A^{(h)}_{:, jT_n:(j+1)T_n}$ into shared memory
 7:          **for** $r = 1$ to $B_r$ **do**
 8:              Sample indices $i_1, \ldots, i_s \in [B_c]$ uniformly and signs $\sigma_1, \ldots, \sigma_s \in \{\pm 1\}$
 9:              $sY_{r,:} \leftarrow sY_{r,:} + \frac{1}{\sqrt{\kappa s}} \sqrt{\frac{d}{k}} \sum_{t=1}^{s} \sigma_t A^{(h)}_{i_t, jT_n:(j+1)T_n}$
10:          **end for**
11:      **end for**
12:      Write $sY$ to $SA[gB_r : (g+1)B_r, jT_n : (j+1)T_n]$
13: **end for**

---

## D. Permutation Wiring and On-The-Fly Generation

The block-level sparsity pattern of BLOCKPERM-SJLT can be viewed as a bipartite graph between output blocks $g \in [M]$ (rows of $S$ grouped into blocks of size $B_r$) and input blocks $h \in [M]$ (rows of the input grouped into blocks of size $B_c$). In the paper we describe this pattern as a *union of permutations*: we draw $\kappa$ bijections $\{\pi_\ell\}_{\ell=1}^{\kappa}$ on $[M]$, and connect each output block $g$ to the $\kappa$ input blocks $\pi_1(g), \ldots, \pi_\kappa(g)$.

**Multiset versus set union.** It is worth being precise about what "union" means. If we keep the $\kappa$ matchings as a *multiset*, then each $g$ has exactly $\kappa$ incident edges and each $h$ also appears exactly $\kappa$ times when edges are counted with multiplicity. This multiset view is the one used in our proofs. It is also consistent with the kernel: if two matchings place an edge on the same block position $(g, h)$, the resulting sketch simply adds two independent SJLT blocks into the same output tile.

If instead we interpret the block pattern as a 0/1 sparsity mask (a *set* of edges), then two distinct permutations can overlap on some edges. In that case, the number of *distinct* nonzero blocks in a block row can be smaller than $\kappa$. For example, with $M = 4$, the identity permutation and a swap of the first two entries are distinct, but their set union gives only one distinct neighbor for $g = 3$ and $g = 4$. For

our purposes this overlap is undesirable because it reduces mixing and can lead to redundant reads.

**Edge-disjoint block wiring.** To avoid such overlaps, BLOCKPERM-SJLT uses the stronger block wiring in which the $\kappa$ neighbors of every $g$ are distinct. Equivalently, the $\kappa$ perfect matchings are edge-disjoint. This yields a simple $\kappa$-regular bipartite graph and ensures that every block row and block column contains exactly $\kappa$ nonzero blocks.

**Full-cycle affine construction.** Rather than materializing $\kappa$ permutation tables, the kernel generates neighbors on the fly using an affine map modulo $M$. Let

$$f(x) := (ax + b) \bmod M. \tag{52}$$

The kernel chooses integers $a$ and $b$ such that the recurrence $x_{t+1} = f(x_t)$ has period $M$. One sufficient set of conditions is the classical full-period criterion for linear congruential generators[1] (Hull & Dobell, 1962):

(a) $\gcd(b, M) = 1$,
(b) $a - 1$ is divisible by every prime factor of $M$,
(c) if $4 \mid M$, then $4 \mid (a - 1)$.

Under these conditions, for any starting value $x_0 \in [M]$, the sequence $x_0, x_1, \ldots, x_{M-1}$ visits every element of $[M]$ exactly once.

We then define the $\kappa$ neighbors of block $g$ by iterating this map:

$$\pi_\ell(g) := f^\ell(g), \qquad \ell = 1, \ldots, \kappa, \tag{53}$$

where $f^\ell$ denotes $\ell$-fold composition. As long as $\kappa \leq M$, these neighbors are distinct for every $g$, and the resulting block sparsity pattern is exactly $\kappa$-regular as desired. This is the permutation wiring used by FLASHSKETCH.

**Within-block hashing.** Once a block neighbor $h \in \mathcal{N}(g)$ is selected, FLASHSKETCH applies a row-partitioned SJLT within the corresponding $(g, h)$ block. The kernel does not store row indices. Instead, it uses a fast 32-bit mixing hash to generate, for each input row inside the block, both a destination row index in $[B_r]$ and an independent Rademacher sign. It generates $s$ unique row indices using an affine permutation map similar to the one above, with scale and shift parameters generated from the hash. This realizes the distributional definition of BLOCKPERM-SJLT while keeping the kernel fully on-the-fly.

---

[1] These conditions are also used by FLASHSKETCH to ensure that iterates visit all block indices before repeating.

# E. Additional Details on GraSS

## E.1. GraSS Pipeline

At a high level, GraSS constructs a *feature cache* for the training set: for each training example $z_i$, it computes a per-example gradient vector $g_i$ (or a structured per-layer analogue), applies gradient sparsification/compression, and then applies a random projection to produce a low-dimensional representation $\phi_i \in \mathbb{R}^k$ that is stored for fast querying. Given a test/query example $z$, GraSS computes an analogous representation $\phi_z$ and then produces an attribution vector $\tau(z) \in \mathbb{R}^n$ (one score per training example) using the cached features and a small downstream solve. In GraSS, the random projection step is a key bottleneck: it is invoked for every training example during cache construction (and for every query at inference time), so kernel efficiency directly impacts end-to-end wall time. We refer the reader to (Hu et al., 2025) for further details on the GraSS pipeline.

## E.2. Details on Linear Datamodeling Score (LDS)

GraSS evaluates quantitative data-attribution quality using the linear datamodeling score (LDS) introduced by TRAK (Park et al., 2023) and adopted by GraSS (Hu et al., 2025). We summarize the protocol here to make our end-to-end evaluation fully explicit.

Let $S = \{z_i\}_{i=1}^N$ denote the training set and let $f(z; \theta)$ denote a scalar model output of interest evaluated at an example $z$ (e.g., a logit or loss, depending on the task and configuration). LDS is a counterfactual benchmark: it compares (a) true model outputs obtained by retraining on randomly subsampled training sets to (b) predictions obtained by summing attribution scores over the same subsampled sets.

**Training subsets.** Following GraSS, we sample $m = 50$ random subsets $\{S_j\}_{j=1}^m$ of the training set, each of size $|S_j| = \alpha N$ with $\alpha = 0.5$ (i.e., 50% subsampling). For each subset $S_j$, we train a model to obtain parameters $\theta^\star(S_j)$ using the same training procedure as the GraSS implementation.

**Attribution-derived counterfactual predictor.** A data attribution method $\tau$ produces a per-example score vector $\tau(z, S) \in \mathbb{R}^N$ for a fixed example of interest $z$ (typically a validation/test example). TRAK defines the attribution-derived prediction for the counterfactual output on subset $S_j$ as the sum of the scores of the training examples included in that subset:

$$g_\tau(z, S_j; S) := \sum_{i:\, z_i \in S_j} \tau(z, S)_i = \tau(z, S)^\top \mathbf{1}_{S_j}, \quad (54)$$

where $\mathbf{1}_{S_j} \in \{0, 1\}^N$ is the indicator vector of $S_j$. This is the "additive datamodel" prediction used by TRAK.

**Definition of LDS.** For each example of interest $z$, we form two length-$m$ sequences:

$$y_j(z) := f(z; \theta^\star(S_j)), \quad (55)$$
$$\hat{y}_j(z) := g_\tau(z, S_j; S). \quad (56)$$

The LDS for the example $z$ is the Spearman rank correlation (Spearman, 1961) between these sequences:

$$\mathrm{LDS}(\tau, z) := \rho_{\mathrm{Spearman}} \left( \{y_j(z)\}_{j=1}^m, \{\hat{y}_j(z)\}_{j=1}^m \right). \quad (57)$$

The reported LDS is the average of $\mathrm{LDS}(\tau, z)$ over the evaluation examples $z$ used by the GraSS protocol.

In our integration experiments, the only change to GraSS is the SJLT projection kernel used inside the gradient-compression step. All other components and the LDS evaluation code path are identical to the official GraSS repository (TRAIS-Lab, 2025). The model used is the default in the GraSS repo, a 3-layer MLP with ReLU activations and a total of $109,386$ parameters. We use the same training hyperparameters as in GraSS, and sketch down from dimension $4k$ to $k$ for $k \in \{1024, 2048, 4096\}$. We use LDS as our primary measure of attribution quality and report Pareto frontiers of LDS vs. projection time.

# F. Additional Experiments and Ablations for RandNLA Tasks

## F.1. Metrics

We define all error metrics exactly as they are computed in the benchmark code. Let $A \in \mathbb{R}^{d \times n}$, let $S \in \mathbb{R}^{k \times d}$, and set $SA \in \mathbb{R}^{k \times n}$.

### F.1.1. GRAM APPROXIMATION

We form $G = A^\top A$ and $\hat{G} = (SA)^\top (SA)$ and report the relative Frobenius error.

$$E_{\mathrm{Gram}} = \|\hat{G} - G\|_F,$$
$$E_{\mathrm{Gram,rel}} = \begin{cases} \|\hat{G} - G\|_F / \|G\|_F, & \|G\|_F > 0, \\ \|\hat{G} - G\|_F, & \|G\|_F = 0. \end{cases}$$

### F.1.2. OSE SPECTRAL ERROR

We build an orthonormal basis $Q \in \mathbb{R}^{d \times r}$. The default is the column-space variant $Q = \mathrm{qr}(A)$ with $r = \min(\mathrm{r}, d, n)$, while probing uses a Gaussian $Q$ with $r = \mathtt{probes}$.

$$\hat{Q} = SQ,$$
$$E_{\mathrm{OSE}} = \|\hat{Q}^\top \hat{Q} - I_r\|_2.$$

### F.1.3. RIDGE REGRESSION RESIDUAL

For $b \in \mathbb{R}^d$ and $\lambda > 0$, we solve

$$x = \arg\min_x \|SAx - Sb\|_2^2 + \lambda\|x\|_2^2,$$

$$E_{\mathrm{ridge}} = \begin{cases} \|Ax - b\|_2/\|b\|_2, & \|b\|_2 > 0, \\ \|Ax - b\|_2, & \|b\|_2 = 0. \end{cases}$$

### F.1.4. SKETCH-AND-SOLVE LEAST SQUARES

We solve $x = \arg\min_x \|SAx - Sb\|_2$ and report the same residual definition $E_{\mathrm{ridge}}$.

## Sketch dimension (k)

| | |
|---|---|
| ● | 64 |
| ■ | 256 |
| ▲ | 512 |
| ◆ | 1024 |
| ✚ | 2048 |
| ✖ | 4096 |

*Figure 5.* **Legend for sketch dimension.** Marker shapes indicate sketch dimension $k$ for the ablation plots in this appendix.

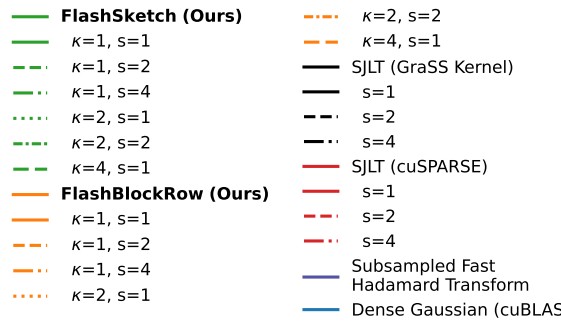

*Figure 6.* **Legend for methods and parameters.** Line colors/styles indicate method families and $(\kappa, s)$ settings used across the appendix ablations.

## F.2. Gram-Matrix Approximation Ablations

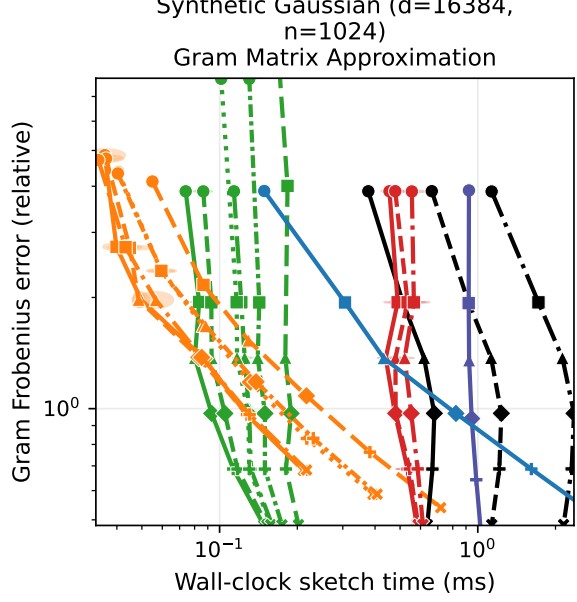

*Figure 7.* **Gram-matrix approximation ablations.** Synthetic Gaussian (d=16384, n=1024). GPU: NVIDIA GeForce RTX 4090.

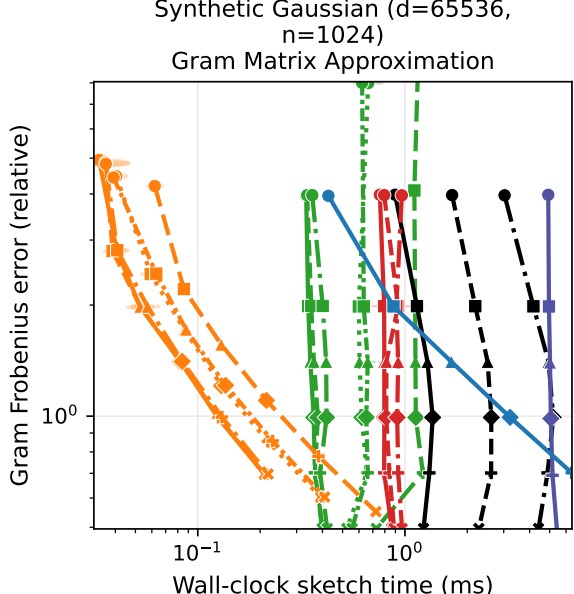

*Figure 8.* **Gram-matrix approximation ablations.** Synthetic Gaussian (d=65536, n=1024). GPU: NVIDIA GeForce RTX 4090.

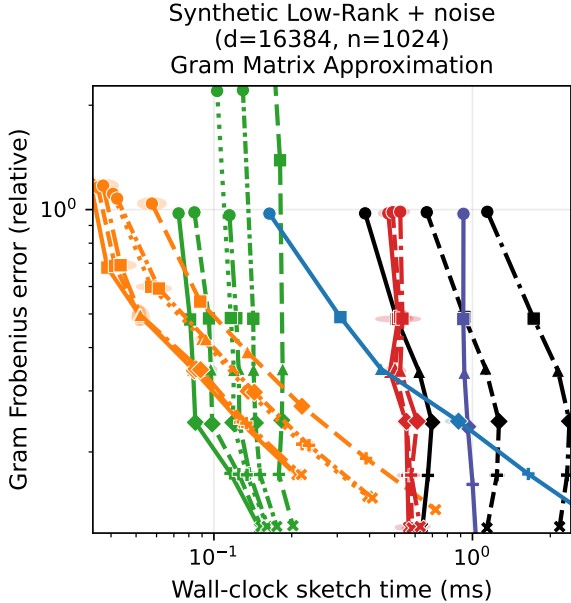

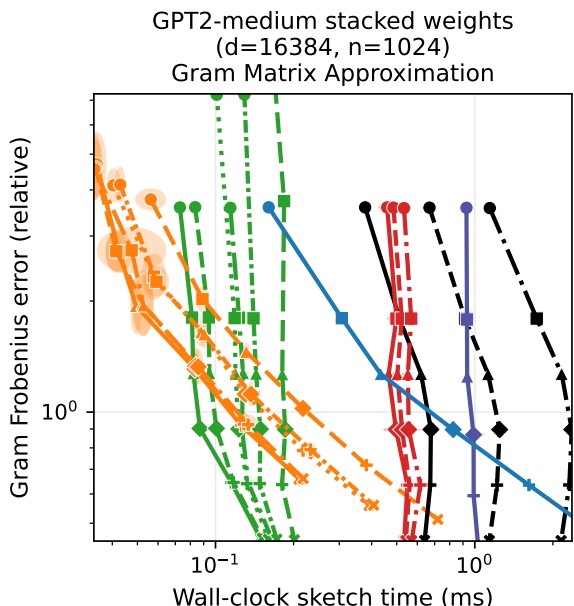

*Figure 9.* **Gram-matrix approximation ablations.** Synthetic Low-Rank + noise (d=16384, n=1024). GPU: NVIDIA GeForce RTX 4090.

*Figure 11.* **Gram-matrix approximation ablations.** GPT2-medium stacked weights (d=16384, n=1024). GPU: NVIDIA GeForce RTX 4090.

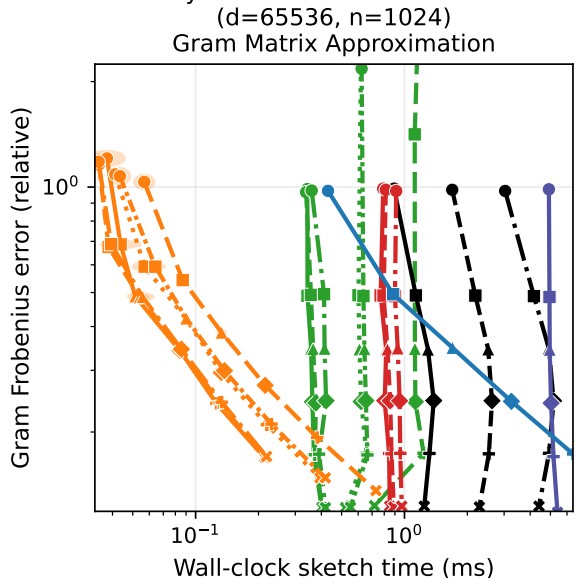

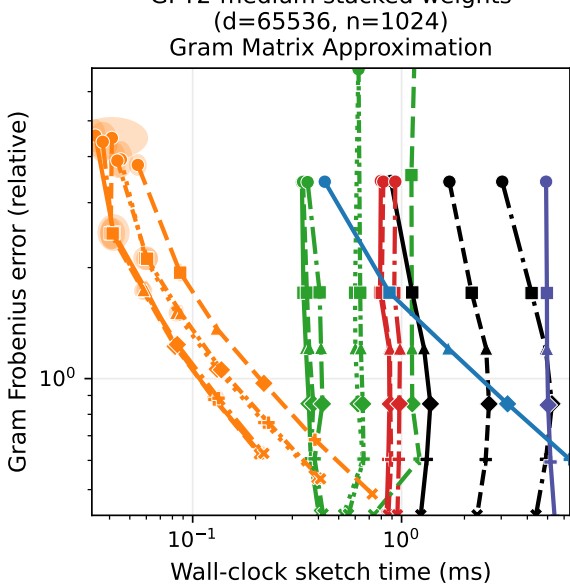

*Figure 10.* **Gram-matrix approximation ablations.** Synthetic Low-Rank + noise (d=65536, n=1024). GPU: NVIDIA GeForce RTX 4090.

*Figure 12.* **Gram-matrix approximation ablations.** GPT2-medium stacked weights (d=65536, n=1024). GPU: NVIDIA GeForce RTX 4090.

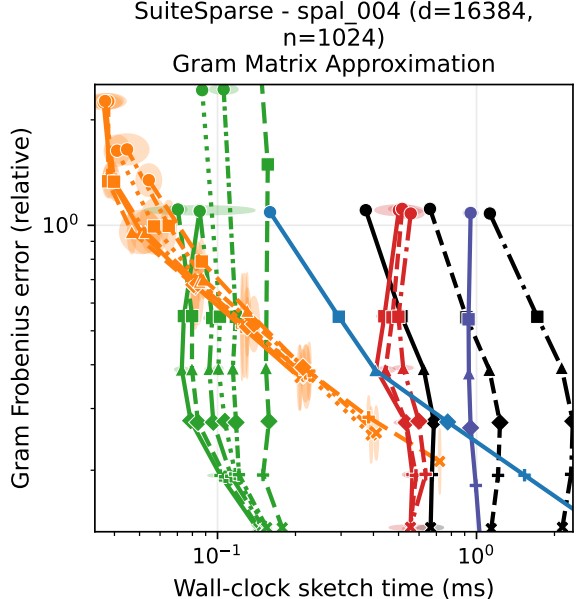

*Figure 13.* **Gram-matrix approximation ablations.** SuiteSparse - spal_004 (d=16384, n=1024). GPU: NVIDIA GeForce RTX 4090.

*Figure 15.* **Gram-matrix approximation ablations.** Synthetic Gaussian (d=131072, n=512). GPU: NVIDIA GeForce RTX 4090.

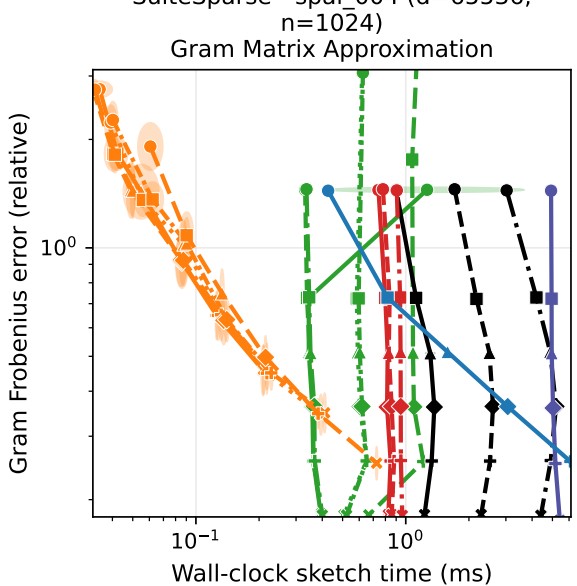
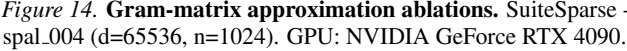
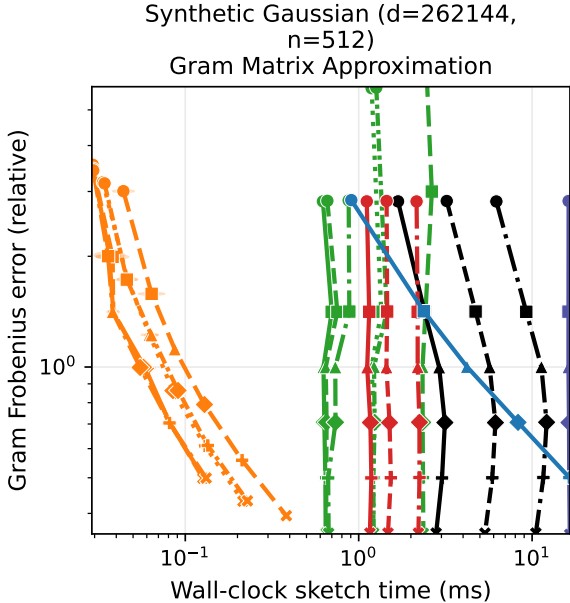

*Figure 14.* **Gram-matrix approximation ablations.** SuiteSparse - spal_004 (d=65536, n=1024). GPU: NVIDIA GeForce RTX 4090.

*Figure 16.* **Gram-matrix approximation ablations.** Synthetic Gaussian (d=262144, n=512). GPU: NVIDIA GeForce RTX 4090.

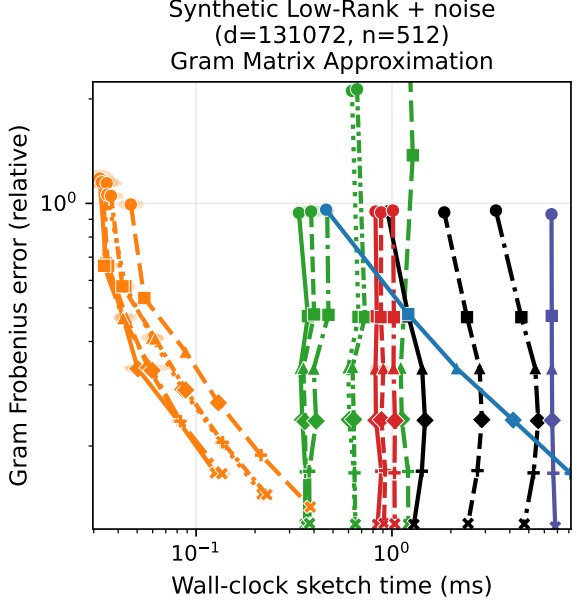

*Figure 17.* **Gram-matrix approximation ablations.** Synthetic Low-Rank + noise (d=131072, n=512). GPU: NVIDIA GeForce RTX 4090.

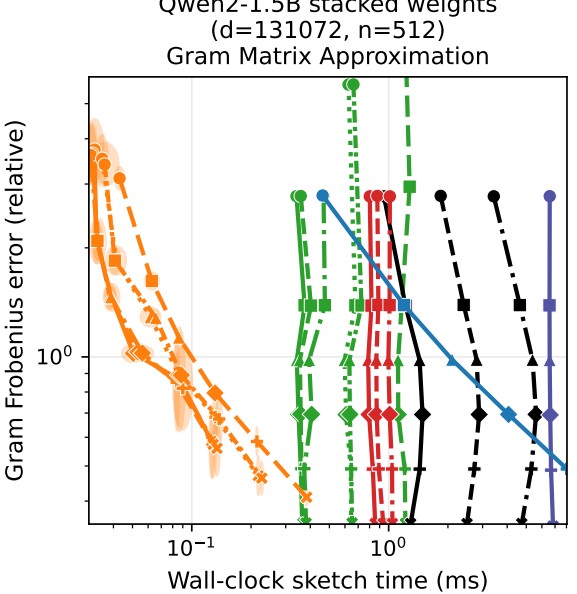

*Figure 19.* **Gram-matrix approximation ablations.** Qwen2-1.5B stacked weights (d=131072, n=512). GPU: NVIDIA GeForce RTX 4090.

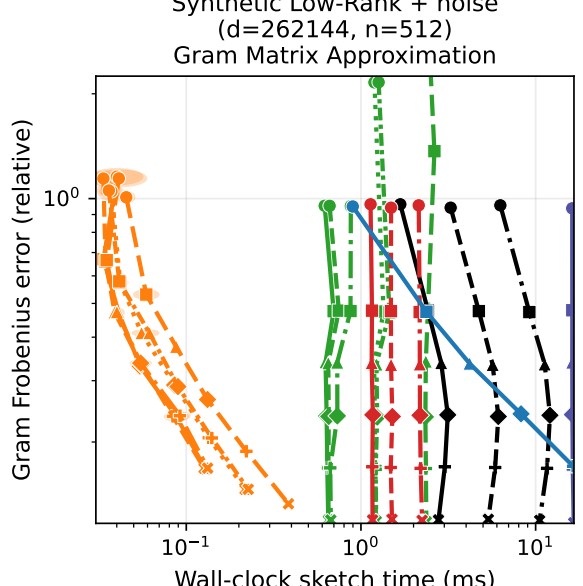

*Figure 18.* **Gram-matrix approximation ablations.** Synthetic Low-Rank + noise (d=262144, n=512). GPU: NVIDIA GeForce RTX 4090.

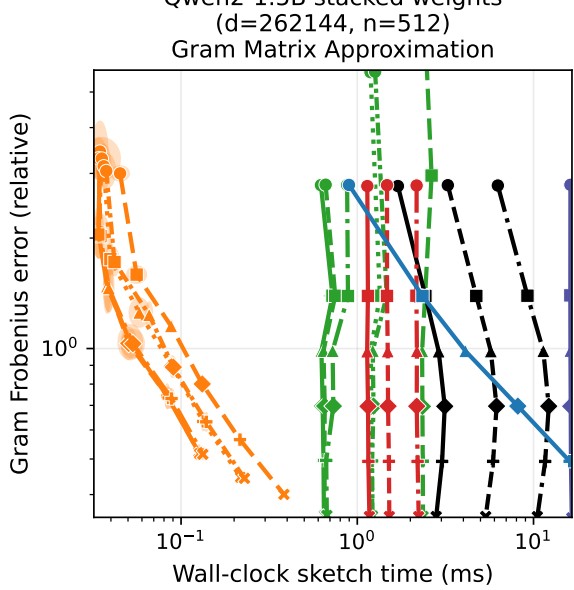

*Figure 20.* **Gram-matrix approximation ablations.** Qwen2-1.5B stacked weights (d=262144, n=512). GPU: NVIDIA GeForce RTX 4090.

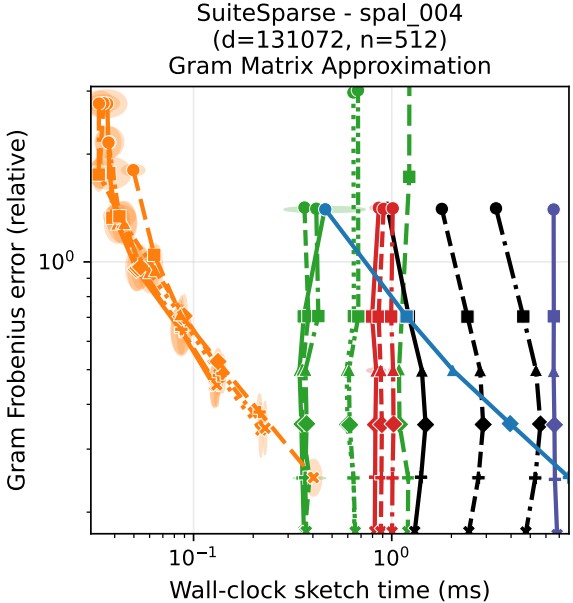

*Figure 21.* **Gram-matrix approximation ablations.** SuiteSparse - spal_004 (d=131072, n=512). GPU: NVIDIA GeForce RTX 4090.

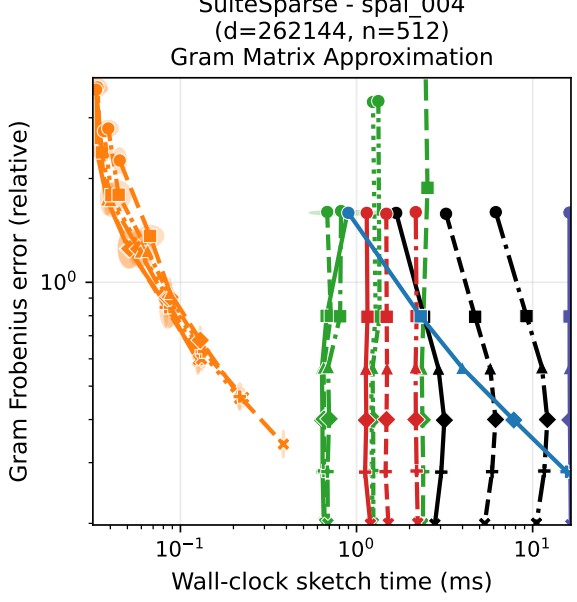

*Figure 22.* **Gram-matrix approximation ablations.** SuiteSparse - spal_004 (d=262144, n=512). GPU: NVIDIA GeForce RTX 4090.

## F.3. OSE Spectral-Error Ablations

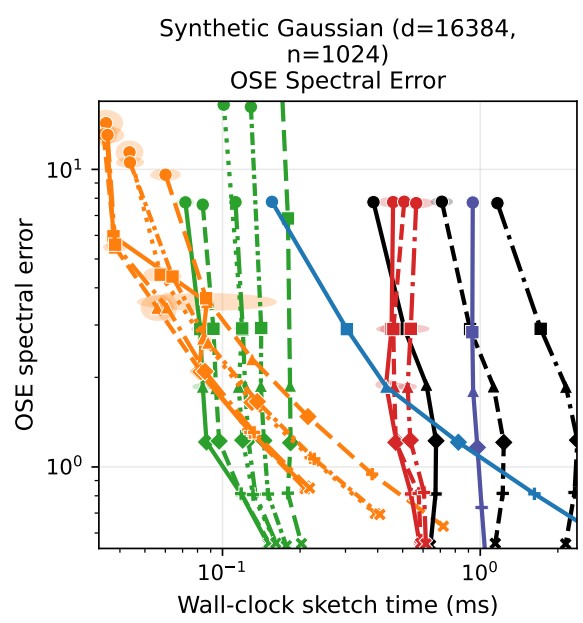

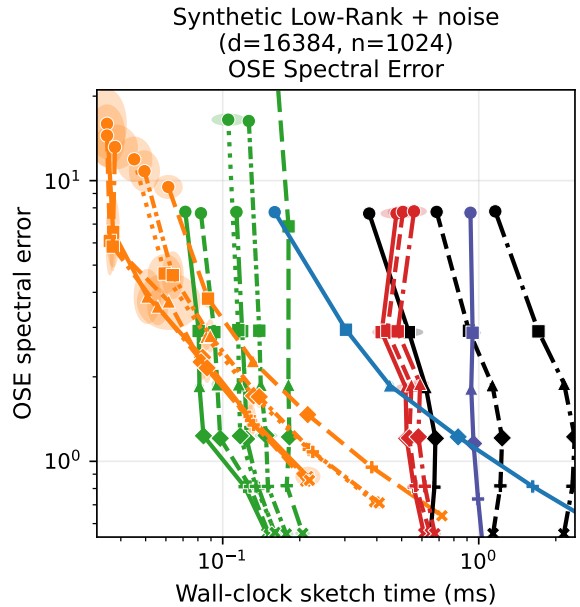

*Figure 23.* **OSE spectral-error ablations.** Synthetic Gaussian (d=16384, n=1024). GPU: NVIDIA GeForce RTX 4090.

*Figure 25.* **OSE spectral-error ablations.** Synthetic Low-Rank + noise (d=16384, n=1024). GPU: NVIDIA GeForce RTX 4090.

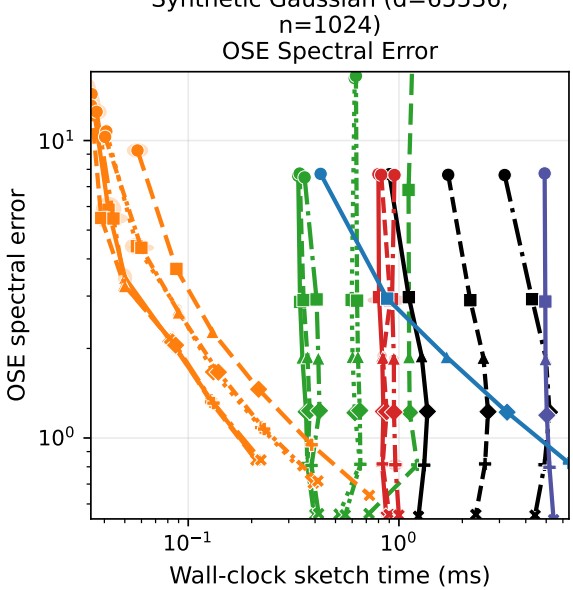

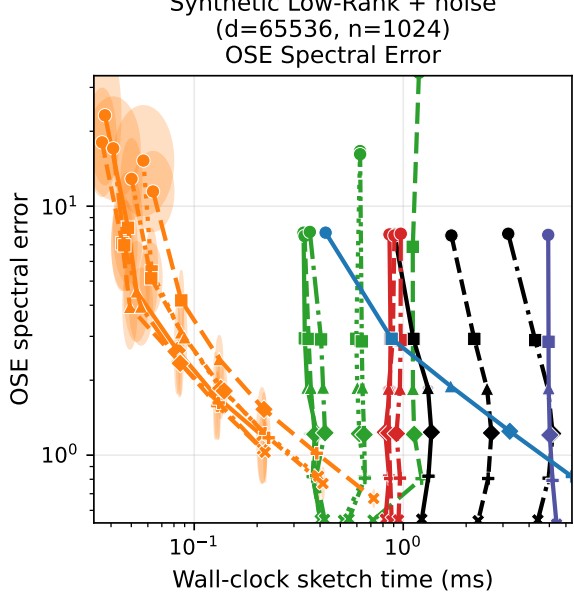

*Figure 24.* **OSE spectral-error ablations.** Synthetic Gaussian (d=65536, n=1024). GPU: NVIDIA GeForce RTX 4090.

*Figure 26.* **OSE spectral-error ablations.** Synthetic Low-Rank + noise (d=65536, n=1024). GPU: NVIDIA GeForce RTX 4090.

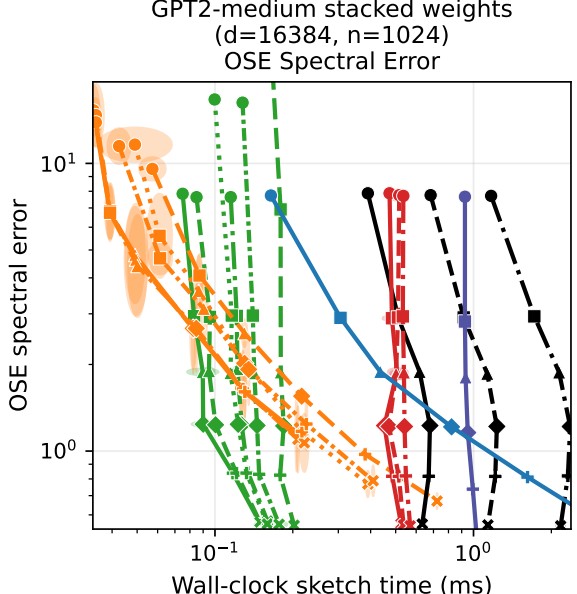

*Figure 27.* **OSE spectral-error ablations.** GPT2-medium stacked weights (d=16384, n=1024). GPU: NVIDIA GeForce RTX 4090.

*Figure 29.* **OSE spectral-error ablations.** SuiteSparse - spal_004 (d=16384, n=1024). GPU: NVIDIA GeForce RTX 4090.

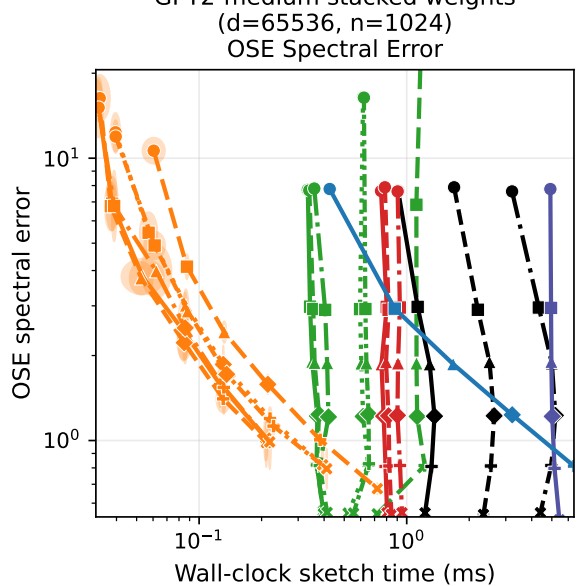

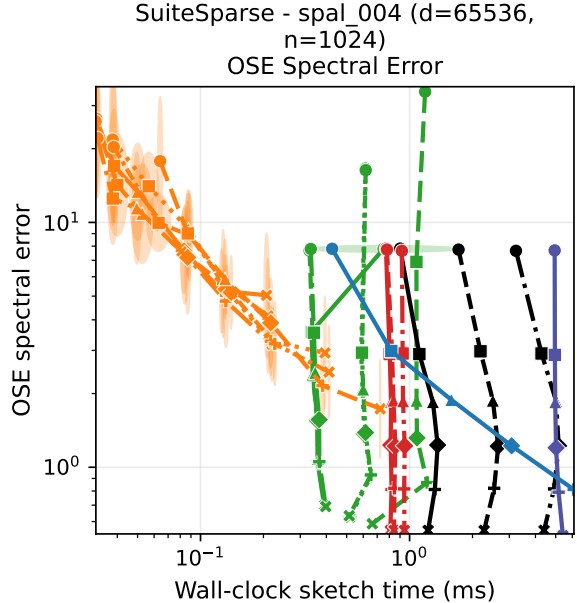

*Figure 28.* **OSE spectral-error ablations.** GPT2-medium stacked weights (d=65536, n=1024). GPU: NVIDIA GeForce RTX 4090.

*Figure 30.* **OSE spectral-error ablations.** SuiteSparse - spal_004 (d=65536, n=1024). GPU: NVIDIA GeForce RTX 4090.

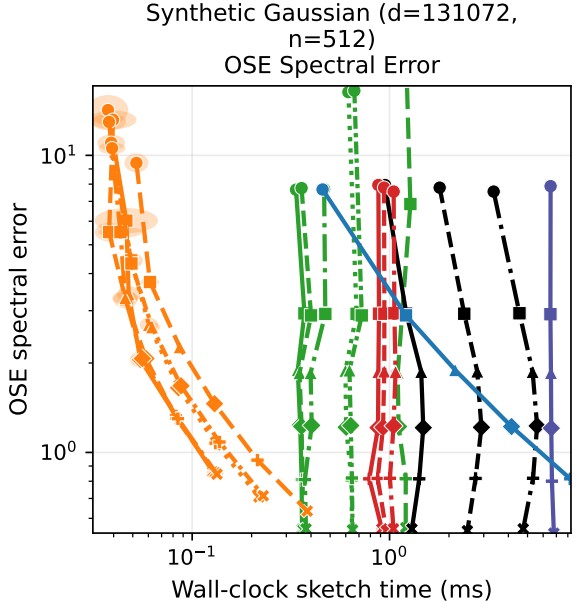

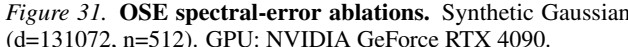

*Figure 31.* **OSE spectral-error ablations.** Synthetic Gaussian (d=131072, n=512). GPU: NVIDIA GeForce RTX 4090.

*Figure 33.* **OSE spectral-error ablations.** Synthetic Low-Rank + noise (d=131072, n=512). GPU: NVIDIA GeForce RTX 4090.

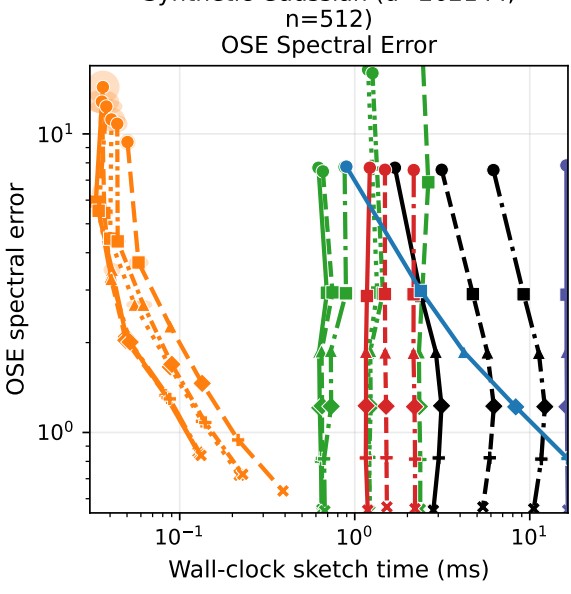

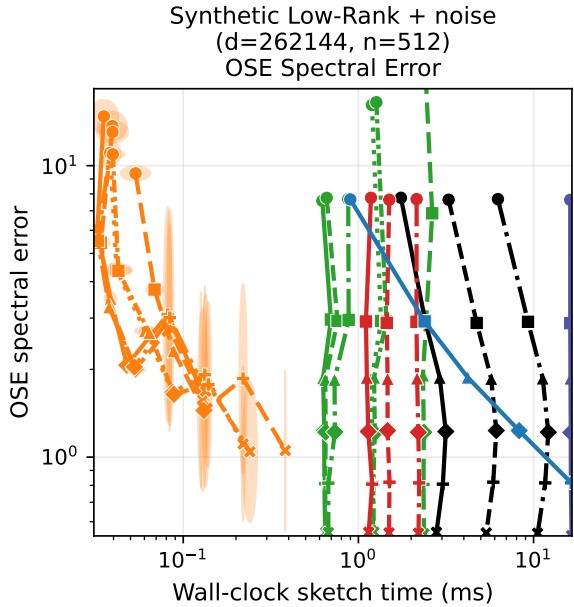

*Figure 32.* **OSE spectral-error ablations.** Synthetic Gaussian (d=262144, n=512). GPU: NVIDIA GeForce RTX 4090.

*Figure 34.* **OSE spectral-error ablations.** Synthetic Low-Rank + noise (d=262144, n=512). GPU: NVIDIA GeForce RTX 4090.

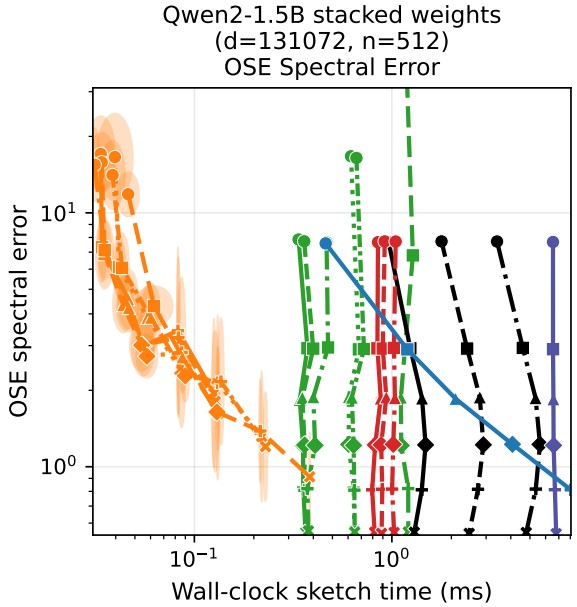

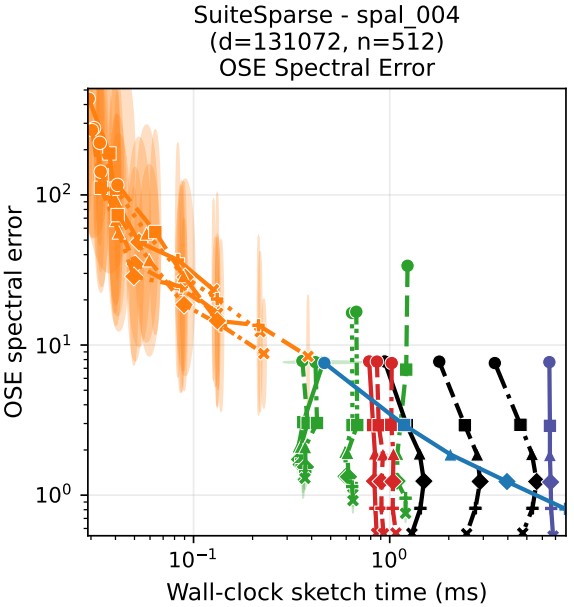

*Figure 35.* **OSE spectral-error ablations.** Qwen2-1.5B stacked weights (d=131072, n=512). GPU: NVIDIA GeForce RTX 4090.

*Figure 37.* **OSE spectral-error ablations.** SuiteSparse - spal_004 (d=131072, n=512). GPU: NVIDIA GeForce RTX 4090.

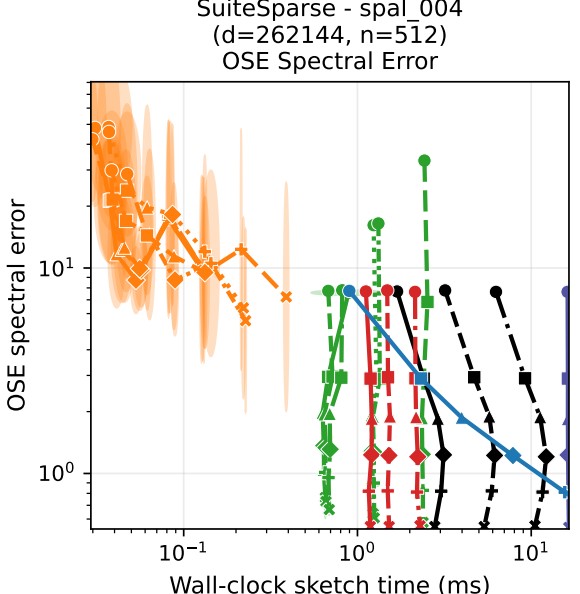

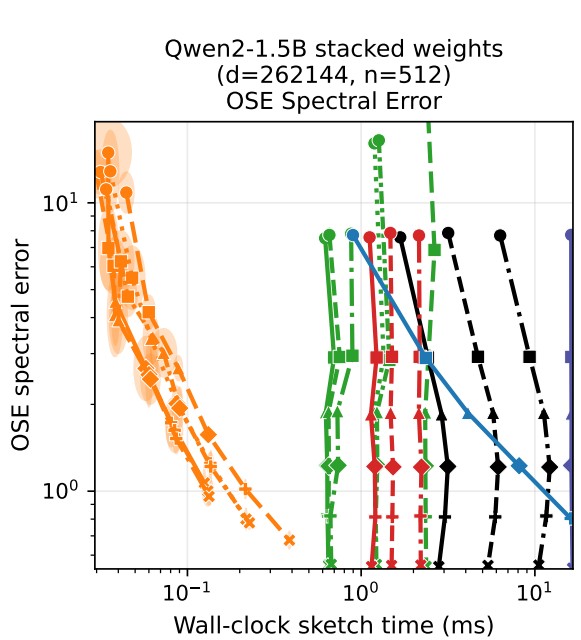

*Figure 38.* **OSE spectral-error ablations.** SuiteSparse - spal_004 (d=262144, n=512). GPU: NVIDIA GeForce RTX 4090.

*Figure 36.* **OSE spectral-error ablations.** Qwen2-1.5B stacked weights (d=262144, n=512). GPU: NVIDIA GeForce RTX 4090.

## F.4. Sketch-and-Ridge Regression Ablations

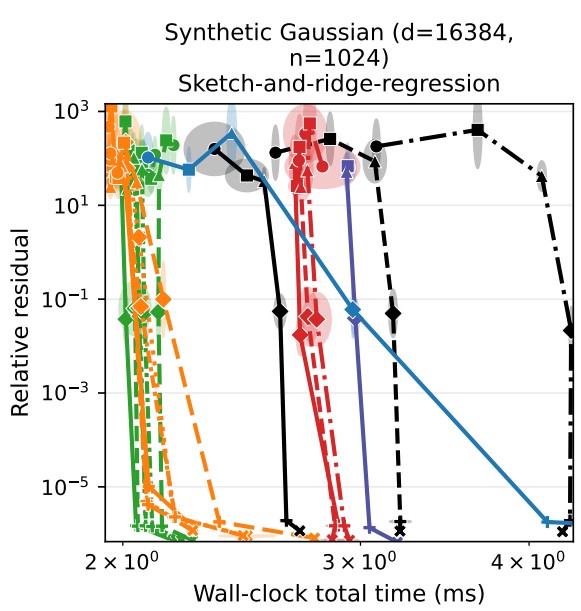

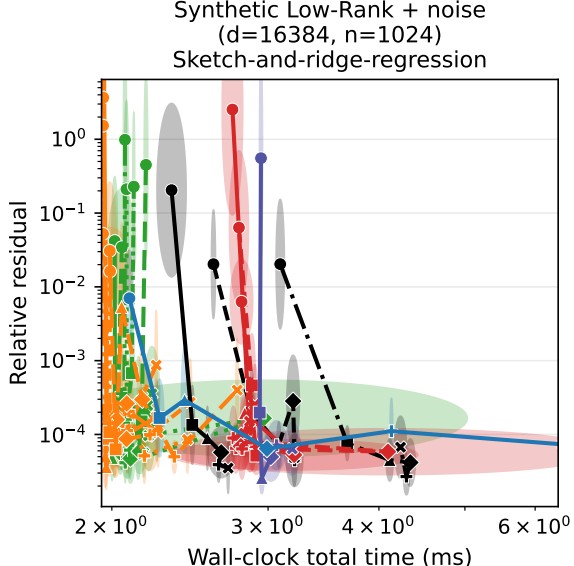

*Figure 41.* **Sketch-and-ridge regression ablations.** Synthetic Low-Rank + noise (d=16384, n=1024). GPU: NVIDIA GeForce RTX 4090.

*Figure 39.* **Sketch-and-ridge regression ablations.** Synthetic Gaussian (d=16384, n=1024). GPU: NVIDIA GeForce RTX 4090.

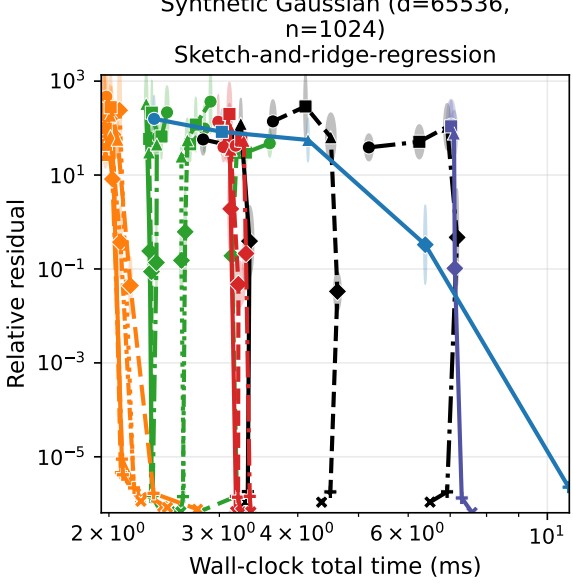

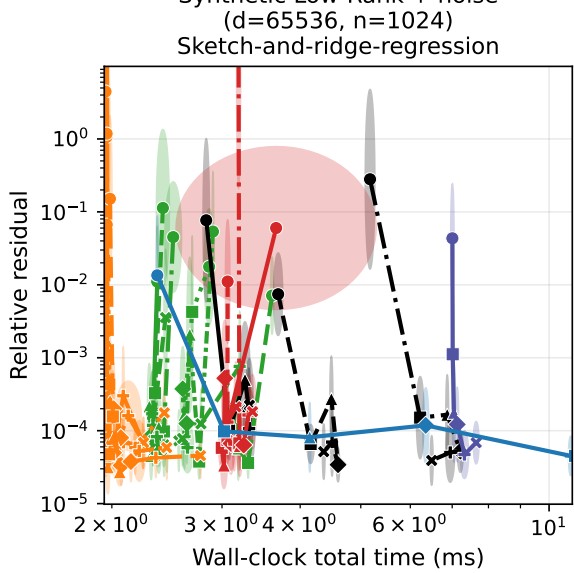

*Figure 40.* **Sketch-and-ridge regression ablations.** Synthetic Gaussian (d=65536, n=1024). GPU: NVIDIA GeForce RTX 4090.

*Figure 42.* **Sketch-and-ridge regression ablations.** Synthetic Low-Rank + noise (d=65536, n=1024). GPU: NVIDIA GeForce RTX 4090.

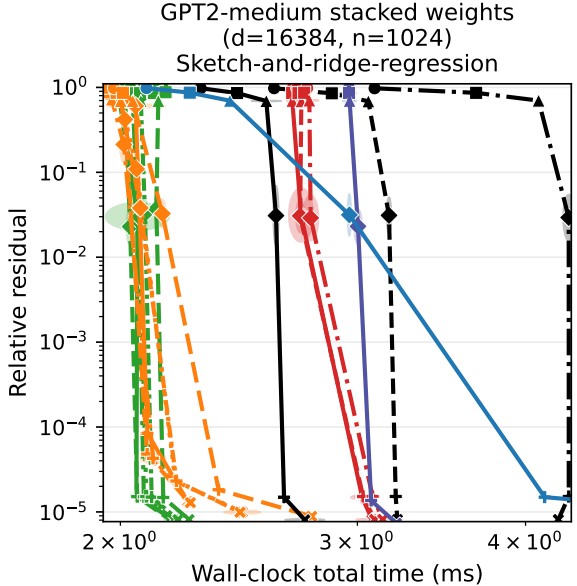

*Figure 43.* **Sketch-and-ridge regression ablations.** GPT2-medium stacked weights (d=16384, n=1024). GPU: NVIDIA GeForce RTX 4090.

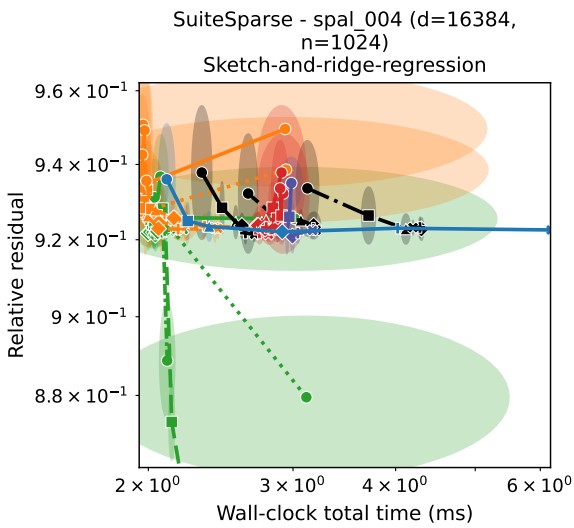

*Figure 45.* **Sketch-and-ridge regression ablations.** SuiteSparse - spal_004 (d=16384, n=1024). GPU: NVIDIA GeForce RTX 4090.

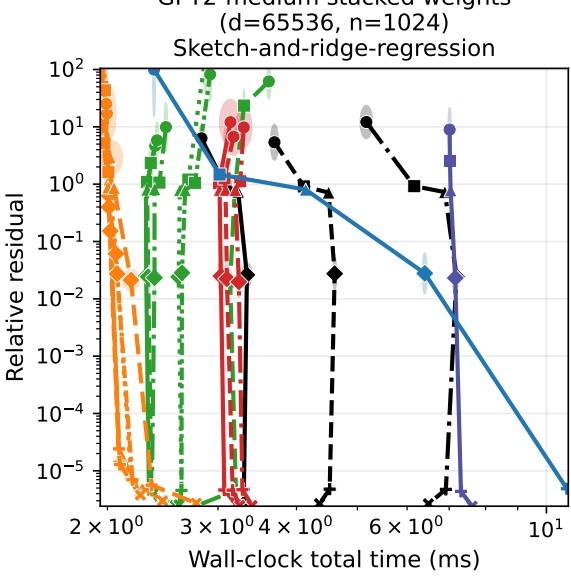

*Figure 44.* **Sketch-and-ridge regression ablations.** GPT2-medium stacked weights (d=65536, n=1024). GPU: NVIDIA GeForce RTX 4090.

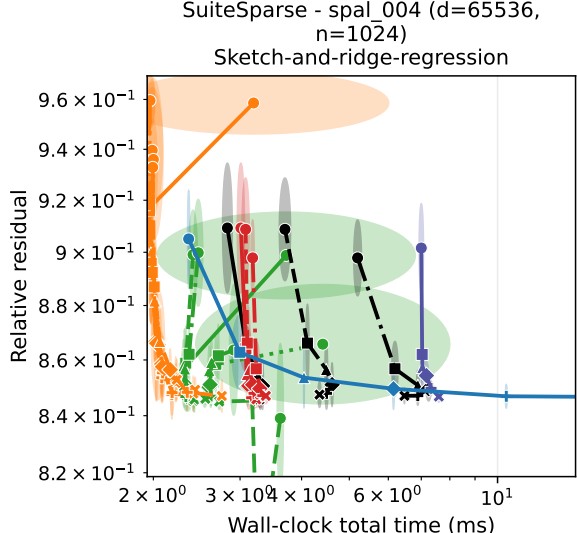

*Figure 46.* **Sketch-and-ridge regression ablations.** SuiteSparse - spal_004 (d=65536, n=1024). GPU: NVIDIA GeForce RTX 4090.

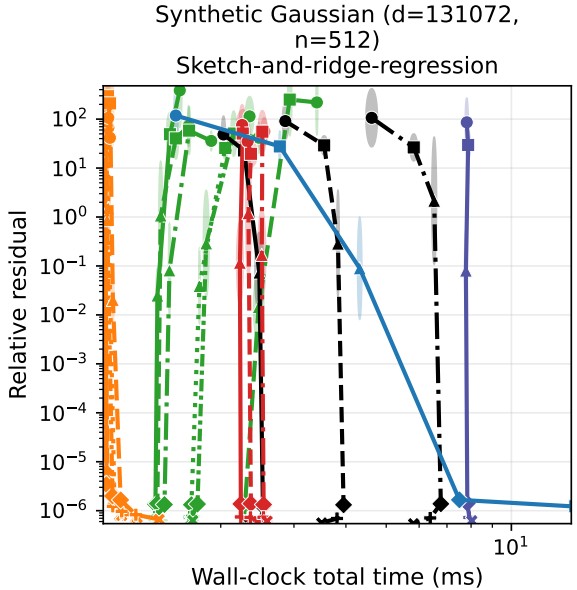

Figure 47. **Sketch-and-ridge regression ablations.** Synthetic Gaussian (d=131072, n=512). GPU: NVIDIA GeForce RTX 4090.

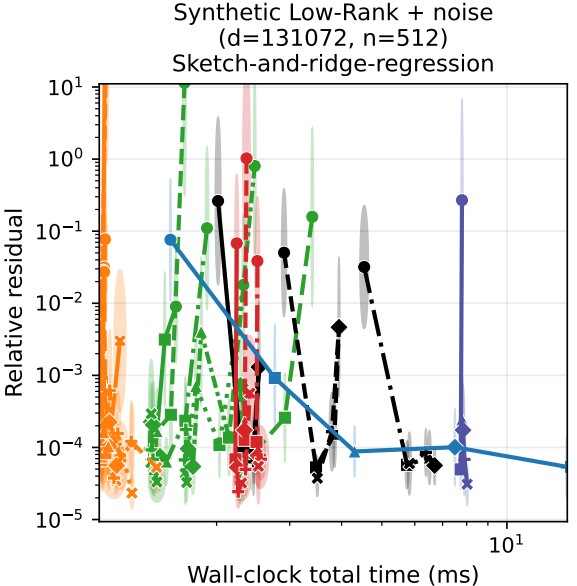

Figure 49. **Sketch-and-ridge regression ablations.** Synthetic Low-Rank + noise (d=131072, n=512). GPU: NVIDIA GeForce RTX 4090.

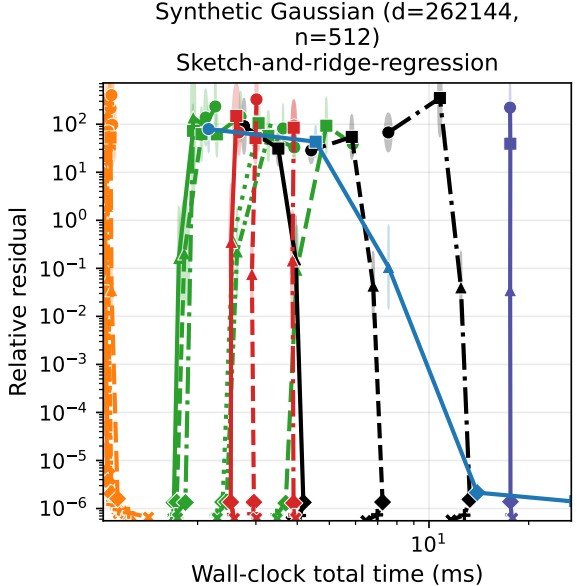

Figure 48. **Sketch-and-ridge regression ablations.** Synthetic Gaussian (d=262144, n=512). GPU: NVIDIA GeForce RTX 4090.

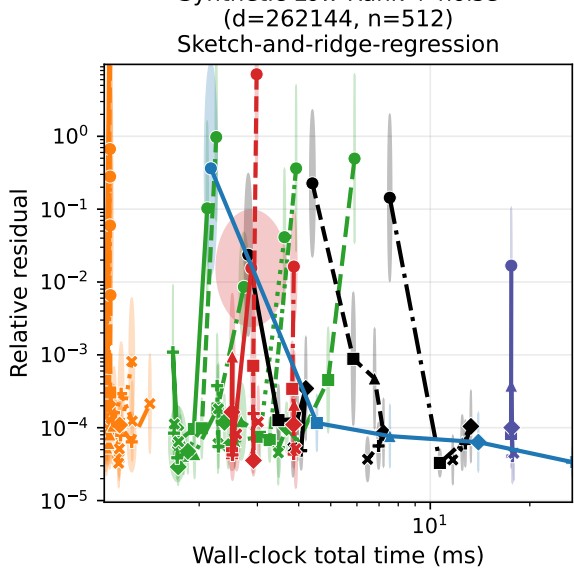

Figure 50. **Sketch-and-ridge regression ablations.** Synthetic Low-Rank + noise (d=262144, n=512). GPU: NVIDIA GeForce RTX 4090.

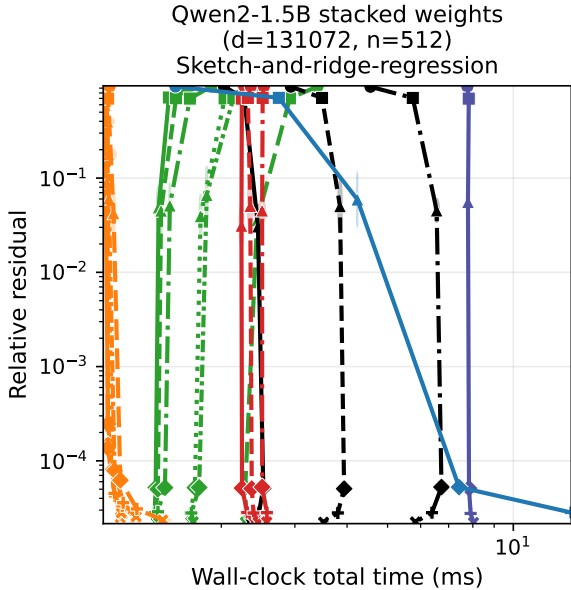

*Figure 51.* **Sketch-and-ridge regression ablations.** Qwen2-1.5B stacked weights (d=131072, n=512). GPU: NVIDIA GeForce RTX 4090.

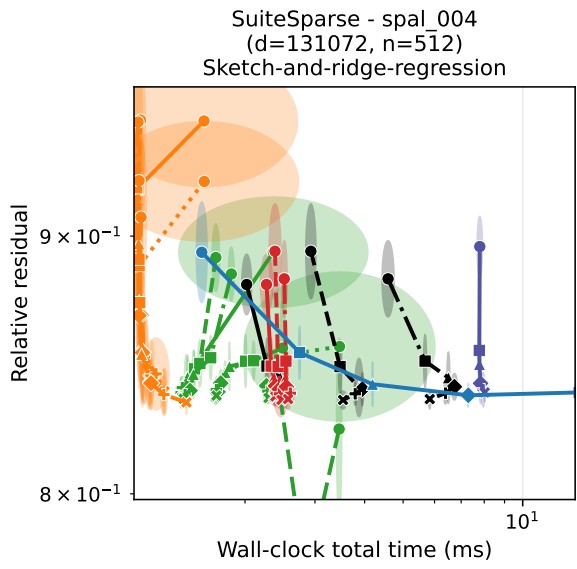

*Figure 53.* **Sketch-and-ridge regression ablations.** SuiteSparse - spal_004 (d=131072, n=512). GPU: NVIDIA GeForce RTX 4090.

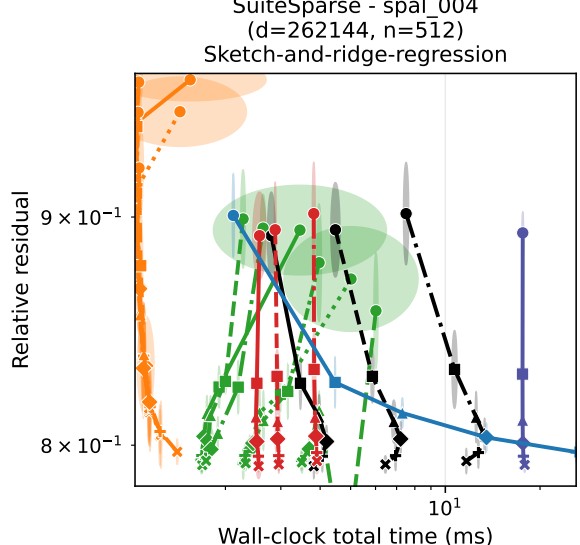

*Figure 54.* **Sketch-and-ridge regression ablations.** SuiteSparse - spal_004 (d=262144, n=512). GPU: NVIDIA GeForce RTX 4090.

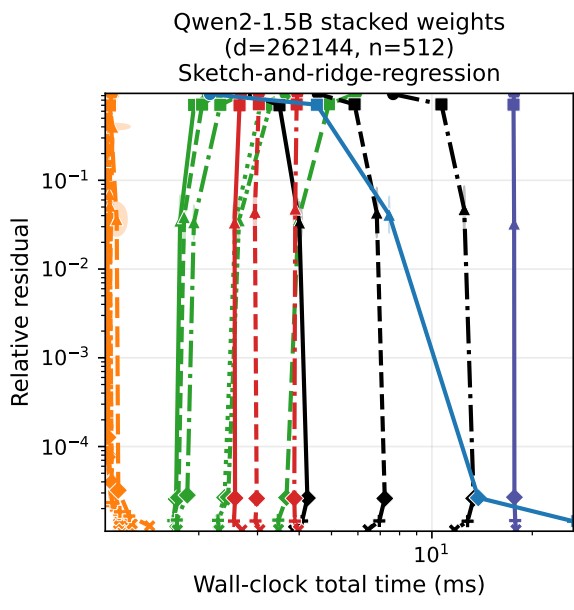

*Figure 52.* **Sketch-and-ridge regression ablations.** Qwen2-1.5B stacked weights (d=262144, n=512). GPU: NVIDIA GeForce RTX 4090.

## F.5. Sketch-and-Solve Ablations

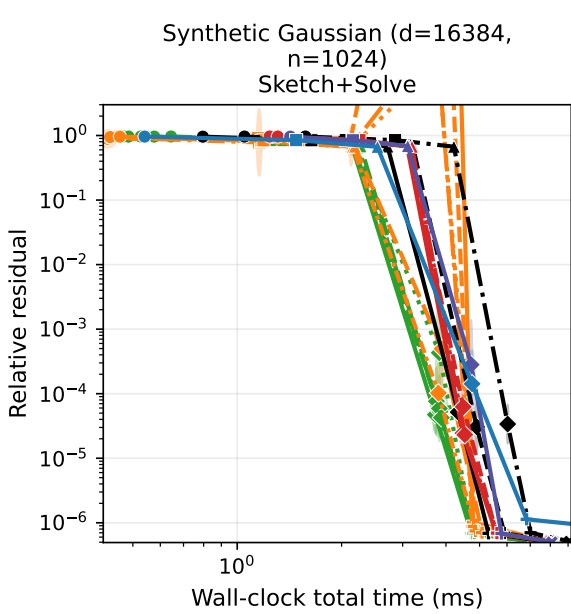

*Figure 55.* **Sketch-and-solve ablations.** Synthetic Gaussian (d=16384, n=1024). GPU: NVIDIA GeForce RTX 4090.

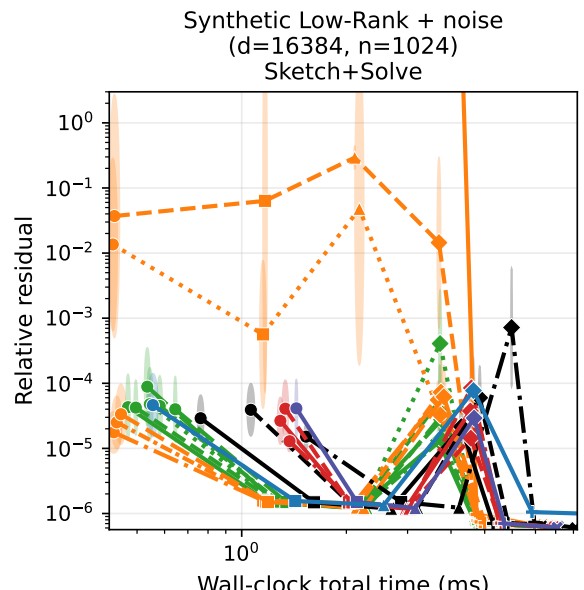

*Figure 57.* **Sketch-and-solve ablations.** Synthetic Low-Rank + noise (d=16384, n=1024). GPU: NVIDIA GeForce RTX 4090.

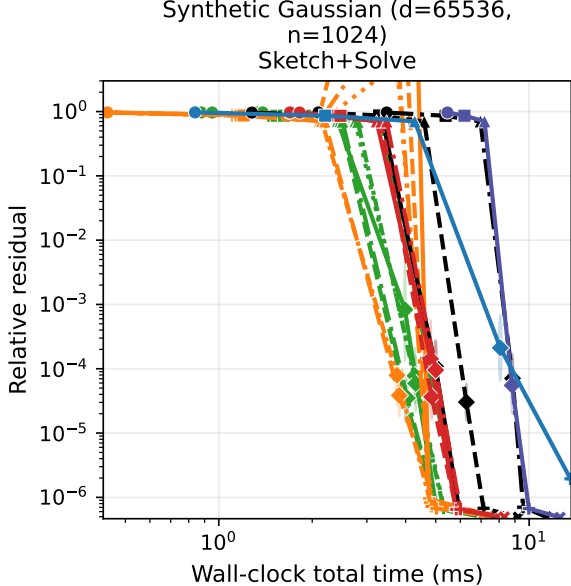

*Figure 56.* **Sketch-and-solve ablations.** Synthetic Gaussian (d=65536, n=1024). GPU: NVIDIA GeForce RTX 4090.

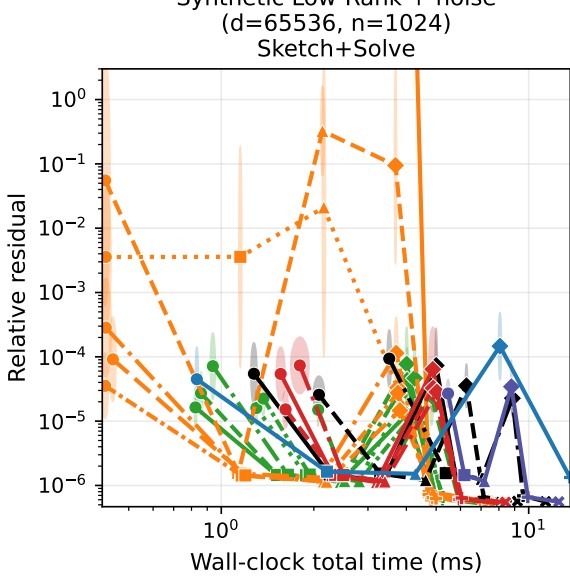

*Figure 58.* **Sketch-and-solve ablations.** Synthetic Low-Rank + noise (d=65536, n=1024). GPU: NVIDIA GeForce RTX 4090.

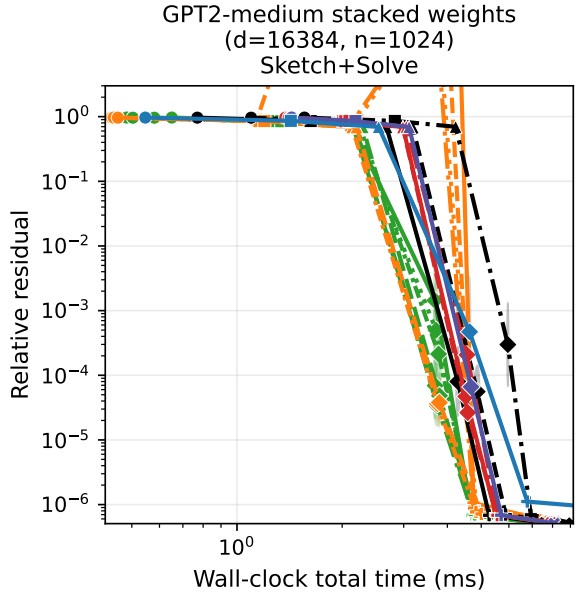

*Figure 59.* **Sketch-and-solve ablations.** GPT2-medium stacked weights (d=16384, n=1024). GPU: NVIDIA GeForce RTX 4090.

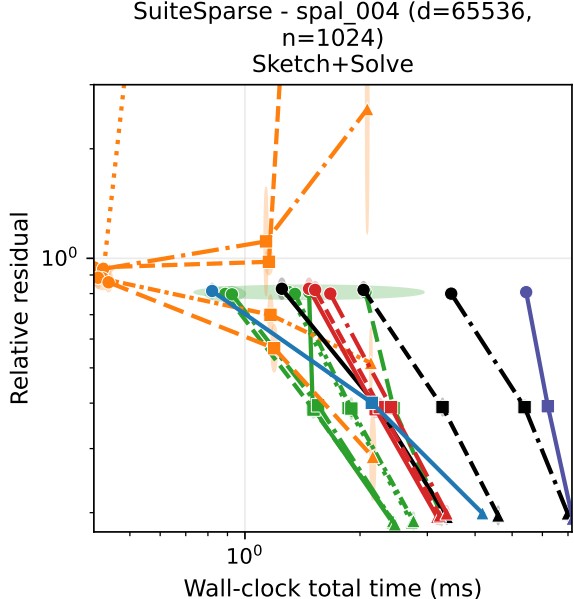

*Figure 61.* **Sketch-and-solve ablations.** SuiteSparse - spal_004 (d=16384, n=1024). GPU: NVIDIA GeForce RTX 4090.

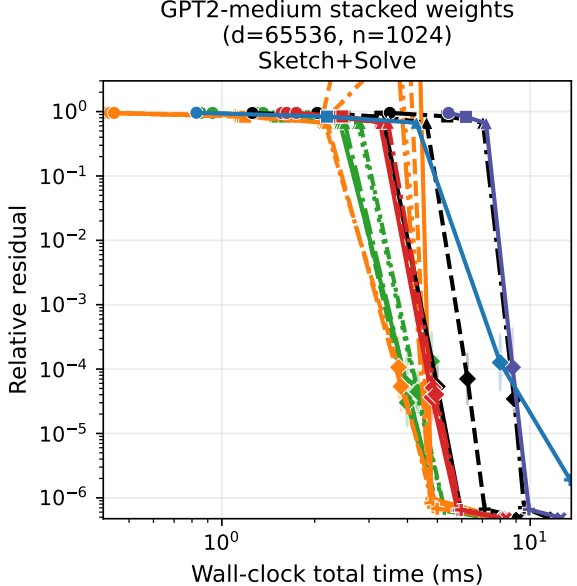

*Figure 60.* **Sketch-and-solve ablations.** GPT2-medium stacked weights (d=65536, n=1024). GPU: NVIDIA GeForce RTX 4090.

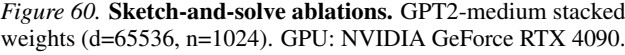

*Figure 62.* **Sketch-and-solve ablations.** SuiteSparse - spal_004 (d=65536, n=1024). GPU: NVIDIA GeForce RTX 4090.

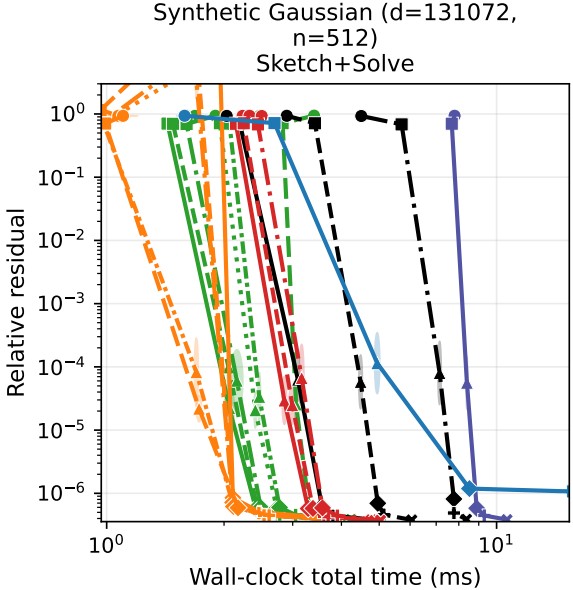

*Figure 63.* **Sketch-and-solve ablations.** Synthetic Gaussian (d=131072, n=512). GPU: NVIDIA GeForce RTX 4090.

*Figure 65.* **Sketch-and-solve ablations.** Synthetic Low-Rank + noise (d=131072, n=512). GPU: NVIDIA GeForce RTX 4090.

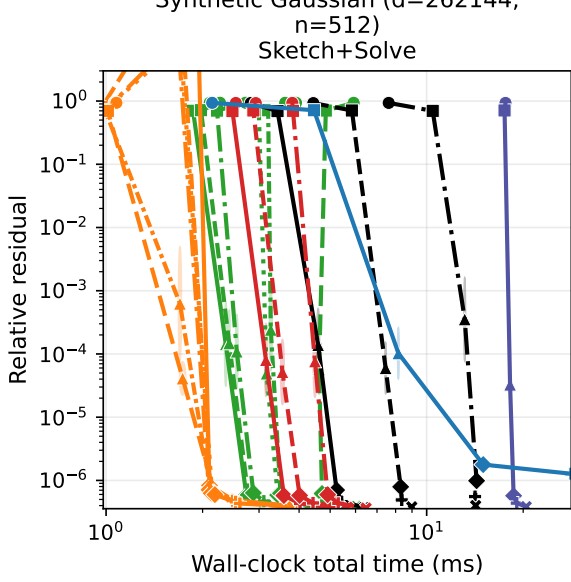

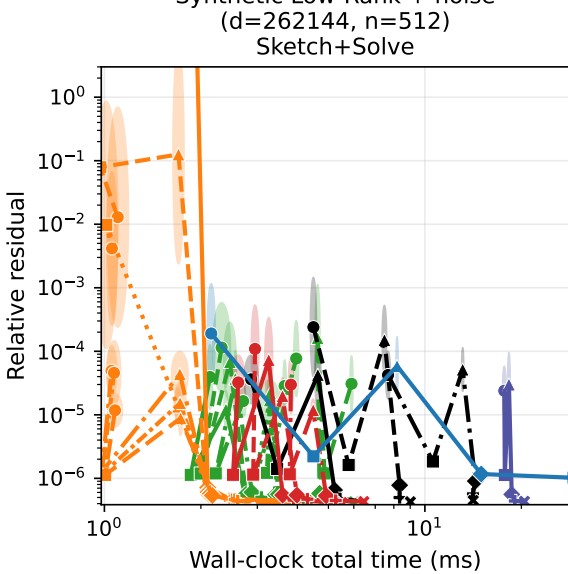

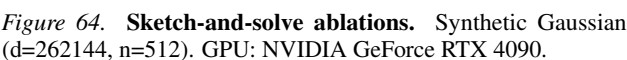

*Figure 64.* **Sketch-and-solve ablations.** Synthetic Gaussian (d=262144, n=512). GPU: NVIDIA GeForce RTX 4090.

*Figure 66.* **Sketch-and-solve ablations.** Synthetic Low-Rank + noise (d=262144, n=512). GPU: NVIDIA GeForce RTX 4090.

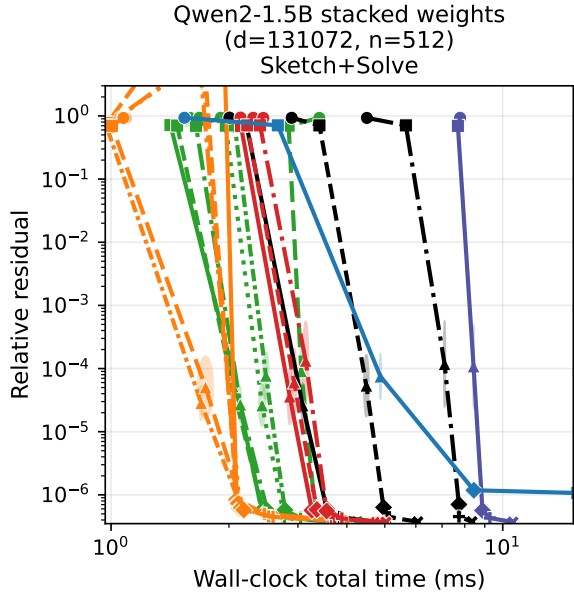

*Figure 67.* **Sketch-and-solve ablations.** Qwen2-1.5B stacked weights (d=131072, n=512). GPU: NVIDIA GeForce RTX 4090.

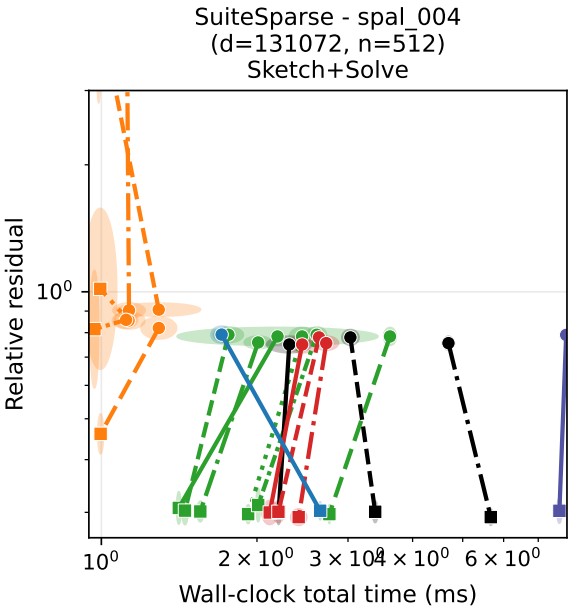

*Figure 69.* **Sketch-and-solve ablations.** SuiteSparse - spal_004 (d=131072, n=512). GPU: NVIDIA GeForce RTX 4090.

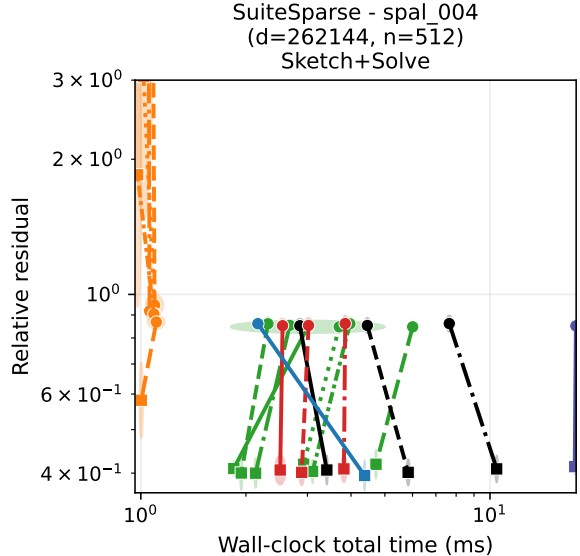

*Figure 70.* **Sketch-and-solve ablations.** SuiteSparse - spal_004 (d=262144, n=512). GPU: NVIDIA GeForce RTX 4090.

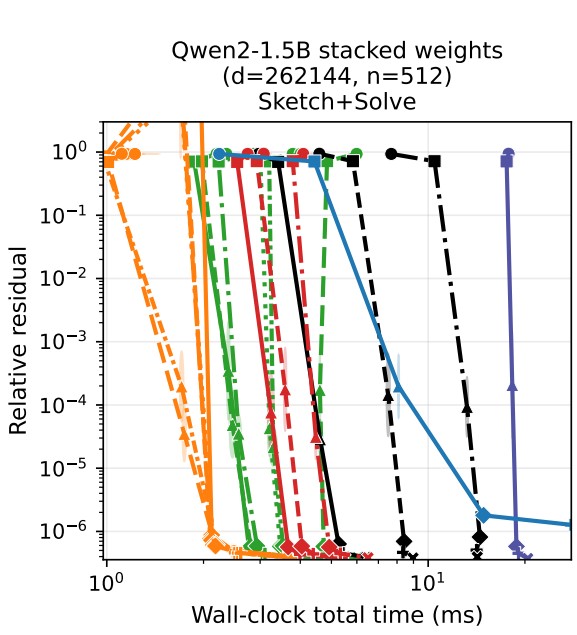

*Figure 68.* **Sketch-and-solve ablations.** Qwen2-1.5B stacked weights (d=262144, n=512). GPU: NVIDIA GeForce RTX 4090.

# G. Additional Experiments and Ablations (RTX A6000)

## G.1. Gram-Matrix Approximation Ablations

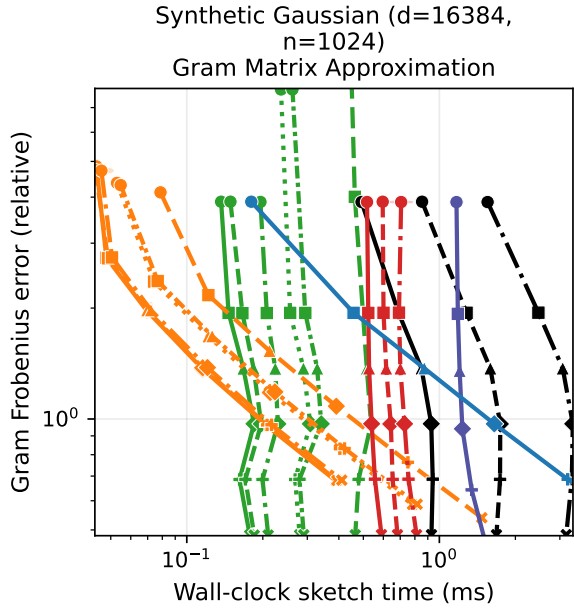

*Figure 71.* **Gram-matrix approximation ablations.** Synthetic Gaussian (d=16384, n=1024). GPU: NVIDIA RTX A6000.

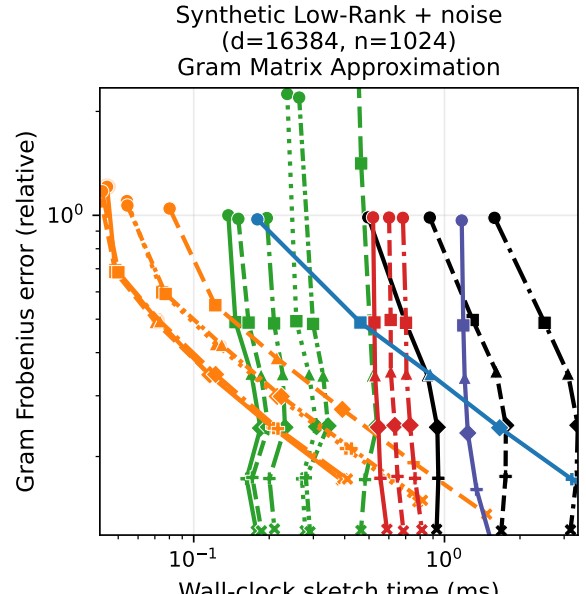

*Figure 73.* **Gram-matrix approximation ablations.** Synthetic Low-Rank + noise (d=16384, n=1024). GPU: NVIDIA RTX A6000.

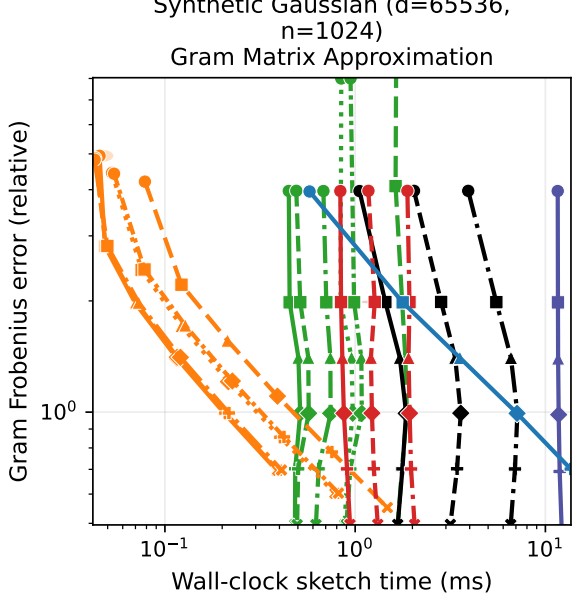

*Figure 72.* **Gram-matrix approximation ablations.** Synthetic Gaussian (d=65536, n=1024). GPU: NVIDIA RTX A6000.

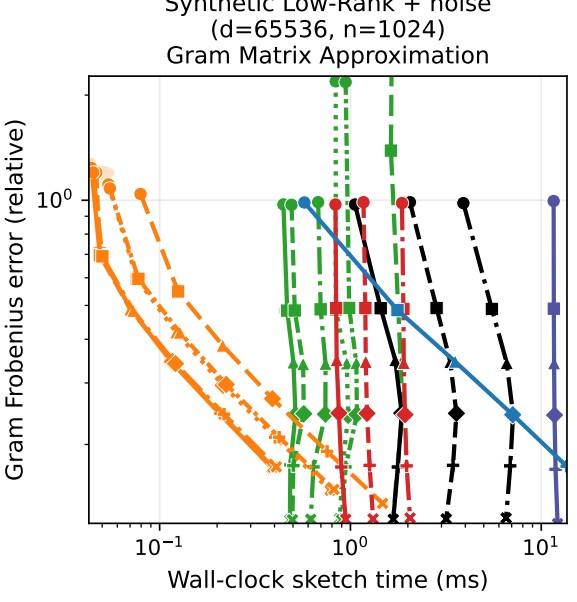

*Figure 74.* **Gram-matrix approximation ablations.** Synthetic Low-Rank + noise (d=65536, n=1024). GPU: NVIDIA RTX A6000.

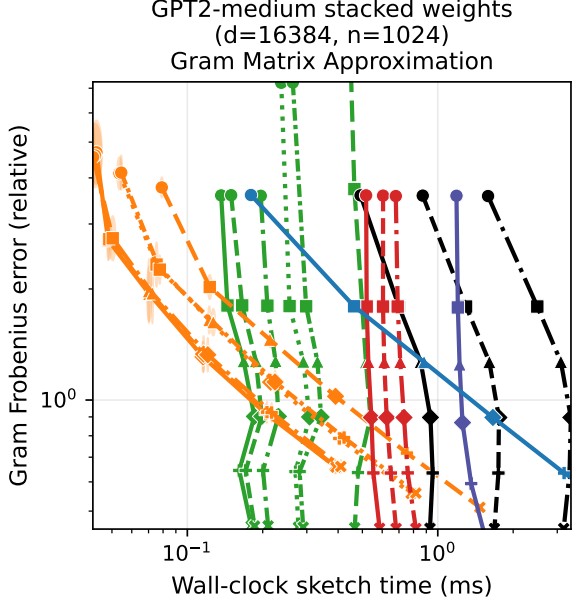

*Figure 75.* **Gram-matrix approximation ablations.** GPT2-medium stacked weights (d=16384, n=1024). GPU: NVIDIA RTX A6000.

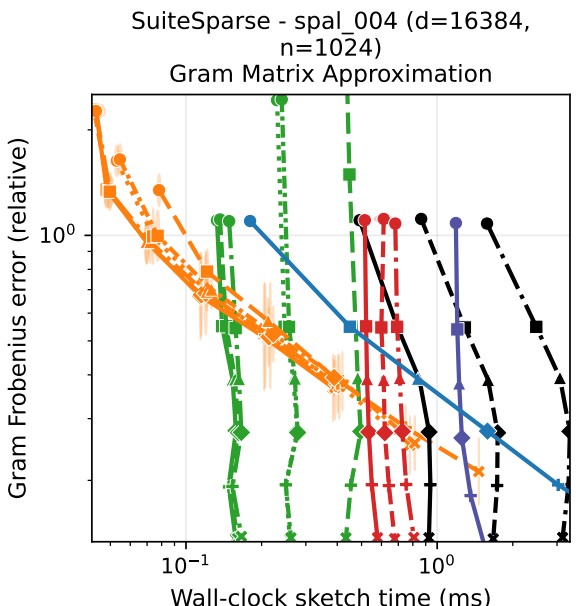

*Figure 77.* **Gram-matrix approximation ablations.** SuiteSparse - spal_004 (d=16384, n=1024). GPU: NVIDIA RTX A6000.

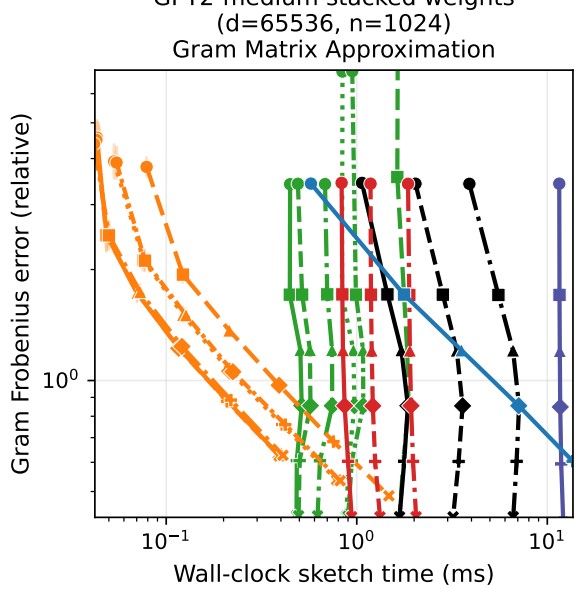

*Figure 76.* **Gram-matrix approximation ablations.** GPT2-medium stacked weights (d=65536, n=1024). GPU: NVIDIA RTX A6000.

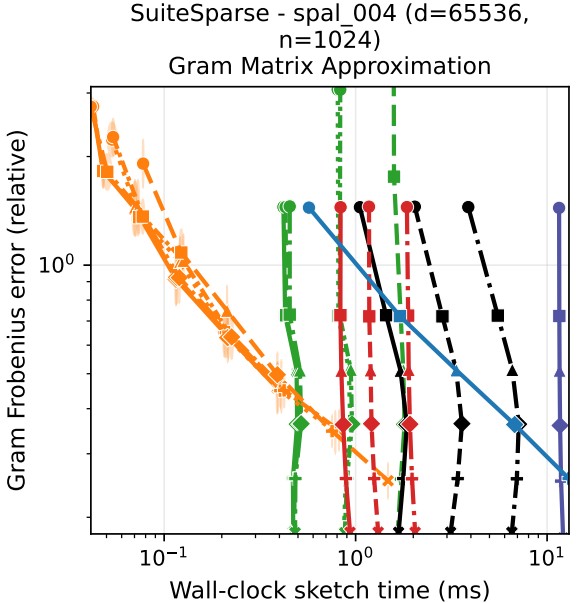

*Figure 78.* **Gram-matrix approximation ablations.** SuiteSparse - spal_004 (d=65536, n=1024). GPU: NVIDIA RTX A6000.

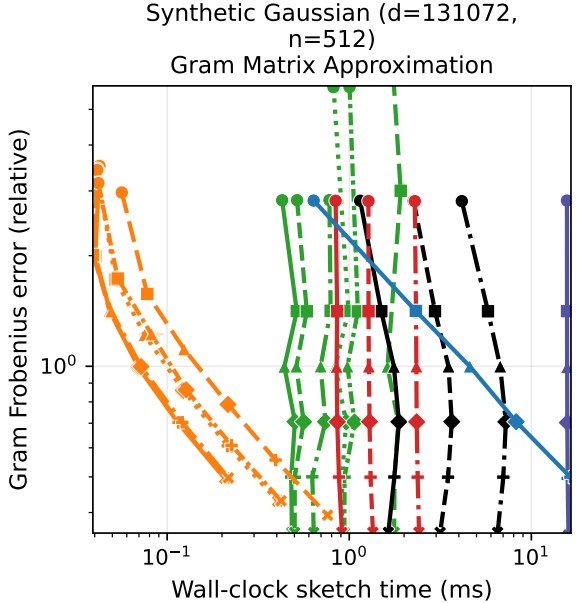

*Figure 79.* **Gram-matrix approximation ablations.** Synthetic Gaussian (d=131072, n=512). GPU: NVIDIA RTX A6000.

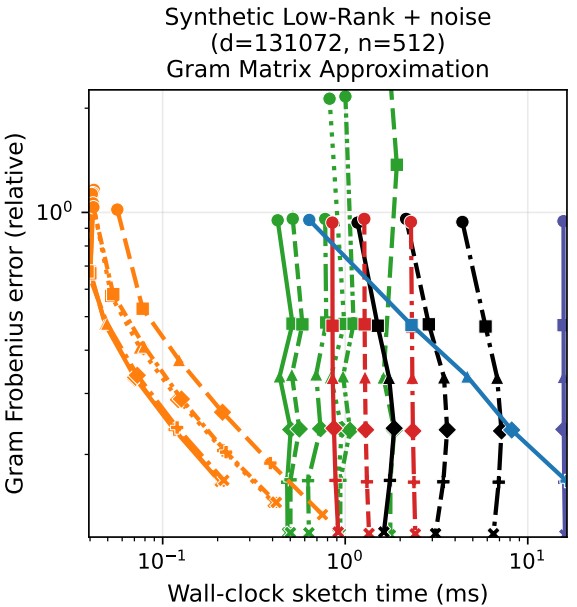

*Figure 81.* **Gram-matrix approximation ablations.** Synthetic Low-Rank + noise (d=131072, n=512). GPU: NVIDIA RTX A6000.

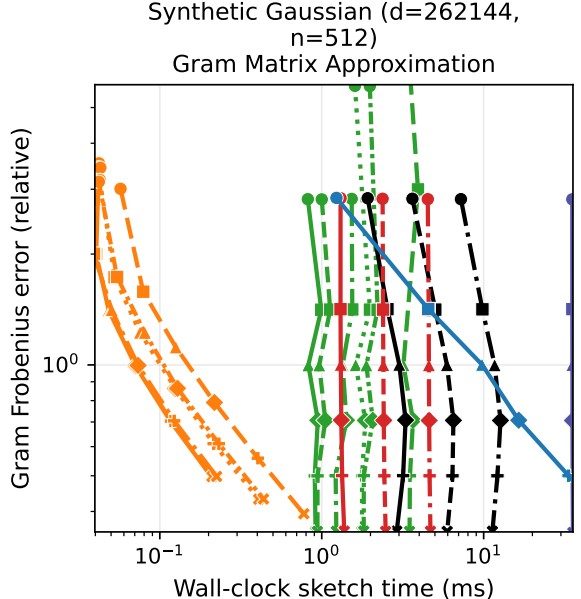

*Figure 80.* **Gram-matrix approximation ablations.** Synthetic Gaussian (d=262144, n=512). GPU: NVIDIA RTX A6000.

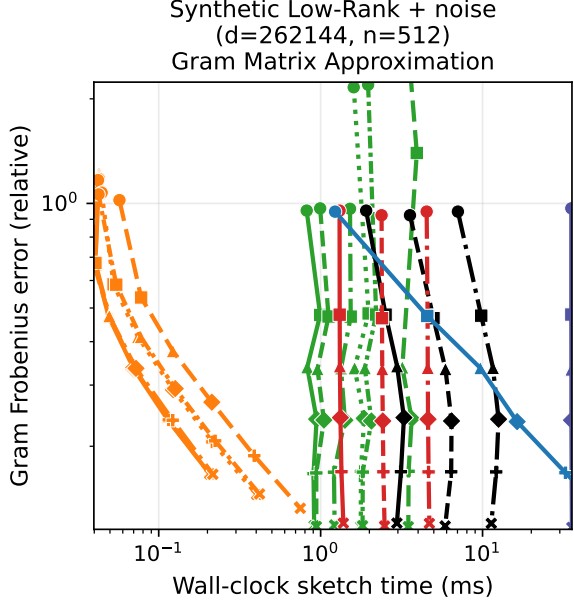

*Figure 82.* **Gram-matrix approximation ablations.** Synthetic Low-Rank + noise (d=262144, n=512). GPU: NVIDIA RTX A6000.

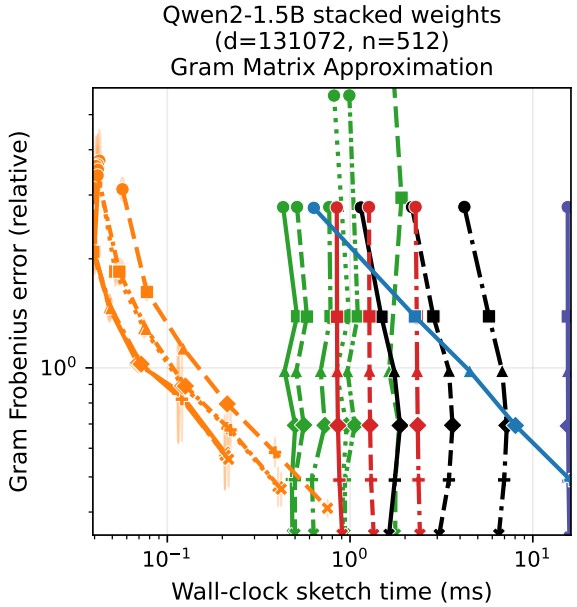

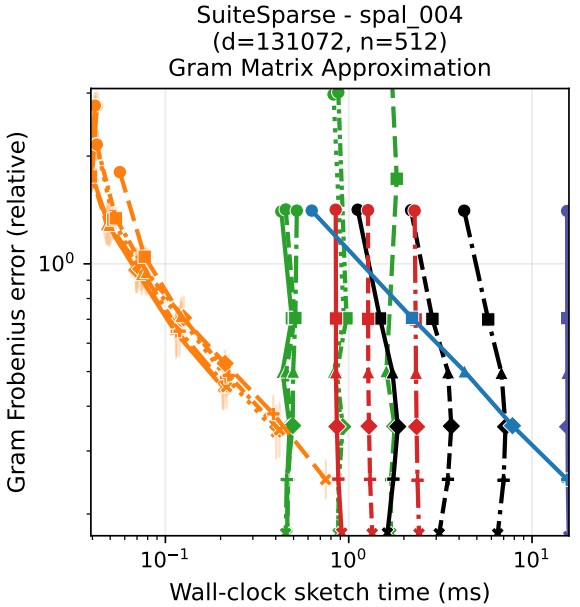

*Figure 83.* **Gram-matrix approximation ablations.** Qwen2-1.5B stacked weights (d=131072, n=512). GPU: NVIDIA RTX A6000.

*Figure 85.* **Gram-matrix approximation ablations.** SuiteSparse - spal_004 (d=131072, n=512). GPU: NVIDIA RTX A6000.

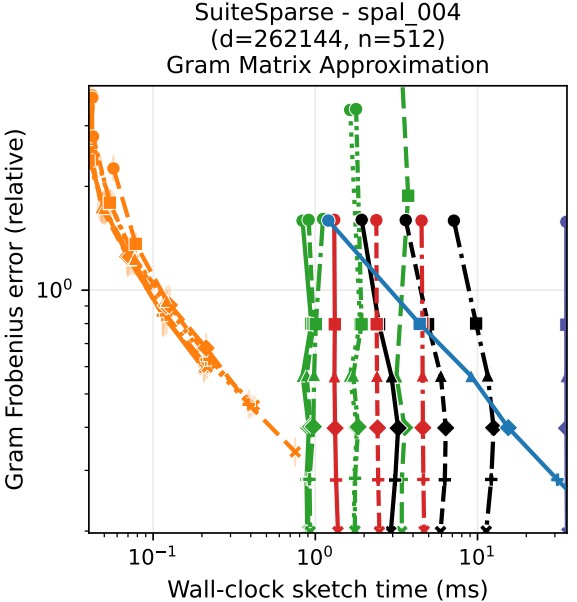

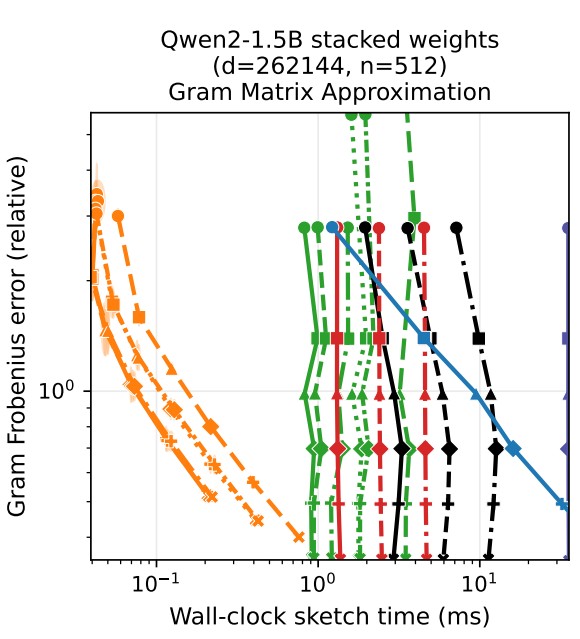

*Figure 86.* **Gram-matrix approximation ablations.** SuiteSparse - spal_004 (d=262144, n=512). GPU: NVIDIA RTX A6000.

*Figure 84.* **Gram-matrix approximation ablations.** Qwen2-1.5B stacked weights (d=262144, n=512). GPU: NVIDIA RTX A6000.

## G.2. OSE Spectral-Error Ablations

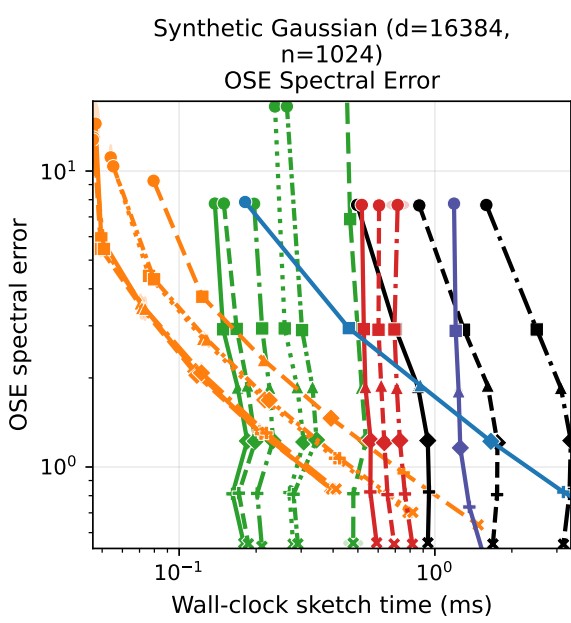

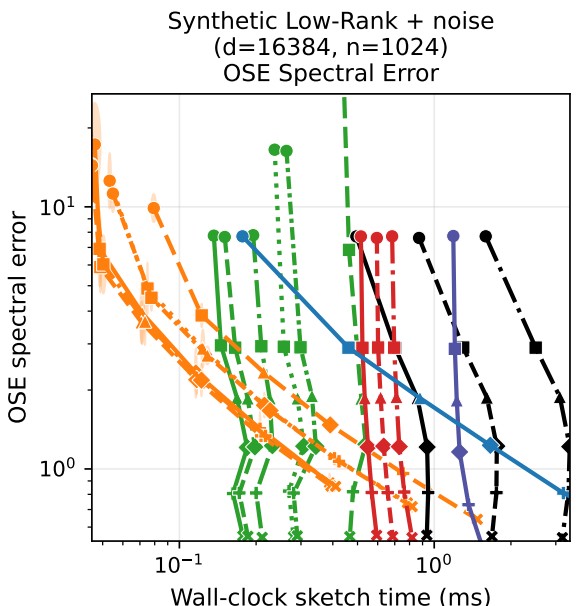

*Figure 89.* **OSE spectral-error ablations.** Synthetic Low-Rank + noise (d=16384, n=1024). GPU: NVIDIA RTX A6000.

*Figure 87.* **OSE spectral-error ablations.** Synthetic Gaussian (d=16384, n=1024). GPU: NVIDIA RTX A6000.

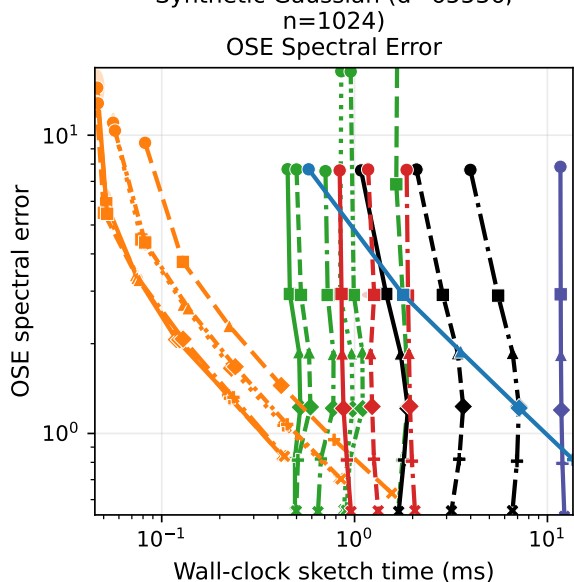

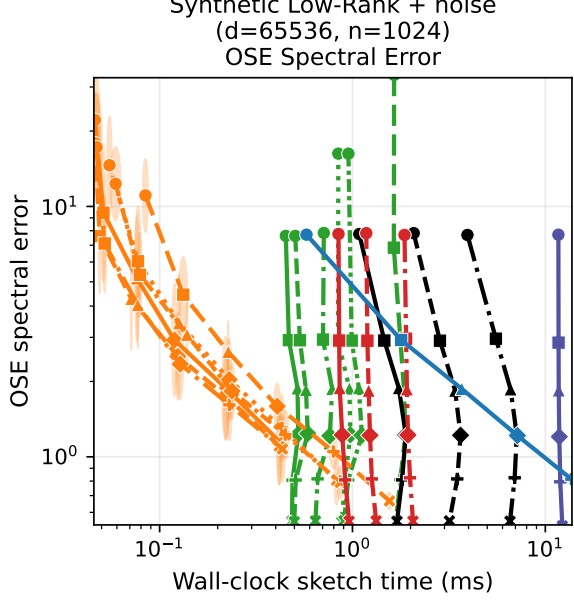

*Figure 88.* **OSE spectral-error ablations.** Synthetic Gaussian (d=65536, n=1024). GPU: NVIDIA RTX A6000.

*Figure 90.* **OSE spectral-error ablations.** Synthetic Low-Rank + noise (d=65536, n=1024). GPU: NVIDIA RTX A6000.

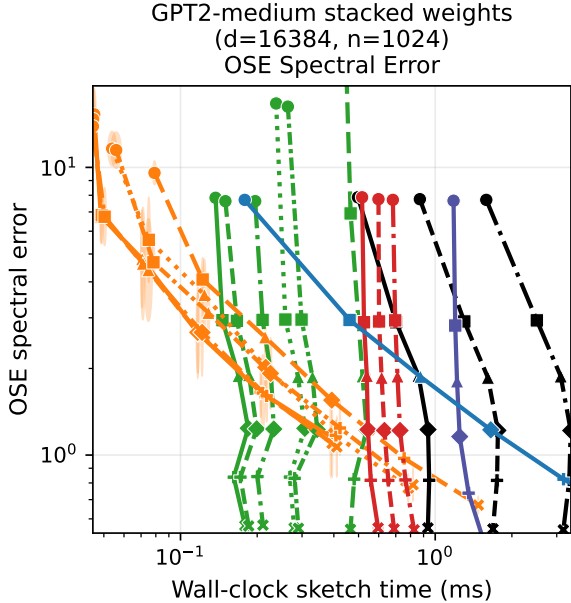

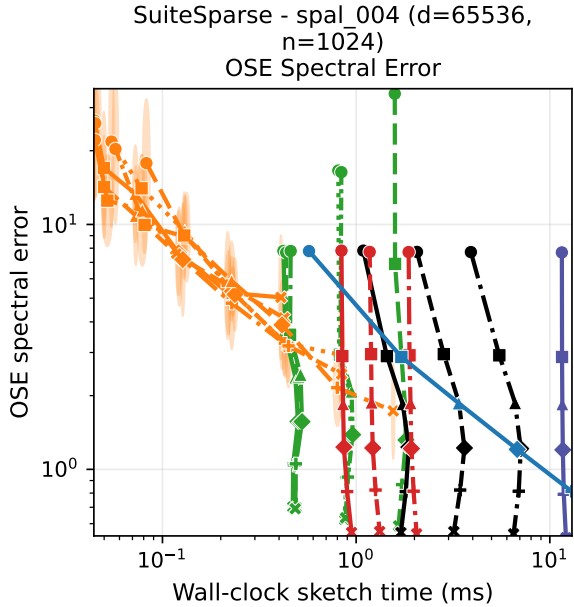

*Figure 91.* **OSE spectral-error ablations.** GPT2-medium stacked weights (d=16384, n=1024). GPU: NVIDIA RTX A6000.

*Figure 93.* **OSE spectral-error ablations.** SuiteSparse - spal_004 (d=16384, n=1024). GPU: NVIDIA RTX A6000.

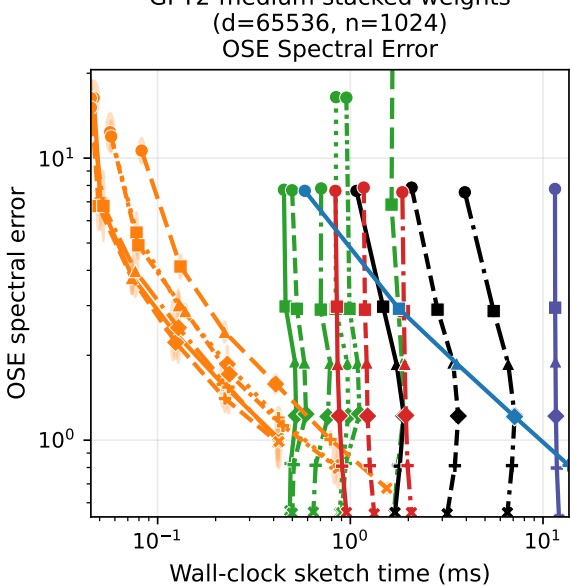

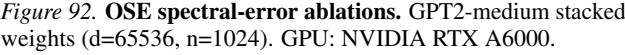

*Figure 92.* **OSE spectral-error ablations.** GPT2-medium stacked weights (d=65536, n=1024). GPU: NVIDIA RTX A6000.

*Figure 94.* **OSE spectral-error ablations.** SuiteSparse - spal_004 (d=65536, n=1024). GPU: NVIDIA RTX A6000.

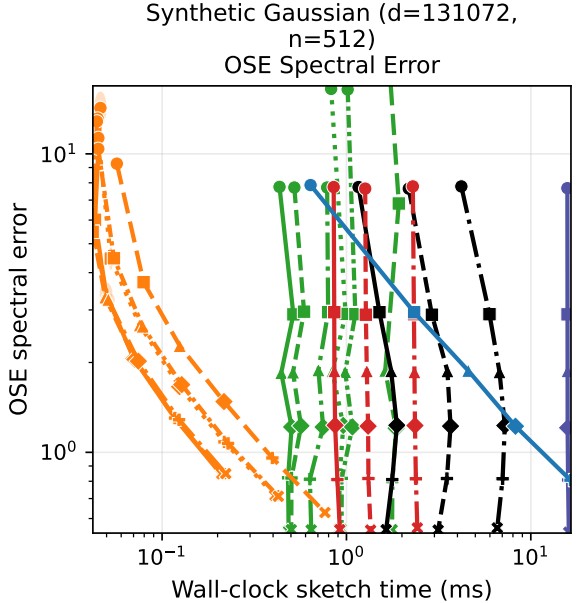

*Figure 95.* **OSE spectral-error ablations.** Synthetic Gaussian (d=131072, n=512). GPU: NVIDIA RTX A6000.

*Figure 97.* **OSE spectral-error ablations.** Synthetic Low-Rank + noise (d=131072, n=512). GPU: NVIDIA RTX A6000.

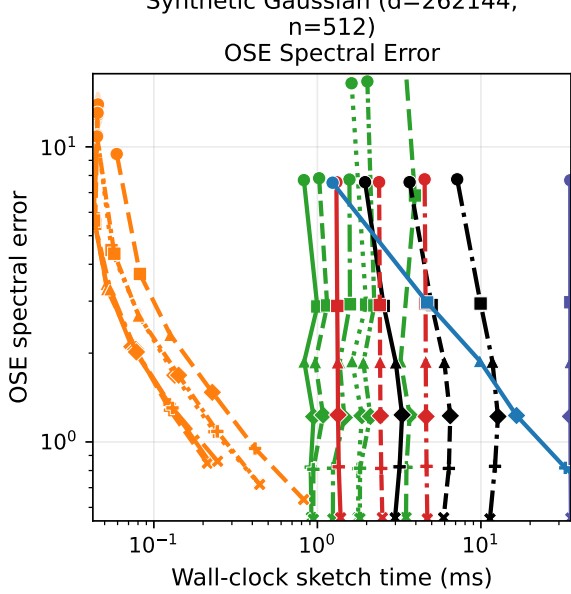

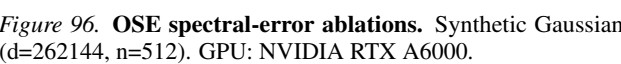

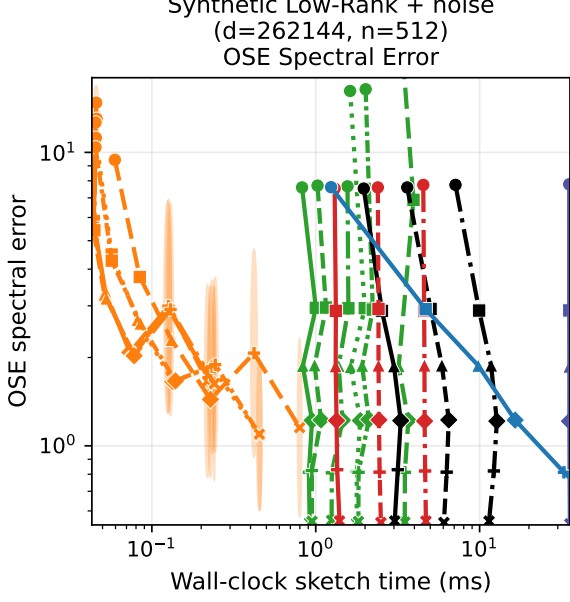

*Figure 96.* **OSE spectral-error ablations.** Synthetic Gaussian (d=262144, n=512). GPU: NVIDIA RTX A6000.

*Figure 98.* **OSE spectral-error ablations.** Synthetic Low-Rank + noise (d=262144, n=512). GPU: NVIDIA RTX A6000.

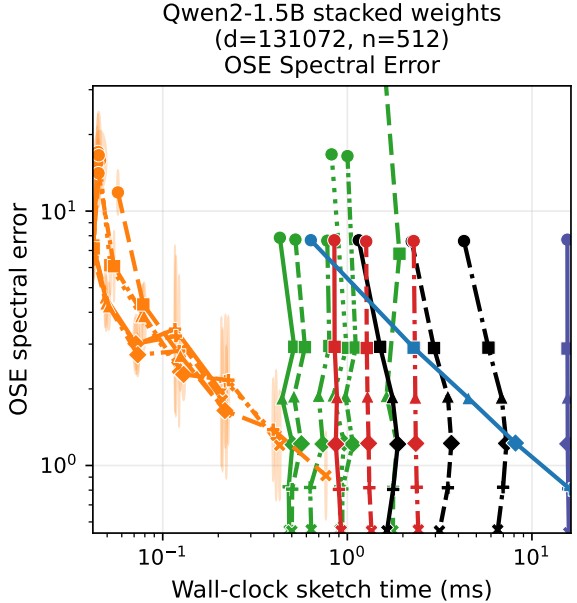

*Figure 99.* **OSE spectral-error ablations.** Qwen2-1.5B stacked weights (d=131072, n=512). GPU: NVIDIA RTX A6000.

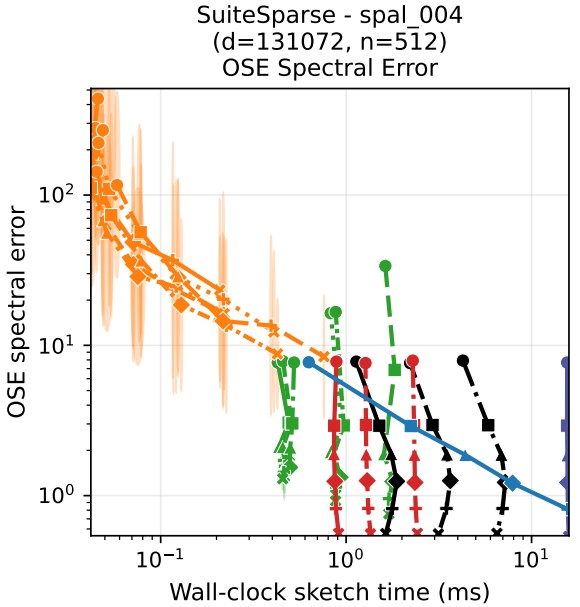

*Figure 101.* **OSE spectral-error ablations.** SuiteSparse - spal_004 (d=131072, n=512). GPU: NVIDIA RTX A6000.

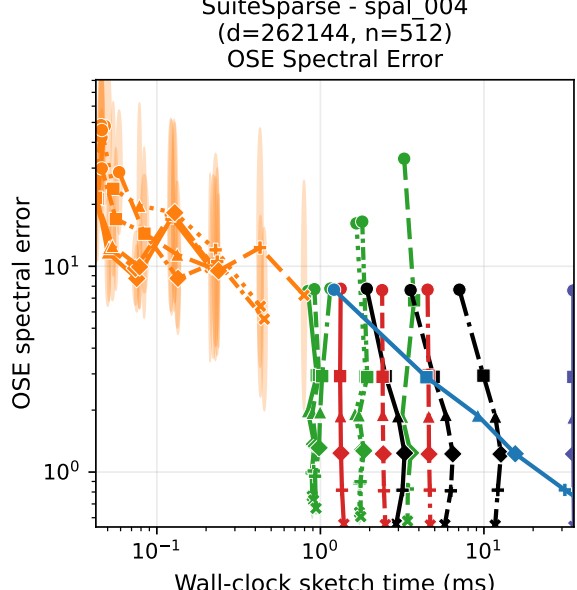

*Figure 102.* **OSE spectral-error ablations.** SuiteSparse - spal_004 (d=262144, n=512). GPU: NVIDIA RTX A6000.

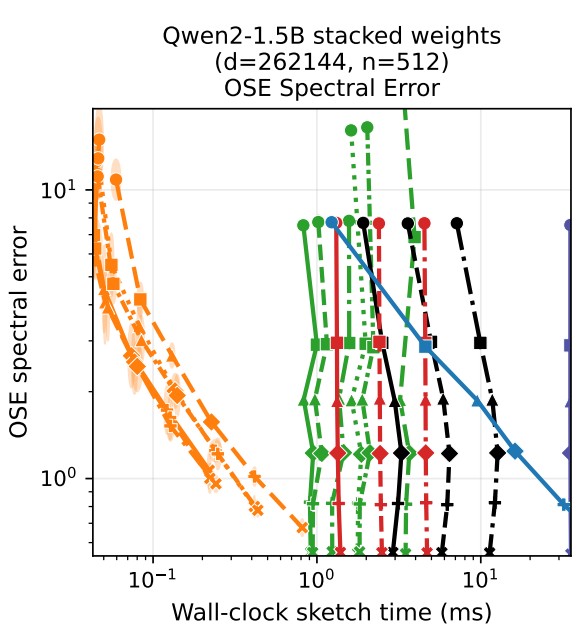

*Figure 100.* **OSE spectral-error ablations.** Qwen2-1.5B stacked weights (d=262144, n=512). GPU: NVIDIA RTX A6000.

## G.3. Sketch-and-Ridge Regression Ablations

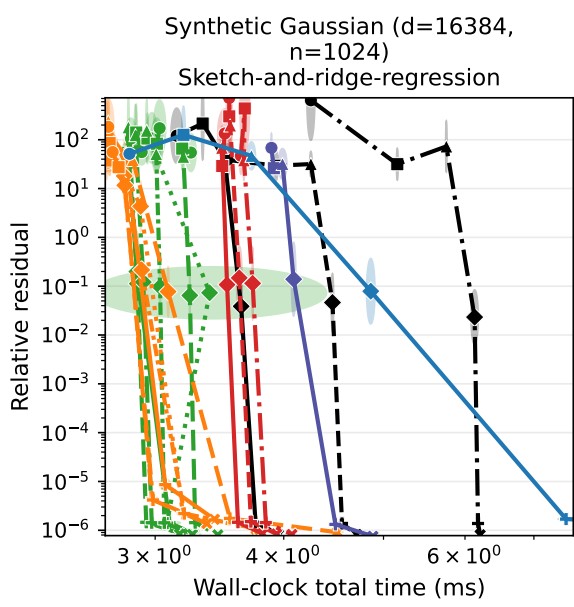

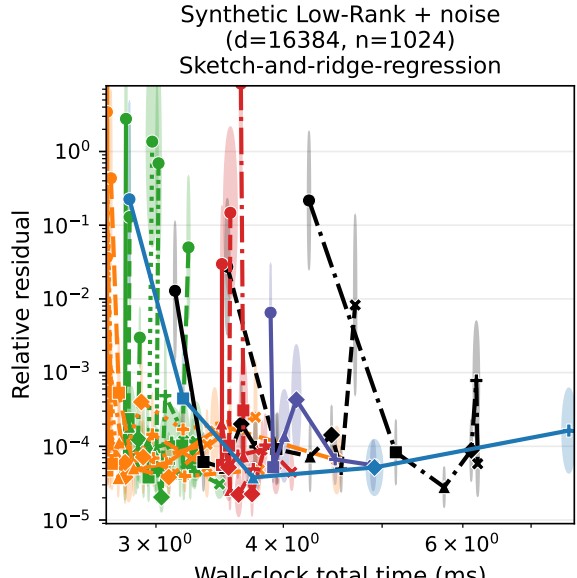

*Figure 103.* **Sketch-and-ridge regression ablations.** Synthetic Gaussian (d=16384, n=1024). GPU: NVIDIA RTX A6000.

*Figure 105.* **Sketch-and-ridge regression ablations.** Synthetic Low-Rank + noise (d=16384, n=1024). GPU: NVIDIA RTX A6000.

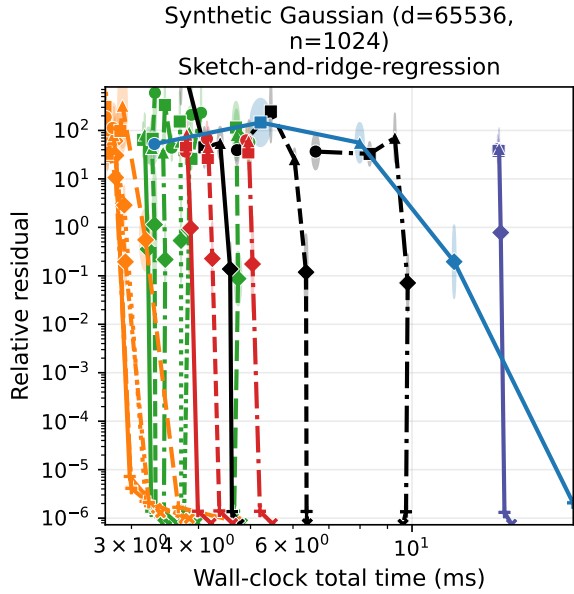

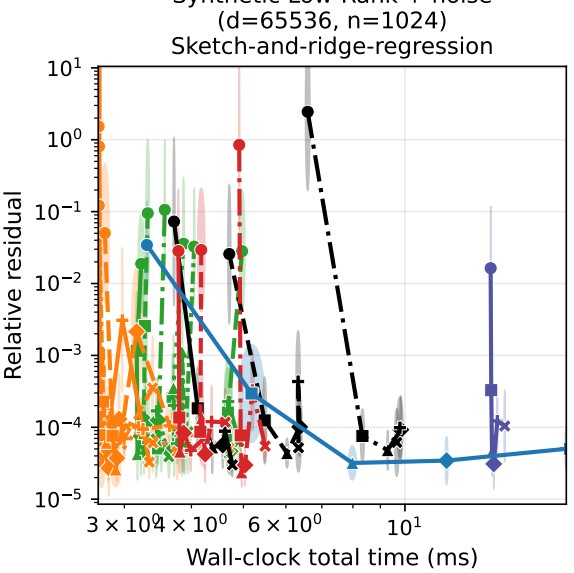

*Figure 104.* **Sketch-and-ridge regression ablations.** Synthetic Gaussian (d=65536, n=1024). GPU: NVIDIA RTX A6000.

*Figure 106.* **Sketch-and-ridge regression ablations.** Synthetic Low-Rank + noise (d=65536, n=1024). GPU: NVIDIA RTX A6000.

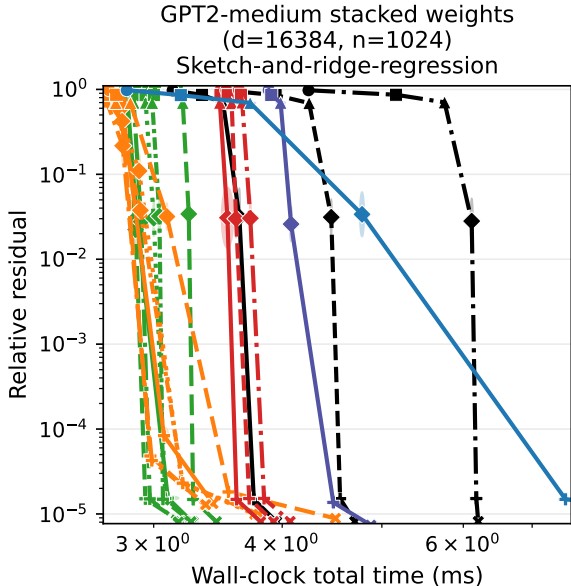

Figure 107. **Sketch-and-ridge regression ablations.** GPT2-medium stacked weights (d=16384, n=1024). GPU: NVIDIA RTX A6000.

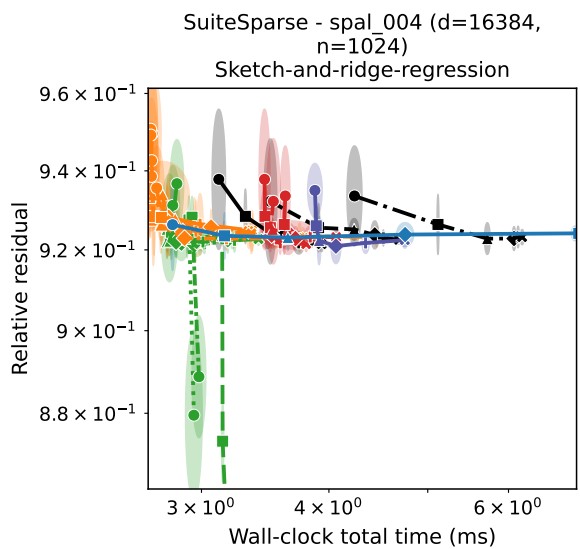

Figure 109. **Sketch-and-ridge regression ablations.** SuiteSparse - spal_004 (d=16384, n=1024). GPU: NVIDIA RTX A6000.

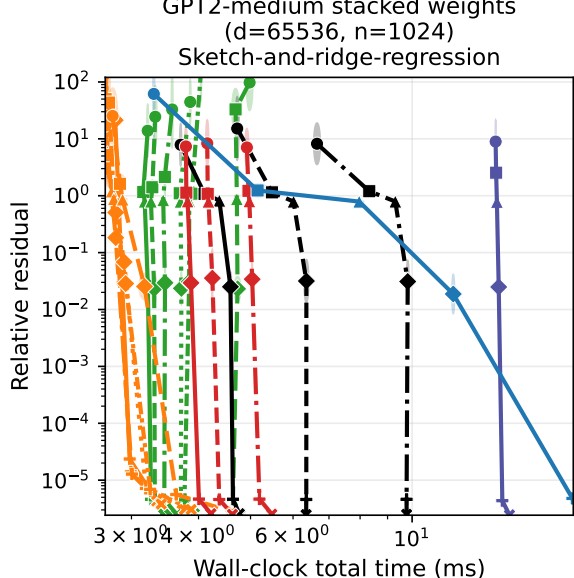

Figure 108. **Sketch-and-ridge regression ablations.** GPT2-medium stacked weights (d=65536, n=1024). GPU: NVIDIA RTX A6000.

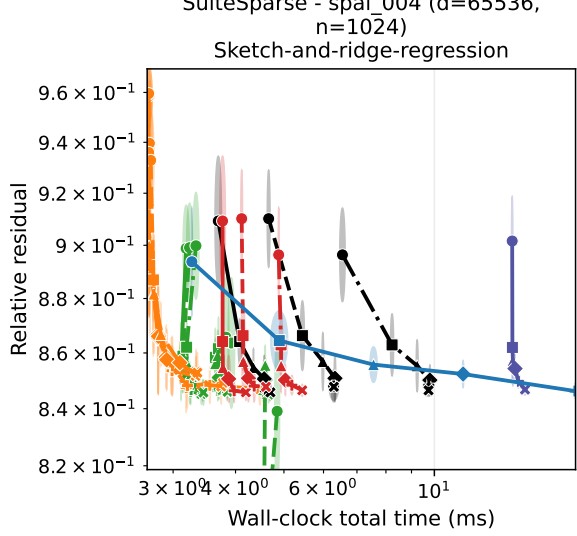

Figure 110. **Sketch-and-ridge regression ablations.** SuiteSparse - spal_004 (d=65536, n=1024). GPU: NVIDIA RTX A6000.

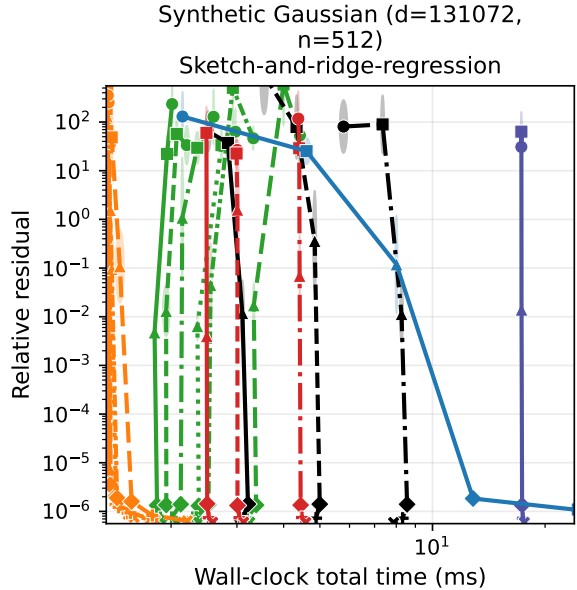

*Figure 111.* **Sketch-and-ridge regression ablations.** Synthetic Gaussian (d=131072, n=512). GPU: NVIDIA RTX A6000.

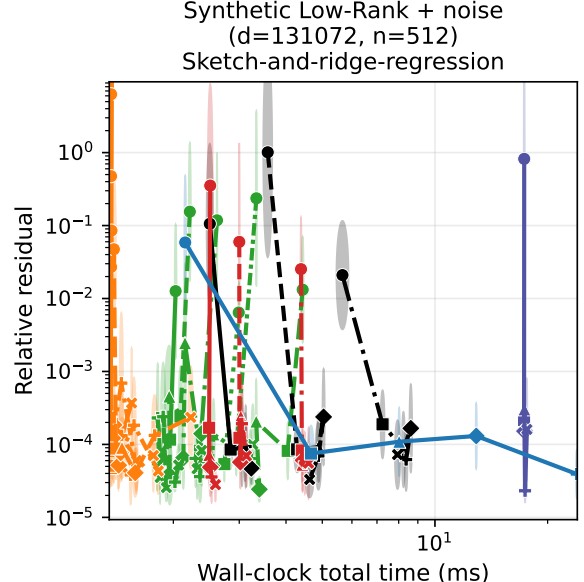

*Figure 113.* **Sketch-and-ridge regression ablations.** Synthetic Low-Rank + noise (d=131072, n=512). GPU: NVIDIA RTX A6000.

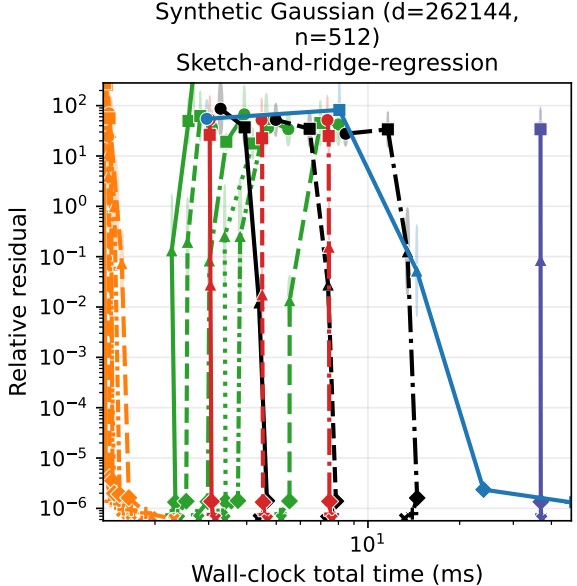

*Figure 112.* **Sketch-and-ridge regression ablations.** Synthetic Gaussian (d=262144, n=512). GPU: NVIDIA RTX A6000.

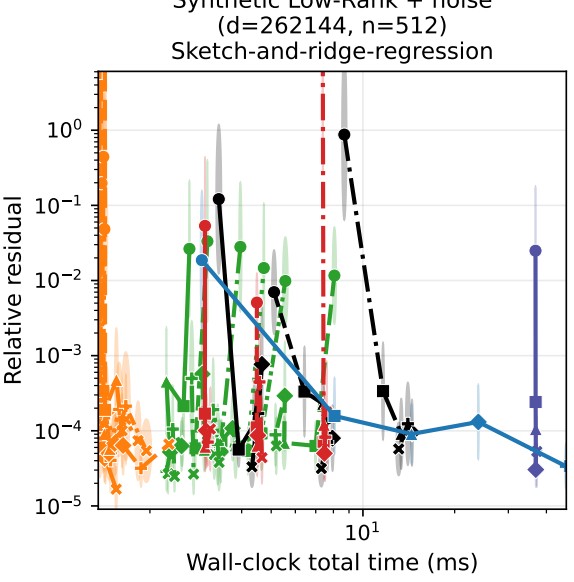

*Figure 114.* **Sketch-and-ridge regression ablations.** Synthetic Low-Rank + noise (d=262144, n=512). GPU: NVIDIA RTX A6000.

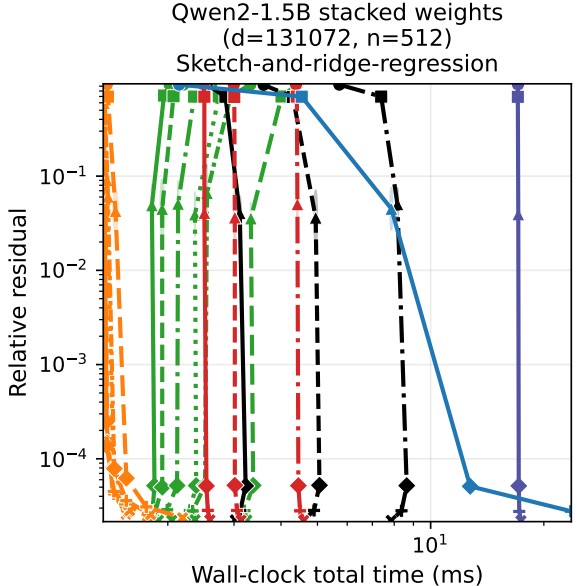

*Figure 115.* **Sketch-and-ridge regression ablations.** Qwen2-1.5B stacked weights (d=131072, n=512). GPU: NVIDIA RTX A6000.

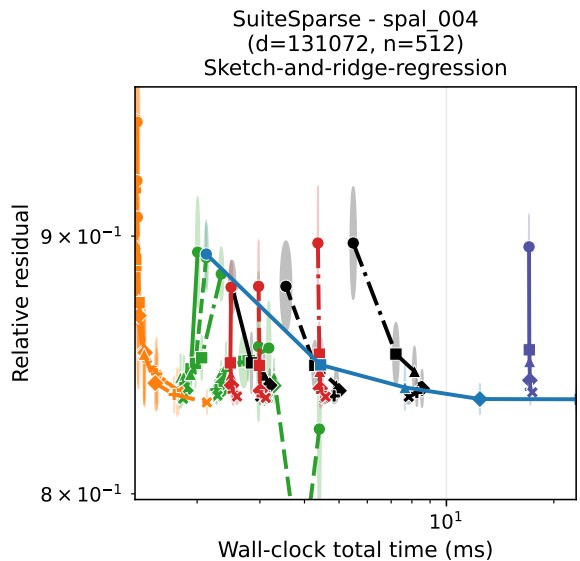

*Figure 117.* **Sketch-and-ridge regression ablations.** SuiteSparse - spal_004 (d=131072, n=512). GPU: NVIDIA RTX A6000.

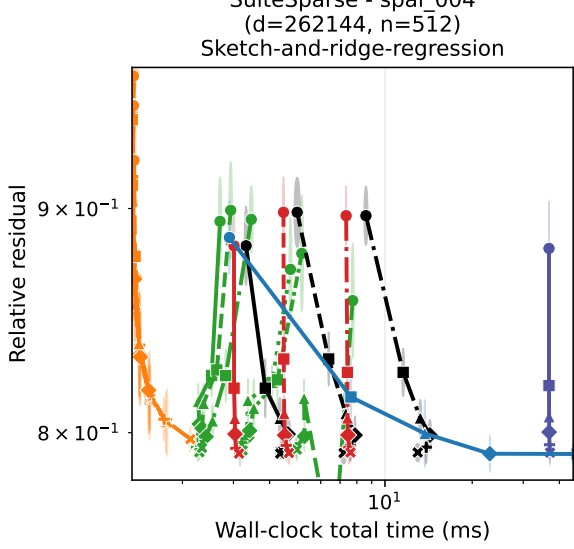

*Figure 118.* **Sketch-and-ridge regression ablations.** SuiteSparse - spal_004 (d=262144, n=512). GPU: NVIDIA RTX A6000.

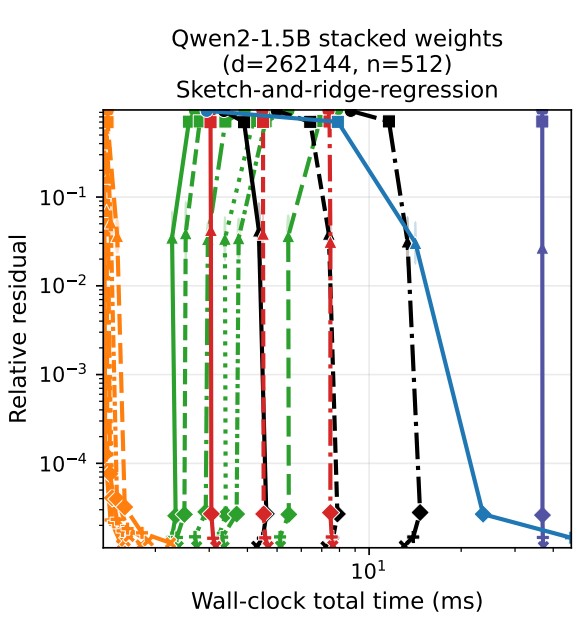

*Figure 116.* **Sketch-and-ridge regression ablations.** Qwen2-1.5B stacked weights (d=262144, n=512). GPU: NVIDIA RTX A6000.

## G.4. Sketch-and-Solve Ablations

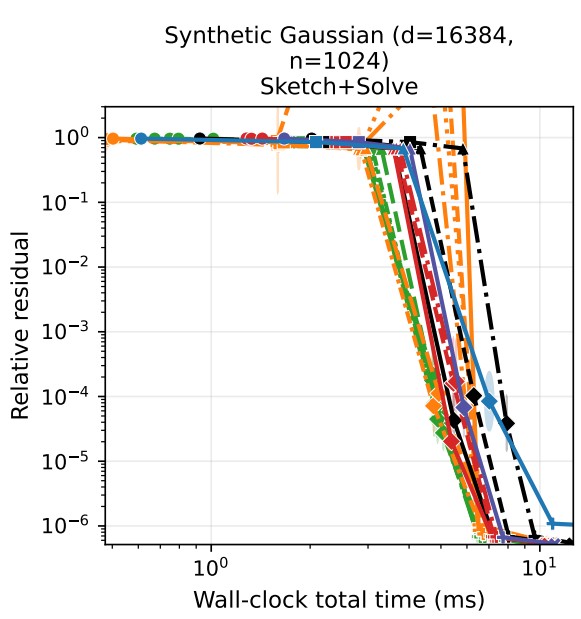

Figure 119. **Sketch-and-solve ablations.** Synthetic Gaussian (d=16384, n=1024). GPU: NVIDIA RTX A6000.

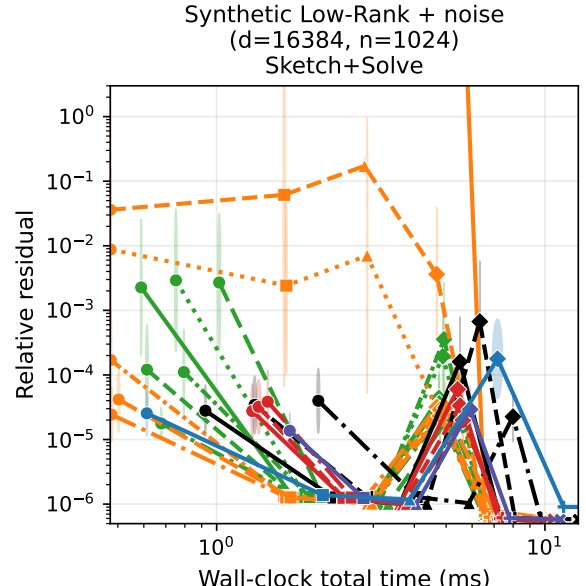

Figure 121. **Sketch-and-solve ablations.** Synthetic Low-Rank + noise (d=16384, n=1024). GPU: NVIDIA RTX A6000.

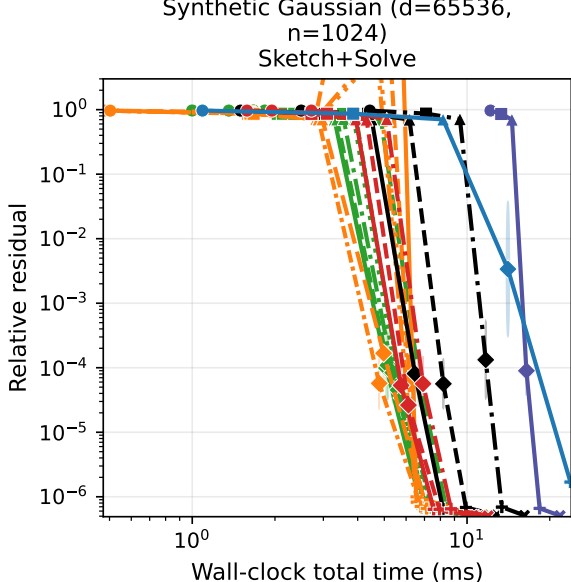

Figure 120. **Sketch-and-solve ablations.** Synthetic Gaussian (d=65536, n=1024). GPU: NVIDIA RTX A6000.

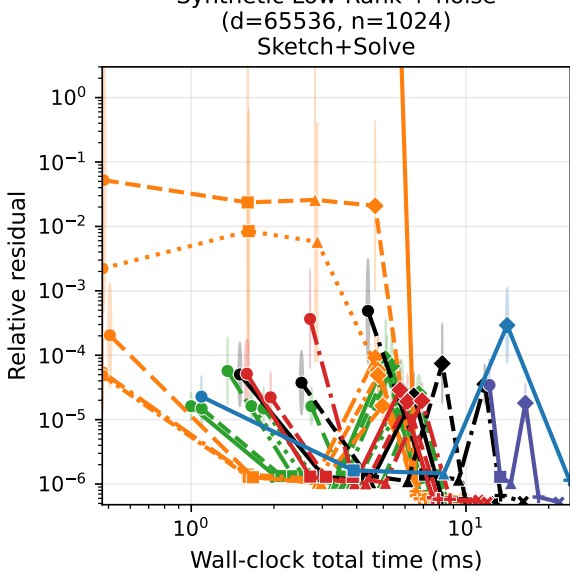

Figure 122. **Sketch-and-solve ablations.** Synthetic Low-Rank + noise (d=65536, n=1024). GPU: NVIDIA RTX A6000.

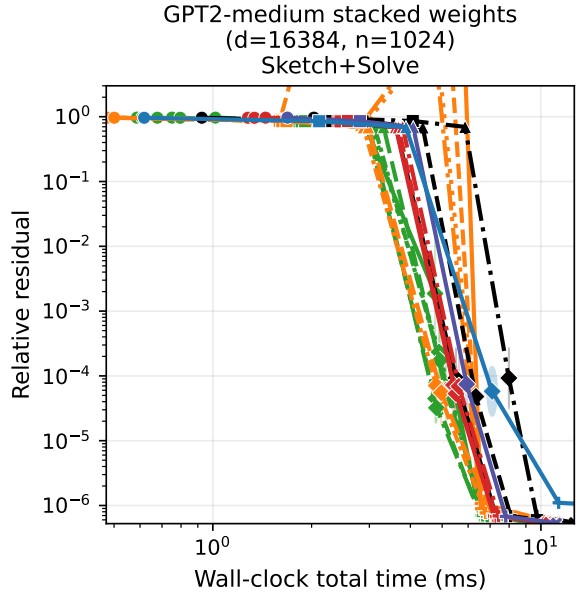

*Figure 123.* **Sketch-and-solve ablations.** GPT2-medium stacked weights (d=16384, n=1024). GPU: NVIDIA RTX A6000.

*Figure 125.* **Sketch-and-solve ablations.** SuiteSparse - spal_004 (d=16384, n=1024). GPU: NVIDIA RTX A6000.

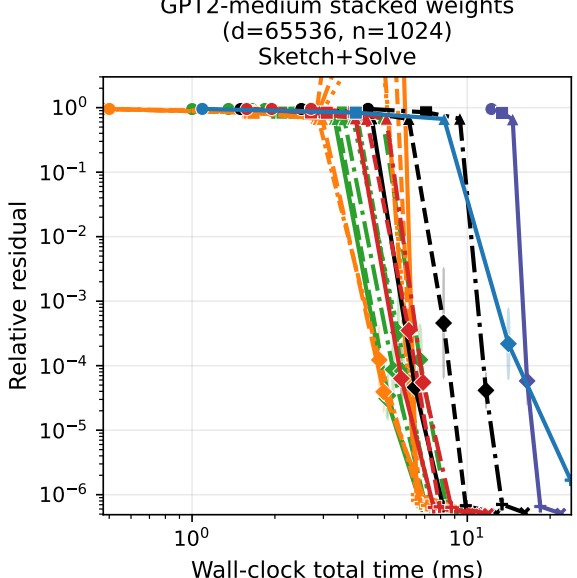
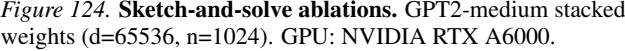

*Figure 124.* **Sketch-and-solve ablations.** GPT2-medium stacked weights (d=65536, n=1024). GPU: NVIDIA RTX A6000.

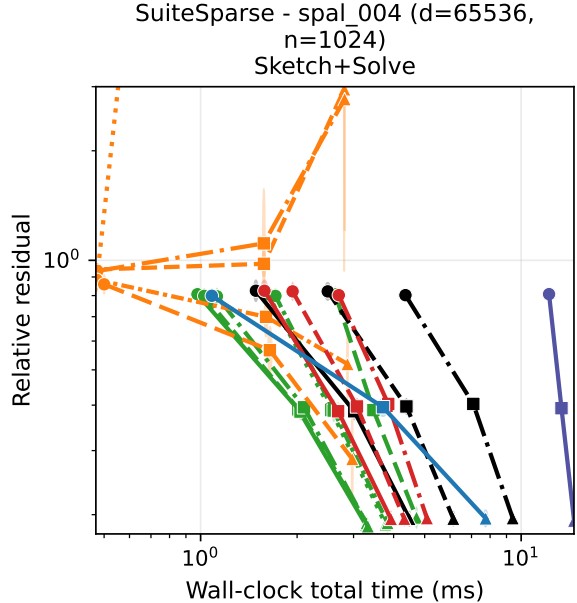

*Figure 126.* **Sketch-and-solve ablations.** SuiteSparse - spal_004 (d=65536, n=1024). GPU: NVIDIA RTX A6000.

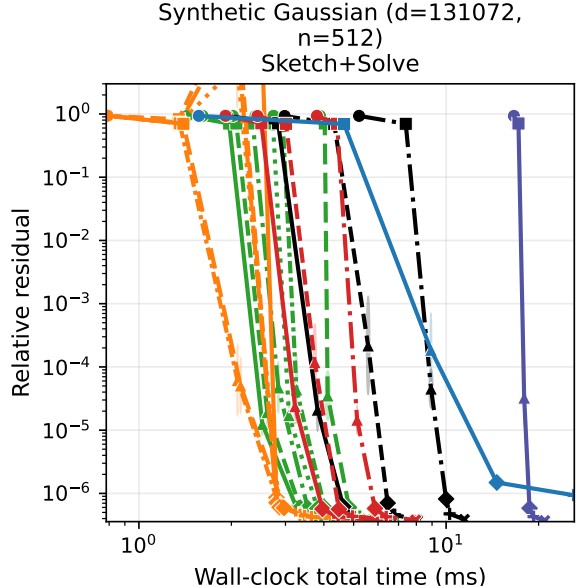

*Figure 127.* **Sketch-and-solve ablations.** Synthetic Gaussian (d=131072, n=512). GPU: NVIDIA RTX A6000.

*Figure 129.* **Sketch-and-solve ablations.** Synthetic Low-Rank + noise (d=131072, n=512). GPU: NVIDIA RTX A6000.

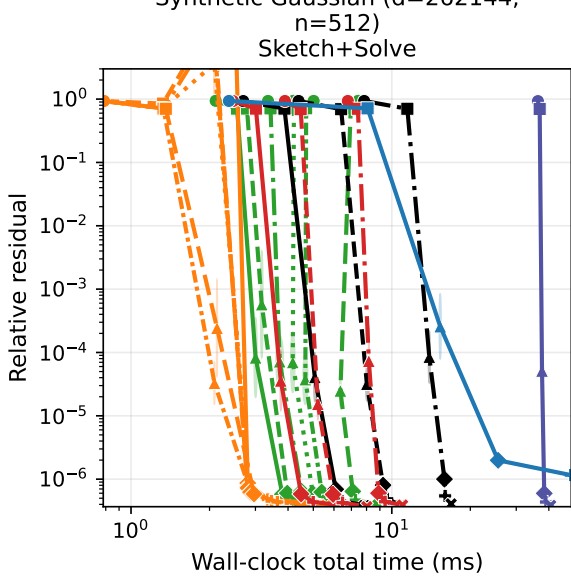

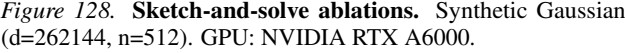

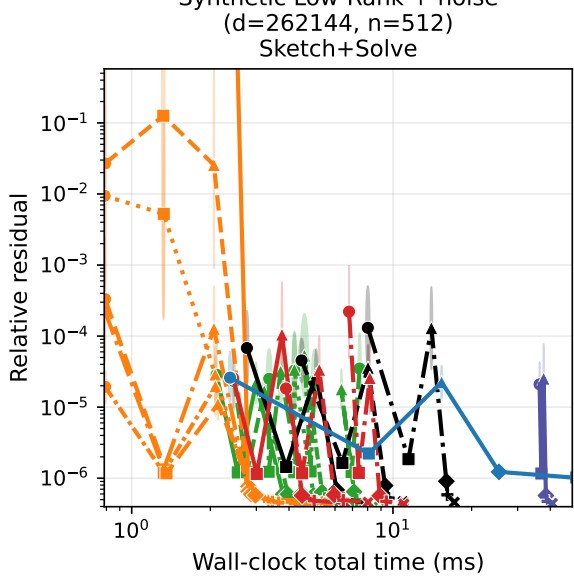

*Figure 128.* **Sketch-and-solve ablations.** Synthetic Gaussian (d=262144, n=512). GPU: NVIDIA RTX A6000.

*Figure 130.* **Sketch-and-solve ablations.** Synthetic Low-Rank + noise (d=262144, n=512). GPU: NVIDIA RTX A6000.

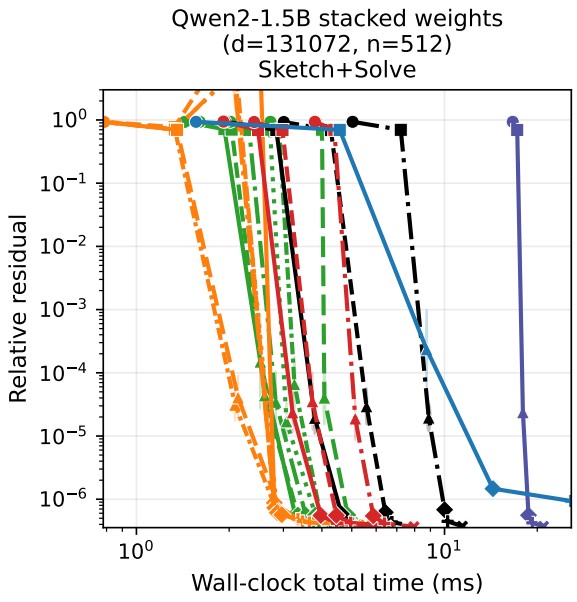

Figure 131. **Sketch-and-solve ablations.** Qwen2-1.5B stacked weights (d=131072, n=512). GPU: NVIDIA RTX A6000.

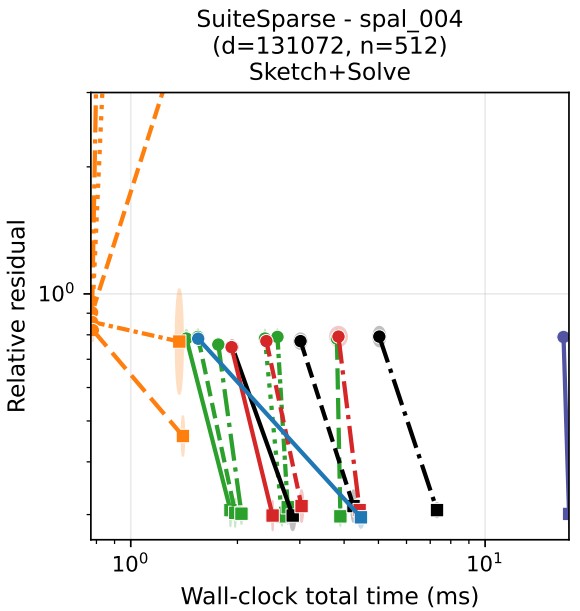

Figure 133. **Sketch-and-solve ablations.** SuiteSparse - spal_004 (d=131072, n=512). GPU: NVIDIA RTX A6000.

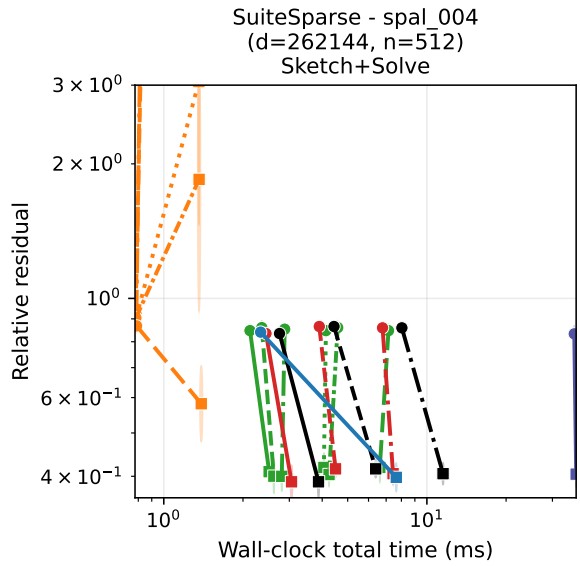

Figure 134. **Sketch-and-solve ablations.** SuiteSparse - spal_004 (d=262144, n=512). GPU: NVIDIA RTX A6000.

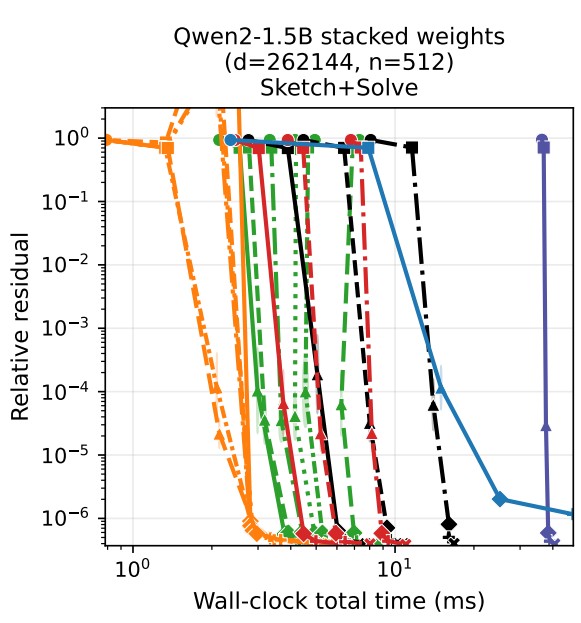

Figure 132. **Sketch-and-solve ablations.** Qwen2-1.5B stacked weights (d=262144, n=512). GPU: NVIDIA RTX A6000.

