# OpenReview forum: "FlashSketch: Sketch-Kernel Co-Design for Fast Sparse Sketching on GPUs"
_ICML.cc/2026/Conference — ICML 2026 spotlight_

### Official Review · Reviewer_h5rN · 2026-03-10

**Soundness:** 3
**Presentation:** 3
**Significance:** 3
**Originality:** 2
**Overall Recommendation:** 4
**Confidence:** 2

**Summary:**

The paper addresses the performance bottleneck of applying sparse sketches on modern GPUs. While these sketches offer theoretical efficiency through sparsity, their random, irregular memory access patterns inherently conflict with the parallel execution and memory hierarchy models of GPUs. To resolve this, the authors propose a hardware-informed co-design approach, which structures sparsity as a union of edge-disjoint permutations. This specific structure enables their custom CUDA kernel, FlashSketch, to eliminate slow global atomic operations in favor of much faster shared-memory atomics.

**Compliance With Llm Reviewing Policy:**

Affirmed.

**Final Justification:**

My concerns have been resolved.

**Key Questions For Authors:**

- Can you briefly compare your work with "SS1: Accelerating Inference with Fast and Expressive Sketch Structured Transform" from the perspective of block-based acceleration?

**Limitations:**

yes

**Strengths And Weaknesses:**

### Strengths

- The demonstrated 1.7ⅹ global geomean speedup over strong baselines seems like a substantial improvement
- Solid mathematical backing for the method.

### Weaknesses

- The novelty of this approach. Sketch itself is not something new, and so does block-based structured sparsity. Further, eliminating global atomic add seems like a quite trivial, engineering effort.

---

> ### Author Rebuttal · Authors · 2026-03-31
>
> We appreciate the reviewer's positive signal on the empirical gain and address the novelty concerns directly below.
>
> ### The novelty of this approach
> We do not claim that sparse sketching, block structure, or avoiding global atomics are individually new. The core claim is that we identify one sketch family that serves both the theory and the kernel: it remains analyzable as an oblivious subspace embedding through neighborhood coherence, while its edge-disjoint permutation structure lets one thread block own an output tile and accumulate locally without global atomics. That alignment between embedding guarantee and GPU execution that we achieve through co-design, together with a kernel that realizes it, is the main novelty of FlashSketch.
>
> ### Why this is more than a trivial engineering effort
> Eliminating atomic adds is an important source of the practical speedup, but the key point is that this became possible because we changed the sketch distribution itself rather than only retuning a kernel for an existing sketch. A kernel-only optimization would still inherit the irregular write pattern of classical sparse sketches, while a theory-only sketch choice could preserve guarantees without removing the main GPU bottleneck. FlashSketch couples the sketch design and the kernel design so that locality improves in a way that is useful both analytically and computationally. That is why we view the result as co-design rather than a standalone implementation tweak.
>
> ### Comparison with SS1 from the perspective of block-based acceleration
> There is a real thematic overlap in that both works introduce structure to better align computation with GPU execution, but the objects being accelerated are different. SS1 replaces learned network layers and relies on model-level adaptation, so its value is tied to an inference pipeline and a learned operator. FlashSketch is an oblivious sparse sketching primitive: it does not require retraining, it is not attached to a specific model family, and it can be dropped into least-squares, regression, Gram-approximation, or GraSS-style pipelines wherever repeated sketch application is the bottleneck. In that sense it is more of a general-purpose GPU sketching primitive, and the contribution is more on the design of the sketch distribution itself rather than the application of a specific learned model. We can include a more explicit comparison in the final revision.

---

> > ### Author Rebuttal · Reviewer_h5rN · 2026-03-31
> >
> > My concerns have been adequately addressed

---

> > > ### Author Response · Authors · 2026-04-07
> > >
> > > We appreciate the reviewer's acknowledgement and positive assessment of our submission. Thank you for taking the time to review our work.

---

### Official Review · Reviewer_kRdg · 2026-03-12

**Soundness:** 3
**Presentation:** 3
**Significance:** 3
**Originality:** 3
**Overall Recommendation:** 5
**Confidence:** 3

**Summary:**

The paper introduces a sketching method tailored for GPU architectures that achieves significant speedups over previous GPU-based sketching approaches. The authors support their claims with both theoretical analysis and extensive experimental evaluation.

**Compliance With Llm Reviewing Policy:**

Affirmed.

**Final Justification:**

The response from the authors was adequate and I will maintain my score.

**Key Questions For Authors:**

1. How should the block degree \kappa and intra-block sparsity s be chosen in practice, and do the authors have guidelines for selecting these parameters based on input data statistics or hardware characteristics?

2. How well does the performance of FLASHSKETCH generalize to other GPU architectures (e.g., different memory hierarchies or tensor-core–optimized pipelines), and are any kernel design choices specific to the tested GPUs?

3. Could the authors further clarify when BLOCKPERM-SJLT is preferable to other fast structured transforms such as SRHT or tensor-core–optimized dense sketches, particularly in regimes where \kappa is small or the input matrix is sparse?

**Limitations:**

Yes

**Strengths And Weaknesses:**

Soundness: The idea and the theory are sound. The experimental results are quite clear and support their claims.

Presentation: The paper is generally well-written and clearly organized, with a logical progression from motivation and design to theory and experiments.

Significance: The paper addresses an important systems–theory gap in randomized numerical linear algebra by demonstrating how sketch design can be co-optimized with GPU kernels.  Improving the efficiency of sketching in GPU systems have practical value. The proposed co-design approach and the resulting speedups could make randomized methods more viable in real-world pipelines, although the impact may depend on how broadly the method generalizes across hardware and application domains.

Originality: Their sketch has both hardware efficiency and theoretical guarantees. They prove oblivious subspace embedding (OSE) guarantees for BLOCKPERM-SJLT, which is novel.

---

> ### Author Rebuttal · Authors · 2026-03-31
>
> We greatly appreciate the reviewer's positive assessment and the concrete questions about parameter choice, portability, and when to prefer `BLOCKPERM-SJLT`.
>
> ### Choosing $\kappa$ and $s$ in practice
> Our practical view is that $\kappa$ is the main quality-throughput knob, while $s$ is a finer intra-block sparsity knob. Larger $\kappa$ improves mixing and reduces the locality penalty in theory, but it also increases input reads, so the right choice is the best point on a quality-throughput Pareto frontier rather than a single closed-form rule. In practice, we first choose $\kappa$ to keep the GPU well utilized without overspending reads, and then use $s$ for finer sparsity control. Primarily these choices are empirically driven. Formalizing this intuition into a more explicit guideline is an interesting future direction that we are currently working on.
>
> ### Generalization across GPU architectures
> The general design motivation of reducing global atomics by localizing computation generalizes very strongly across GPU architectures. The choice of kernel parameters like tile and block sizes is what we empirically tune for the specific GPUs we test on, because these depend on factors like available shared memory and register-file sizes. However, we would like to emphasize that this kind of tuning is standard, and in fact the baselines we compare to, like cuBLAS and cuSparse, are heavily tuned to the specific hardware, and arguably more so than our kernel because they are more mature. This keeps the comparison very fair. Further, the architectures we test on have tensor cores, and baselines like SRHT and dense cuBLAS GEMM heavily leverage and are optimized for these tensor-core pipelines.
>
> ### When `BLOCKPERM-SJLT` is preferable
> The strongest regime is dense $A$, $k \ll d$, and $k$ large enough to keep the GPU busy, because that is where sparse sketching is applied repeatedly and the kernel's locality advantage compounds. Across Appendix F and Appendix G, FlashSketch is better in a large majority of the tested dense settings. The weaker cases are exactly the ones the reviewer highlights: sparse $A$ and some small-$k$ regimes, where SRHT or dense tensor-core-friendly sketches might achieve slightly better quality. However, we would like to emphasize that even in these settings, the main weakness comes only in the quality axis, and FlashSketch still dominates in terms of speed. This ties to our theoretical analysis, which shows that the quality gap to classical sparse sketches narrows as $\kappa$ increases, so in principle one can still use `BLOCKPERM-SJLT` in these regimes by increasing $\kappa$ at the cost of some speed. We can make this discussion more explicit in the final revision.

---

> > ### Author Rebuttal · Reviewer_kRdg · 2026-03-31
> >
> > The authors have responded well to the comments.

---

> > > ### Author Response · Authors · 2026-04-07
> > >
> > > We appreciate the reviewer's acknowledgement and are glad that our responses were well received. Thank you for taking the time to review our work.

---

### Official Review · Reviewer_YqnY · 2026-03-12

**Soundness:** 3
**Presentation:** 3
**Significance:** 4
**Originality:** 3
**Overall Recommendation:** 5
**Confidence:** 4

**Summary:**

This paper introduces FlashSketch, a co-designed sparse sketching method that makes randomized linear algebra more efficient on modern GPUs. The classical sparse sketching methods such as SJLT are theoretically attractive but at odds with efficient implementations on GPUs, since they lead to irregular memory access patterns. To address this, the paper proposed BlockPerm-SJLT, a new family of sketching matrices in which block-level connections are determined by a small number of permutations while each nonzero block has an SJLT structure. Alongside, the paper proposes FlashSketch, a corresponding optimized CUDA kernel that implements these sketches efficiently. The paper then gives a theoretical analysis on the required dimension of the new proposed sketch family. Finally, the paper gives an empirical evaluation which suggests the advantage of the proposed methods.

**Compliance With Llm Reviewing Policy:**

Affirmed.

**Final Justification:**

My concerns have been addressed and I will maintain my score.

**Key Questions For Authors:**

The main theorem (Thm. 6.2) has a term $\mu_{nbr} (U; \pi)$. What is the worst case for this term? And is there some evidence that suggests that this term is necessary?

**Limitations:**

Yes

**Strengths And Weaknesses:**

Strengths

- The paper studies an important problem. Classical sketching methods have strong theoretical guarantees, but their random sparse patterns make efficient GPU implementations challenging. The paper proposes a new family of sketching matrices that have a block-level SJLT structure, and complements this design with theoretical analysis. In my view, this paper provides a good starting point for exploring this direction.

- The paper gives a detailed empirical evaluation of the proposed sketch method.

- The organization and presentation of this paper are generally good and clear.

Weaknesses

- I do not see a serious weakness at this moment, but I feel like it would be better if the authors can give a few more details about the construction of the sketching matrices in Section 4 to improve the presentation.

- There are still some aspects of the theoretical part that need to be better understood. For example, in the implementation, the paper uses a different method for generating the permutations. And it is not very clear how the proposed dimension bound compares with those of classical sketching methods.

---

> ### Author Rebuttal · Authors · 2026-03-31
>
> We are glad the reviewer found the direction important and the empirical study useful, and we greatly appreciate their positive feedback.
>
> ### What is the worst case for $\mu_{nbr}(U; \pi)$, and is this term necessary?
> In an absolute sense, the worst case is a 1-dimensional subspace concentrated inside one input block. If $U = e_i$ for a coordinate in a single block, then any neighborhood containing that block has operator norm 1, so $\mu_{nbr}(U; \pi) = M / \kappa$, which is the largest value possible because every row-restricted submatrix of an orthonormal $U$ has operator norm at most 1.
>
> Relative to the standard block coherence $\mu_{blk}(U)$, it is possible to adversarially construct a $U$ that has $\mu_{nbr}(U; \pi) = \mu_{blk}(U)$ for any *fixed* permutation $\pi$
> by extending the 1-dimensional example above to have every block in the permutation contain one coordinate of $U$ with value 1. This is basically the instance that achieves the upper bound $\mu_{nbr}(U; \pi) \le \mu_{blk}(U)$ in Eq. (7). This is exactly the motivation behind randomizing the permutations, which yields a stronger guarantee that is formalized in Theorem A.11 in the appendix.
>
> Since there exist example subspaces that do achieve the worst case value of $\mu_{nbr}(U; \pi)$, this term is necessary in the sense that it cannot be removed from the bound in the current theoretical framing, which is similar to how the block coherence $\mu_{blk}(U)$ is necessary in the classical localized sketching bounds. The main contribution is that by randomizing the permutations, we can get a stronger guarantee that $\mu_{nbr}(U; \pi)$ is likely to be much smaller than $\mu_{blk}(U)$. We hope this addresses the reviewer's question, and we would be happy to elaborate further in the discussion period if needed.
>
>
> ### More detail on the sketch construction and implementation permutations
> Thank you for the specific feedback on this section. We can include a crisper description of the sketch construction motivated by the theory and GPU architecture together in our final revision of Section 4. Regarding the permutations, at the theorem level, the block connections are defined by independent random permutations because that gives the cleanest analysis. In the kernel, we instantiate a lightweight family of distinct permutations that preserves the same block-level ownership pattern needed to avoid global atomics, but is cheaper to generate and store on GPU. An important intuitive point that we would like to emphasize is that this structure of edge-disjoint permutations actually encourages a more robust sketch than independent permutations, which could result in colliding blocks. Formalizing this intuition into theory is an interesting future direction that we are currently working on.
>
> ### How does the dimension bound compare with classical sketching methods?
> The dimension bound is built on existing literature as described in the related work section. Specifically, localized sketching effectively studies the $\kappa=1$ case and provides a dimension bound, which itself is heavily based on the existing classical SJLT/OSNAP-style bounds. Our contribution is to extend this analysis to `BlockPerm-SJLT`, via introducing the new neighborhood coherence quantity $\mu_{nbr}$, which appears as a multiplicative penalty factor on the classical SJLT/OSNAP-style bounds. As $\kappa$ increases, this term decreases, so the gap to classical sparse sketches narrows. We concretely characterize how this gap scales with $\kappa$ in Theorem A.11 in the appendix. We can make the exposition of this direct comparison clearer in the final revision.

---

> > ### Author Rebuttal · Reviewer_YqnY · 2026-04-02
> >
> > My concerns have been addressed.

---

> > > ### Author Response · Authors · 2026-04-07
> > >
> > > We sincerely appreciate the reviewer's positive assessment of our work and thoughtful questions. Thank you for taking the time to review our work.

---

### Official Review · Reviewer_Qakc · 2026-03-13

**Soundness:** 3
**Presentation:** 3
**Significance:** 3
**Originality:** 3
**Overall Recommendation:** 4
**Confidence:** 3

**Summary:**

Authors consider Subspace embeddings and sketches from a perspective that focuses on modern GPUs. THey design a new family of sparse sketches, BLOCKPERM-SJLT, whose sparsity structure is chosen to enable FLASHSKETCH, acorresponding optimized CUDA kernel that implements these sketches efficiently.
More precisely theeir sketching algorithm uses a block structure where the dataset as well as the sketch are divided into $M$ blocks. Further each input block is connected to exactly $\kappa$ output blocks and vice verasa via permutations which induces a set of edges. Then they draw an independent sparse JL matrix for each edge. The advantage of FLASHSKETCH is that it performs no global atomics and only $\theta(k n) $ global writes as and collisions are rare if $\kappa$ is large enough
Handeling the parameters is one of the challenges.
They also provide experiments showing that their sketch is 1.83 times faster than the best previous sketch.

**Compliance With Llm Reviewing Policy:**

Affirmed.

**Final Justification:**

As mentioned in my response I feel not competent enough in the field to raise my score but my tendency to accept remains. However I appreciate the asnwer to the question in the rebuttal.

**Key Questions For Authors:**

How significant is the empirical improvement?

**Limitations:**

yes

**Strengths And Weaknesses:**

The paper is well written and well structured.
The submission seems technically sound and claims well supported by theoretical analysis as well as experiments. I did not check the details but everything I read sounded plausible to me.
The ideas are not new but they are combined cleverly.
I cannot rate the significance of the results as I am only familiar with sketching from a theoretical perspective that does not consider GPUs.

---

> ### Author Rebuttal · Authors · 2026-03-31
>
> We appreciate the reviewer's positive assessment and the request to better contextualize the empirical gains.
>
> ### How significant is the empirical improvement?
> The speedups are large enough to matter operationally, not just numerically: a 1.73x global geomean improvement over the next best baseline, and a 1.41x-4.17x gain over the prior GPU SJLT kernel. This means that sketching is much less likely to remain the dominant bottleneck when it is applied repeatedly inside least-squares, regression, Gram-approximation, and GraSS-style pipelines. We would also like to emphasize that this is not coming from a single cherry-picked point. Table 1 and Appendices F-G show the same pattern across numerous tasks, shapes, and datasets, while Section 7.4 already shows up to about 3.2x end-to-end speedup in GraSS while maintaining LDS quality.
> Further, these speedups are quite significant when we observe them in the context of pure GPU kernel optimization, where even a 10-20% improvement can be considered a major win. The fact that we achieve a 1.73x improvement over a strong baseline is empirical evidence that our co-design approach produces non-trivial gains by not relying purely on kernel optimization but also on a sketch design that is more amenable to GPU execution.
>
> We would be happy to provide further technical detail on the improvement in any specific experiment or empirical setting in the discussion period.

---

> > ### Author Rebuttal · Reviewer_Qakc · 2026-04-01
> >
> > Thank you for the detailed answer! I feel not competent enough in the field to raise my score but my tendency to accept remains.

---

> > > ### Author Response · Authors · 2026-04-07
> > >
> > > We appreciate the reviewer's acknowledgement and positive assessment of our paper. Thank you for taking the time to review our work.

---

### Decision · Program_Chairs · 2026-04-30

**Decision:**

Accept (spotlight)

**Comment:**

The paper presents sparse sketches (methods, theorems and implementations) for linear-algebraic problems such as regression, targeted towards GPU architectures. The proposed class of sketches, called BlockPerm-SJLT, allow for a tradeoff between speed and accuracy. Experimental evaluation on multiple tasks shows an average speed up of close to 2x over best existing baselines, including dense methods.

All reviewers recommended acceptance. They appreciated the importance of the problem (making sparse sketches work on GPUs), careful experimental design, robust empirical speedups and solid theoretical backing of the proposed sketching method. There were some concerns, including about limited novelty of the approach, but the authors addressed them during the rebuttal, and will add further discussion and description in the final revision.

Altogether, a very nice paper, with contributions ranging from theory to robust implementation, for problems of broad interest in the machine learning community.